# Phase retrieval in high dimensions:
# Statistical and computational phase transitions

**Antoine Maillard**[1]     **Bruno Loureiro**[2,3]     **Florent Krzakala**[1,3]     **Lenka Zdeborová**[2,4]

[1] Lab. de Physique de l'École Normale Supérieure, PSL, CNRS & Sorbonne Universités, Paris.
[2] Institut de Physique Théorique, CNRS, CEA, Université Paris-Saclay, France.
[3] IdePHICS laboratory, EPFL, Switzerland.
[4] SPOC laboratory, EPFL, Switzerland.
`antoine.maillard@ens.fr`

## Abstract

We consider the *phase retrieval* problem of reconstructing a $n$-dimensional real or complex signal $\mathbf{X}^\star$ from $m$ (possibly noisy) observations $Y_\mu = |\sum_{i=1}^n \Phi_{\mu i} X_i^\star / \sqrt{n}|$, for a large class of correlated real and complex random sensing matrices $\mathbf{\Phi}$, in a high-dimensional setting where $m, n \to \infty$ while $\alpha = m/n = \Theta(1)$. First, we derive sharp asymptotics for the lowest possible estimation error achievable statistically and we unveil the existence of sharp phase transitions for the weak- and full-recovery thresholds as a function of the singular values of the matrix $\mathbf{\Phi}$. This is achieved by providing a rigorous proof of a result first obtained by the replica method from statistical mechanics. In particular, the information-theoretic transition to perfect recovery for full-rank matrices appears at $\alpha = 1$ (real case) and $\alpha = 2$ (complex case). Secondly, we analyze the performance of the best-known polynomial time algorithm for this problem — approximate message-passing— establishing the existence of a statistical-to-algorithmic gap depending, again, on the spectral properties of $\mathbf{\Phi}$. Our work provides an extensive classification of the statistical and algorithmic thresholds in high-dimensional phase retrieval for a broad class of random matrices.

## 1 Introduction

Consider the reconstruction problem of a real or complex signal from $m$ observations of its modulus

$$Y_\mu = \left| \frac{1}{\sqrt{n}} \sum_{i=1}^n \Phi_{\mu i} X_i^\star \right|, \qquad \mu = 1, \cdots, m, \tag{1}$$

where the $m \times n$ *sensing* matrix $\mathbf{\Phi} \in \mathbb{K}^{m \times n}$ is known, with $\mathbf{X}^\star \in \mathbb{K}^n$ ($\mathbb{K} \in \{\mathbb{R}, \mathbb{C}\}$). More generally, measurements can be a noisy function of the modulus, for example by an additive Gaussian noise. This inverse problem, known in the literature under the umbrella of *phase retrieval*, is relevant to a series of signal processing [1–3] and statistical estimation [4–7] tasks. It appears in setups in optics and crystallography where detectors can often only measure information about the amplitude of signals, thus losing the information about its phase. It is also a challenging example of a non-convex problem and non-convex optimization with a complex loss landscape [8–10]. Here we are interested in understanding the fundamental limitations of phase retrieval. We focus on the following questions:

   i) What is the *lowest possible error* one can get in estimating the signal $\mathbf{X}^\star$?
   ii) What is the *minimal* number of measurements needed to produce an estimator positively correlated with the signal (that is with non-trivial error in the $n, m \to \infty$ limit)?
   iii) How to *efficiently* reconstruct $\mathbf{X}^*$ *in practice* with a polynomial time algorithm?

We provide a sharp answer to these questions for a large set of random sensing matrices $\mathbf{\Phi}$ that hold with high probability in the *high-dimensional limit* where $m, n \to \infty$ keeping the rate $\alpha = m/n$ fixed.

**Main contributions and related work —** There has been an extensive amount of work on phase retrieval with random matrices. The performance of the Bayes-optimal estimator has been heuristically derived for real orthogonally invariant matrices $\mathbf{\Phi}$ and real signals drawn from generic but separable distributions [11, 12]. Results for the i.i.d. (real) Gaussian matrix case were rigorously proven in [13], where the algorithmic gap is also studied. This analysis was later non-rigorously extended to the case of non-separable prior distributions [14]. The weak-recovery transition discussed here was studied in detail in [15, 16] for i.i.d. Gaussian matrices, while the case of unitary-column matrices was discussed in [17–19]. Our analysis extends these results by considering arbitrary matrices with orthogonal or unitary invariance properties, encapsulating all the cases described above. Message passing algorithms, in particular the generalized vector-approximate message-passing (G-VAMP), have been studied in [20, 21]. In the present setting these algorithms are conjectured to be optimal among all polynomial-time ones. To test the performance of the G-VAMP algorithm, we used the TrAMP library [22] that provides an open-source implementation. In the present work we derive sharp asymptotics for the lowest possible estimation error achievable statistically and algorithmically, locate the phase transitions for weak- and full-recovery as a function of the singular values of the matrix $\mathbf{\Phi}$ and also discuss the existence of a statistical-to-algorithmic gap. Our main contributions are:

- We extend the results of [12] to the complex case, by using the heuristic replica method from statistical physics to derive a unified single-letter formula for the performance of the Bayes-optimal estimator under a separable signal distribution $P_0$, and for $\mathbf{\Phi}$ taken from a right-orthogonally (unitarily in the complex case) invariant ensemble with arbitrary spectrum.
- We rigorously prove the aforementioned formula in two particular cases. First, when the distribution $P_0$ is Gaussian (real or complex) and $\mathbf{\Phi} = \mathbf{WB}$ is the product of a Gaussian matrix $\mathbf{W}$ with an arbitrary matrix $\mathbf{B}$. Second, for a Gaussian matrix $\mathbf{\Phi}$ (real or complex) with any separable distribution $P_0$. These are non-trivial extensions of the the proofs of [13, 23–25].
- In the $n, m \to \infty$ limit, with $\alpha = m/n = \Theta(1)$, we identify (as a function of the singular values distribution of $\mathbf{\Phi}$) the *algorithmic weak-recovery* threshold $\alpha_{\mathrm{WR,Algo}}$ above which better-than-random inference reconstruction of $\mathbf{X}^\star$ is possible in polynomial time.
- We establish the *information-theoretic full recovery* threshold $\alpha_{\mathrm{FR,IT}}$ above which full reconstruction of $\mathbf{X}^\star$ (meaning that the recovery is perfect up to the possible rank deficiency of $\mathbf{\Phi}$) is statistically possible, as a function of the singular values distribution of $\mathbf{\Phi}$.
- We provide a measure of the intrinsic algorithmic hardness of phase retrieval by studying the performance of the G-VAMP algorithm, which can be rigorously tracked for orthogonally (unitarily) invariant $\mathbf{\Phi}$ [20, 21]. We use this rigorous analysis to numerically establish the existence or absence of a statistical-to-algorithmic gap for reconstruction in the following cases $\mathbf{\Phi} \in \{$real/complex Gaussian, orthogonal/unitary, product of complex Gaussians$\}$, for which such an analysis was, to the best of our knowledge, lacking.

Our findings for the statistical and algorithmic thresholds are summarized in Table 1, for different real and complex ensembles of $\mathbf{\Phi}$. Entries in bold emphasize new results obtained in this manuscript, filling a gap between the different previous works in the phase retrieval literature.

Throughout the manuscript we adopt the following notation. Let $\beta \in \{1, 2\}$. We denote $\mathbb{K} = \mathbb{R}$ if $\beta = 1$ and $\mathbb{K} = \mathbb{C}$ if $\beta = 2$. $\mathcal{U}_\beta(n)$ denotes the orthogonal (respectively unitary) group. For $m \geq n$, a matrix $\mathbf{A} \in \mathbb{K}^{m \times n}$ is said to be *column-orthogonal (unitary)* if $\mathbf{A}^\dagger \mathbf{A} = \mathbb{1}_n$. For $x, y \in \mathbb{K}$, we define a 'dot product' as $x \cdot y \equiv xy$ if $\mathbb{K} = \mathbb{R}$ and $x \cdot y \equiv \mathrm{Re}[\overline{x}y]$ if $\mathbb{K} = \mathbb{C}$. In particular $x \cdot x = |x|^2$. The Gaussian measure $\mathcal{N}_\beta(0, 1)$ is defined as $\mathcal{D}_\beta z \equiv (2\pi/\beta)^{-\beta/2} \exp(-\beta|z|^2/2) \, \mathrm{d}z$ and $\mathrm{D}_{\mathrm{KL}}$ is the Kullback–Leibler divergence. $\nu$ will denote the asymptotic spectral density of $\mathbf{\Phi}^\dagger \mathbf{\Phi}/n$ and we designate $\langle f(\lambda) \rangle_\nu \equiv \int \nu(\mathrm{d}\lambda) f(\lambda)$ the linear statistics of $\nu$.

**Some consequences of our results —** We list here some interesting (and often surprising) consequences of our analysis. Since our rigorous results concern a subclass of orthogonally invariant matrices, proving and/or interpreting these statements more generally is an interesting future direction.

- One sees from eq. (11) that maximizing $\alpha_{\mathrm{WR,Algo}}$ implies maximizing $\langle \lambda \rangle_\nu^2 / \langle \lambda^2 \rangle_\nu$. The highest ratio is reached when $\nu$ is a delta distribution: for any symmetric channel and prior (see (10)) the ensemble that maximizes $\alpha_{\mathrm{WR,Algo}}$ is thus the one of uniformly-sampled column-orthogonal

| Matrix ensemble and value of $\beta$ | $\alpha_{\mathrm{WR,Algo}}$ | $\alpha_{\mathrm{FR,IT}}$ | $\alpha_{\mathrm{FR,Algo}}$ |
|---|---|---|---|
| Real Gaussian $\boldsymbol{\Phi}$ ($\beta = 1$) | 0.5 [15, 16] | 1 [26] | $\simeq 1.12$ [13] |
| Complex Gaussian $\boldsymbol{\Phi}$ ($\beta = 2$) | 1 [15, 16] | **2** | **$\simeq 2.027$** |
| Real column-orthogonal $\boldsymbol{\Phi}$ ($\beta = 1$) | **1.5** | 1 [26] | **$\simeq 1.584$** |
| Complex column-unitary $\boldsymbol{\Phi}$ ($\beta = 2$) | 2 [17, 18] | **2** | **$\simeq 2.265$** |
| $\boldsymbol{\Phi} = \mathbf{W}_1\mathbf{W}_2$ ($\beta = 1$, aspect ratio $\gamma$) | $\gamma/(2(1 + \gamma))$ [14] | $\min(1, \gamma)$ [26] | Thm. 2.2 [14] |
| $\boldsymbol{\Phi} = \mathbf{W}_1\mathbf{W}_2$ ($\beta = 2$, aspect ratio $\gamma$) | **$\gamma/(1 + \gamma)$** | **$\min(2, 2\gamma)$** | **Thm. 2.2** |
| $\boldsymbol{\Phi}, \beta \in \{1,2\}, \mathrm{rk}[\boldsymbol{\Phi}^\dagger\boldsymbol{\Phi}]/n = r$ | **Eq. (13)** | **$\beta r$** | **Conj. 2.1** |
| Gauss. $\boldsymbol{\Phi}, \beta \in \{1,2\}$, symm. $P_0, P_{\mathrm{out}}$ | Eq. (12) [15, 16] | **Thm. 2.2** | **Thm. 2.2** |
| $\boldsymbol{\Phi}, \beta \in \{1,2\}$, symm. $P_0, P_{\mathrm{out}}$ | **Eq. (11)** | **Conj. 2.1** | **Conj. 2.1** |

Table 1: Values of the algorithmic weak recovery, information-theoretic full recovery, and algorithmic full recovery thresholds for several random matrix ensembles. When the ensemble of $\boldsymbol{\Phi}$ is not specified, we consider any right-orthogonally (unitarily) invariant ensemble with well-defined asymptotic spectral density. The last two lines are given for any symmetric (cf eq. (10)) prior $P_0$ and channel $P_{\mathrm{out}}$, while all other results are for Gaussian $P_0$ and a noiseless phase retrieval channel. We reference results of this manuscript when the value is not given by a closed-form expression, but can be computed from the formulas herein. In some particular ensembles, we have numerically analyzed these equations in Section 4. The new results obtained in our work are written in bold style, and we give references to papers in which the previously known thresholds were computed.

($\beta\!=\!1$) or column-unitary ($\beta\!=\!2$) matrices. Conversely, $\alpha_{\mathrm{WR,Algo}}$ can be made arbitrarily small using a product of many Gaussian matrices, both in the real and complex cases.

- In complex noiseless phase retrieval the information-theoretic weak-recovery threshold for column-unitary matrices is located at $\alpha_{\mathrm{WR,IT}} = 2$ [18]. Our results (Table 1) imply that this corresponds to an "all-or-nothing" transition located precisely at $\alpha\!=\!2$. Moreover, the derivations of $\alpha_{\mathrm{WR,Algo}}$ and $\alpha_{\mathrm{FR,IT}}$ in Sections 3,4 show that for any complex matrix $\alpha_{\mathrm{WR,Algo}} = 2\langle\lambda\rangle_\nu^2/\langle\lambda^2\rangle_\nu \leq \alpha_{\mathrm{FR,IT}} = 2(1 - \nu(\{0\}))$, with the equality only being attained for $\nu$ a delta distribution. Uniformly sampled column-unitary matrices are thus the only right-unitarily invariant complex matrices which present an "all-or-nothing" transition in complex noiseless phase retrieval (for a Gaussian prior). To the best of our knowledge, this is a first establishment of such a transition in a "dense" problem (as opposed to a sparse setting [27, 28]). Investigating further the existence of these transitions, e.g. as a function of the prior, is left for future work.

- Consider again noiseless phase retrieval with Gaussian prior. For real orthogonal matrices, one has $\alpha_{\mathrm{WR,Algo}}-\alpha_{\mathrm{FR,IT}}\!=\!0.5\!>\!0$. Since $\alpha_{\mathrm{WR,Algo}}$ is a smooth function of the eigenvalue density $\nu$, we expect that the inequality holds for many real random matrix ensembles. However, in the complex case, by our previous point, $\alpha_{\mathrm{WR,Algo}}\!\leq\!\alpha_{\mathrm{FR,IT}}$. The gap thus only occurs in the real setting.

## 2 Analysis of information-theoretically optimal estimation

The phase reconstruction task introduced in eq. (1) belongs to the large class of *generalized linear estimation problems*. In this section, we provide a Bayesian analysis of the statistically optimal estimator $\hat{\mathbf{X}}_{\mathrm{opt}} \in \mathbb{K}^n$ for this general class of problems. In the sections that follow, we draw the consequences for the case of the phase reconstruction problem we are interested in in this manuscript. In the generalized linear model, the goal is to reconstruct a *signal* $\mathbf{X}^\star \in \mathbb{K}^n$, with components drawn i.i.d. from a fixed prior distribution $P_0$ over $\mathbb{K}$, from the *observations* $\mathbf{Y} \in \mathbb{R}^m$ generated as:

$$Y_\mu = \varphi_{\mathrm{out}}\Big(\frac{1}{\sqrt{n}}\sum_{i=1}^n \Phi_{\mu i} X_i^\star, A_\mu\Big), \quad 1 \leq \mu \leq m, \tag{2}$$

where $(A_\mu)_{\mu=1}^m \in \mathbb{K}^m$ are i.i.d. random variables with (known) distribution $P_A$ accounting for a possible noise, $\varphi_{\mathrm{out}}$ is the observation channel and $\boldsymbol{\Phi}$ is a random matrix with elements in $\mathbb{K}$. We let $P_{\mathrm{out}}(\cdot|z)$ denote the probability density function associated to the stochastic function $\varphi_{\mathrm{out}}(z, A)$. Further, we assume that $P_0$ has a second moment given by $\rho \equiv \mathbb{E}[|x|^2] > 0$. Note that the phase reconstruction problem introduced in eq. (1) corresponds to a likelihood $P_{\mathrm{out}}(y|z)$ that only depends on $z$ through $|z|^2$. For instance, for Gaussian additive noise it is explicitly given by $P_{\mathrm{out}}(y|z) = \mathcal{N}_1(y; |z|^2, \Delta)$, while the noiseless case corresponds to the limit $\Delta \downarrow 0$ :

$P_{\text{out}}(y|z) = \delta(y - |z|^2)$. In this work, we consider a large class of random matrices $\mathbf{\Phi}$ distributed as $\mathbf{\Phi} \stackrel{d}{=} \mathbf{USV}^\dagger$, with arbitrary $\mathbf{U} \in \mathcal{U}_\beta(m)$, $\mathbf{V}$ drawn uniformly from $\mathcal{U}_\beta(n)$, and $\mathbf{S}$ the pseudo-diagonal of singular values of $\mathbf{\Phi}$. We assume that the spectral measure of $\mathbf{S}^\mathsf{T}\mathbf{S}/n$ almost surely converges (in the weak sense) [1] to a probability measure $\nu$ with compact support $\text{supp}(\nu) \subset \mathbb{R}_+$. Crucially, we assume that the statistician knows how the observations were generated - i.e. she has access to $P_0, P_{\text{out}}$ and the distribution of $\mathbf{\Phi}$, therefore reducing the problem to the reconstruction of the specific realization of $\mathbf{X}^\star$. In this setting, commonly known as *Bayes-optimal*, the statistically optimal estimator $\hat{\mathbf{X}}$ minimizing the mean-squared error $\text{mse}(\hat{\mathbf{X}}) \equiv ||\hat{\mathbf{X}} - \mathbf{X}^\star||_2^2$ is simply given by the posterior mean $\hat{\mathbf{X}}_{\text{opt}} = \mathbb{E}[\mathbf{x}|\mathbf{Y}]$, where the posterior distribution is explicitly given by:

$$P(\mathrm{d}\mathbf{x}|\mathbf{Y}) \equiv \frac{1}{\mathcal{Z}_n(\mathbf{Y})} \prod_{i=1}^n P_0(\mathrm{d}x_i) \prod_{\mu=1}^m P_{\text{out}}\left(Y_\mu \Big| \frac{1}{\sqrt{n}} \sum_{i=1}^n \Phi_{\mu i} x_i\right). \tag{3}$$

Exact sampling from the posterior is intractable for large values of $n, m \in \mathbb{N}^\star$. However, certain information theoretical quantities are accessible analytically precisely in this limit. Indeed, our first set of results concerns a rigorous evaluation of the mutual information $I(\mathbf{X}^\star; \mathbf{Y}) \equiv \mathrm{D}_{\text{KL}}(P_{X,Y}|P_0 \otimes P_Y)$ between the signal $\mathbf{X}^\star$ and the observations $\mathbf{Y}$ for the generalized linear model in the high-dimensional limit of $n, m \to \infty$ with $m/n \to \alpha > 0$ fixed. This quantity fully characterizes the asymptotic performance of the Bayes-optimal estimator $\hat{\mathbf{X}}_{\text{opt}}$ in high dimensions via the I-MMSE theorem [29].

**Asymptotic mutual information and minimum mean-squared error—** The mutual information between the observations and the hidden variables can be decomposed into two terms:

$$I(\mathbf{X}^\star; \mathbf{Y}|\mathbf{\Phi}) = H(\mathbf{Y}|\mathbf{\Phi}) - H(\mathbf{Y}|\mathbf{X}^\star, \mathbf{\Phi}). \tag{4}$$

The entropy $H(\mathbf{Y}|\mathbf{X}^\star, \mathbf{\Phi}) = \mathbb{E} \ln P(\mathbf{Y}|\mathbf{X}^\star, \mathbf{\Phi}) = -m\mathbb{E} \ln P_{\text{out}}(Y_1|(\mathbf{\Phi}\mathbf{X}^\star)_1/\sqrt{n})$ is easily computed in the high-dimensional limit for a given channel $P_{\text{out}}$:

$$\lim_{n\to\infty} -\frac{1}{n} H(\mathbf{Y}|\mathbf{X}^\star, \mathbf{\Phi}) = \alpha \int_\mathbb{R} \mathrm{d}y \int_\mathbb{K} \mathcal{D}_\beta \xi \, P_{\text{out}}(y|\sqrt{Q_z}\xi) \ln P_{\text{out}}(y|\sqrt{Q_z}\xi), \tag{5}$$

with $Q_z \equiv \rho\langle\lambda\rangle_\nu/\alpha$. Indeed, as $n \to \infty$, the law of $(\mathbf{\Phi}\mathbf{X}^\star)_1/\sqrt{n}$ asymptotically approaches $\mathcal{N}_\beta(0, Q_z)$ by the central limit theorem. The challenge in computing the mutual information therefore reduces to the evaluation of the *free entropy* $H(\mathbf{Y}|\mathbf{\Phi}) = \mathbb{E} \ln \mathcal{Z}_n(\mathbf{Y})$, related to the log-normalization of the posterior. Our first result is a single-letter formula for the asymptotic free entropy density of right-orthogonally (unitarily) invariant sensing matrices:

**Conjecture 2.1.** *Under the assumptions above, the asymptotic free entropy density for the posterior distribution defined in eq.* (3) *with right-orthogonally (unitarily) invariant sensing matrix $\mathbf{\Phi}$ is:*

$$\lim_{n\to\infty} \frac{1}{n} \mathbb{E}_{\mathbf{Y},\mathbf{\Phi}} \ln \mathcal{Z}_n(\mathbf{Y}) = \sup_{q_x \in [0,\rho]} \sup_{q_z \in [0,Q_z]} [I_0(q_x) + \alpha I_{\text{out}}(q_z) + I_{\text{int}}(q_x, q_z)], \tag{6}$$

*where* $I_0(q_x) \equiv \inf_{\hat{q}_x \geq 0} \left[ -\frac{\beta\hat{q}_x q_x}{2} + \mathbb{E}_\xi \mathcal{Z}_0(\sqrt{\hat{q}_x}\xi, \hat{q}_x) \log \mathcal{Z}_0(\sqrt{\hat{q}_x}\xi, \hat{q}_x) \right],$

$$I_{\text{out}}(q_z) \equiv \inf_{\hat{q}_z \geq 0} \left[ -\frac{\beta\hat{q}_z q_z}{2} - \frac{\beta}{2}\ln(\hat{Q}_z + \hat{q}_z) + \frac{\beta\hat{q}_z}{2\hat{Q}_z} \right.$$

$$\left. +\mathbb{E}_\xi \int_\mathbb{R} \mathrm{d}y \, \mathcal{Z}_{\text{out}}\left(y; \sqrt{\frac{\hat{q}_z}{\hat{Q}_z(\hat{Q}_z + \hat{q}_z)}}\xi, \frac{1}{\hat{Q}_z + \hat{q}_z}\right) \log \mathcal{Z}_{\text{out}}\left(y; \sqrt{\frac{\hat{q}_z}{\hat{Q}_z(\hat{Q}_z + \hat{q}_z)}}\xi, \frac{1}{\hat{Q}_z + \hat{q}_z}\right) \right],$$

$$I_{\text{int}}(q_x, q_z) \equiv \inf_{\gamma_x, \gamma_z \geq 0} \left[ \frac{\beta}{2}(\rho - q_x)\gamma_x + \frac{\alpha\beta}{2}(Q_z - q_z)\gamma_z - \frac{\beta}{2}\langle\ln(\rho^{-1} + \gamma_x + \lambda\gamma_z)\rangle_\nu \right]$$

$$- \frac{\beta}{2}\ln(\rho - q_x) - \frac{\beta q_x}{2\rho} - \frac{\alpha\beta}{2}\ln(Q_z - q_z) - \frac{\alpha\beta q_z}{2Q_z}.$$

*We defined $Q_z \equiv \rho\langle\lambda\rangle_\nu/\alpha$ and $\hat{Q}_z \equiv 1/Q_z$, $\xi \sim \mathcal{N}_\beta(0,1)$ and the following auxiliary functions:*

$$\mathcal{Z}_0(b,a) \equiv \mathbb{E}_z\left[P_0(z)e^{-\frac{\beta}{2}a|z|^2+\beta b\cdot z}\right], \qquad \mathcal{Z}_{\mathrm{out}}(y;\omega,v) \equiv \mathbb{E}_z\left[P_{\mathrm{out}}\left(y\Big|\sqrt{v}z+\omega\right)\right], \qquad (7)$$

*with $z \sim \mathcal{N}_\beta(0,1)$. Moreover, the asymptotic minimum mean squared error, achieved by the Bayes-optimal estimator, is equal to $\rho - q_x^\star$, with $q_x^\star$ the solution of the above extremization problem;*

$$\lim_{n\to\infty} \mathrm{MMSE} = \lim_{n\to\infty} \frac{1}{n}\mathbb{E}\|\boldsymbol{X}^\star - \hat{\boldsymbol{X}}_{\mathrm{opt}}\|^2 = \rho - q_x^\star. \qquad (8)$$

This formula, derived in Appendix A using the heuristic (hence the *conjecture*) replica method from statistical physics [30], holds for any separable signal distribution $P_0$ and for any choice of likelihood $P_{\mathrm{out}}$. It extends the formula from [12] to complex signals $\mathbf{X}^\star$ and sensing matrices $\mathbf{\Phi}$. In particular, it also holds in the case of complex matrices $\mathbf{\Phi}$ and real signal $\mathbf{X}^\star$, by adding a constraint on the imaginary part of $\mathbf{X}^\star$ in $P_0$. It also encompasses the case of sparse signals, which is of wide interest in the compressive sensing literature [31–35]. Proving Conjecture 2.1 is a challenging open problem. We provide a significant step by proving Conjecture 2.1 for a broad class of likelihoods $P_{\mathrm{out}}$ and in two settings: a restricted signal distribution $P_0$ and a broad class of real and complex likelihoods and sensing matrices $\mathbf{\Phi}$, or a broad class of prior distribution $P_0$ and (real or complex) Gaussian $\mathbf{\Phi}$.

**Theorem 2.2.** *Let us denote*

*(H0) $\varphi_{\mathrm{out}} : \mathbb{K}^2 \to \mathbb{R}$ is $\mathcal{C}^2$, and $(z,a) \mapsto (\varphi_{\mathrm{out}}(z,a), \partial_z\varphi_{\mathrm{out}}(z,a), \partial_z^2\varphi_{\mathrm{out}}(z,a))$ is bounded.*
*(h1) $P_0$ is a centered Gaussian distribution, without loss of generality $P_0 = \mathcal{N}_\beta(0,1)$.*
*(h2) $\mathbf{\Phi}$ is distributed as $\mathbf{\Phi} \overset{d}{=} \mathbf{WB}/\sqrt{p}$, with $\mathbf{W} \in \mathbb{K}^{m\times p}$ an i.i.d. standard Gaussian matrix, and $\mathbf{B} \in \mathbb{K}^{p\times n}$ an arbitrary matrix (random or deterministic), independent of $\mathbf{W}$. Moreover, as $n \to \infty$, $p/n \to \delta > 0$.*
*(h3) The empirical spectral distribution of $\mathbf{B}^\dagger\mathbf{B}/n$ weakly converges (a.s.) to a compactly-supported measure $\nu_B \neq \delta_0$. Moreover, there is $\lambda_{\max} \geq 0$ such that a.s. $\lambda_{\max}(\mathbf{B}^\dagger\mathbf{B}/n) \to_{n\to\infty} \lambda_{\max}$.*
*(h'1) $P_0$ has a finite second moment, and $\Phi_{\mu i} \overset{\mathrm{i.i.d}}{\sim} \mathcal{N}_\beta(0,1)$.*

*Assume that all $(H0),(h1),(h2),(h3)$ or that all $(H0),(h'1)$ stand. Then Conjecture 2.1 holds with $\nu$ the asymptotic eigenvalue distribution of $\mathbf{\Phi}^\dagger\mathbf{\Phi}/n$[1].*

The proof is based on the adaptive interpolation method[2] [25], and is provided in Appendix D. In particular, Theorem 2.2 allows to rigorously compute the asymptotic minimum mean-squared error (MMSE) achieved by the Bayes-optimal estimator. Theorem 2.2 extends the rigorous results of [13] to a larger class of sensing matrices and to the complex case, including both real orthogonally invariant matrices and the products of i.i.d. Gaussian matrices, heuristically studied respectively in [12] and [14].

**Remark 2.3.** *Following the arguments of [13, 24], hypothesis $(H0)$ can be relaxed to continuity a.e. and the existence of moments of $\varphi_{\mathrm{out}}$, so that our theorem also covers noiseless phase retrieval.*

This single-letter formula reduces the high-dimensional computation of the MMSE to a simple low-dimensional extremization problem. The MMSE as a function of the sample complexity $\alpha$ can be readily computed from eqs. (6) and (8) for a given signal distribution $P_0$ (determining $I_0$), likelihood $P_{\mathrm{out}}$ (determining $I_{\mathrm{out}}$) and spectral density $\nu$ (determining $\hat{I}_{\mathrm{int}}$).

**Statistical vs algorithmic performance —** Conjecture 2.1 and Theorem 2.2 show that the global maximum of the potential in eq. (6) describes the performance of the statistically optimal estimator $\hat{\mathbf{X}}_{\mathrm{opt}}$ for generalized linear estimation. Interestingly, eq. (6) also contains rich information about the algorithmic aspects of this problem. Indeed, it has been shown that the performance of the G-VAMP algorithm, the best-known polynomial time algorithm for this problem, corresponds precisely to the MSE achieved by running gradient descent on the potential in eq. (6) from the trivial initial condition $q_x = q_z = 0$ [20, 21]. In the sections that follow, we exploit this result to derive the thresholds

characterizing the statistical and algorithmic limitations of signal estimation. We adopt the subscript IT for the thresholds related to the Bayes-optimal estimator and Algo for the G-VAMP ones[1].

## 3   Weak-recovery transition

A natural question to ask is: what is the minimum sample complexity $\alpha_{\mathrm{WR,Algo}} \geq 0$ such that for all $\alpha \geq \alpha_{\mathrm{WR,Algo}}$ we can algorithmically reconstruct $\mathbf{X}^\star$ better than a trivial random draw from the known signal distribution $P_0$? Also known as the *algorithmic weak-recovery* threshold, $\alpha_{\mathrm{WR,Algo}}$ can also be characterized in terms of the MSE achieved by G-VAMP:

$$\alpha_{\mathrm{WR,Algo}} \equiv \operatorname*{argmin}_{\alpha \geq 0} \{\mathrm{MSE}_{\mathrm{GVAMP}}(\alpha) < \rho\}.$$

In this section, we establish sufficient conditions for the existence of the algorithmic weak-recovery threshold $\alpha_{\mathrm{WR,Algo}} \geq 0$, and we derive an analytical expression for this threshold.

**G-VAMP State Evolution —**  Recalling that $q_x \in [0, \rho]$, from eq. (8) it is easy to see that the weak-recovery threshold is the smallest sample complexity $\alpha$ such that the potential of eq. (6) has no longer a local maximum in $q_x = 0$. In opposition, the region for which the MSE is maximal ($\mathrm{MSE} = \rho$) corresponds to the existence of a *trivial* maximum in eq. (6) with $q_x = q_z = 0$. The extrema of the potential in eq. (6) can be characterized by the solutions of the following *State Evolution* (SE) equations, obtained by looking at the zero-gradient points:

$$
\begin{cases}
q_x = \mathbb{E}_\xi \mathcal{Z}_0 |f_0|^2, & q_z = \frac{1}{\hat{Q}_z + \hat{q}_z}\big[\frac{\hat{q}_z}{\hat{Q}_z} + \mathbb{E}_\xi \int \mathrm{d}y\, \mathcal{Z}_{\mathrm{out}} |f_{\mathrm{out}}|^2\big], & \text{(9a)} \\[2mm]
\hat{q}_x = \frac{q_x}{\rho(\rho - q_x)} - \gamma_x, & \hat{q}_z = \frac{q_z}{Q_z(Q_z - q_z)} - \gamma_z, & \text{(9b)} \\[2mm]
\rho - q_x = \Big\langle \frac{1}{\rho^{-1} + \gamma_x + \lambda \gamma_z} \Big\rangle_\nu, & \alpha(Q_z - q_z) = \Big\langle \frac{\lambda}{\rho^{-1} + \gamma_x + \lambda \gamma_z} \Big\rangle_\nu. & \text{(9c)}
\end{cases}
$$

where $f_0(b, a) = \partial_b \log \mathcal{Z}_0(b, a)$ and $f_{\mathrm{out}}(y; \omega, v) = \partial_\omega \log \mathcal{Z}_{\mathrm{out}}(y; \omega, v)$ are evaluated at $(b, a) = (\sqrt{\hat{q}_x}\xi, \hat{q}_x)$ and $(\omega, v) = \big(\sqrt{\frac{\hat{q}_z}{\hat{Q}_z(\hat{Q}_z + \hat{q}_z)}}\xi, \frac{1}{\hat{Q}_z + \hat{q}_z}\big)$ respectively. Note in particular that eq. (9c) has to be solved over $(\gamma_x, \gamma_z)$ in order to be iterated. Since the algorithmic performance is characterized by precisely maximizing eq. (6) starting from the trivial point, the algorithmic weak-recovery threshold $\alpha_{\mathrm{WR,Algo}}$ can be analytically computed from a local stability analysis of this point. Note that in general $\alpha_{\mathrm{WR,IT}} \neq \alpha_{\mathrm{WR,Algo}}$ since $q_x = q_z = 0$ can be just a local maximum of eq. (6).

**Existence and location of the weak-recovery threshold —**  It is easy to verify that the state evolution equations (9) admit a trivial fixed point in which $q_x = q_z = \hat{q}_x = \hat{q}_z = \gamma_x = \gamma_z = 0$ when $P_0$ and $P_{\mathrm{out}}$ are *symmetric*, that is when for any $y \in \mathbb{R}$ and $x_1, x_2, z_1, z_2 \in \mathbb{K}$:

$$|x_1| = |x_2| \Rightarrow P_0(x_1) = P_0(x_2) \qquad \text{and} \qquad |z_1| = |z_2| \Rightarrow P_{\mathrm{out}}(y|z_1) = P_{\mathrm{out}}(y|z_2). \quad \text{(10)}$$

In particular, this symmetry condition holds for the phase retrieval likelihood and for Gaussian signals considered here. When it exists, the trivial extremizer $q_x = q_z = 0$ can be a (local) maximum or a minimum, corresponding to whether the trivial fixed point of the state evolution equations is stable or unstable. The weak-recovery threshold can therefore be determined by looking at the Jacobian around the trivial fixed point. The details of the stability analysis are given in Appendix B. The result is that a linear instability of the trivial fixed point appears at $\alpha = \alpha_{\mathrm{WR,Algo}}$ satisfying the equation:

$$\alpha_{\mathrm{WR,Algo}} = \frac{\langle \lambda \rangle_\nu^2}{\langle \lambda^2 \rangle_\nu}\left(1 + \Big[\int_\mathbb{R} \mathrm{d}y\, \frac{\big|\int_\mathbb{K} \mathcal{D}_\beta z\, (|z|^2 - 1)\, P_{\mathrm{out}}\big(y\big|\sqrt{\frac{\rho \langle \lambda \rangle_\nu}{\alpha_{\mathrm{WR,Algo}}}}z\big)\big|^2}{\int_\mathbb{K} \mathcal{D}_\beta z\, P_{\mathrm{out}}\big(y\big|\sqrt{\frac{\rho \langle \lambda \rangle_\nu}{\alpha_{\mathrm{WR,Algo}}}}z\big)}\Big]^{-1}\right). \quad \text{(11)}$$

Note that the integrand and the averages $\langle \cdot \rangle_\nu$ depend on $\alpha_{\mathrm{WR,Algo}}$, so that this is an implicit equation on $\alpha_{\mathrm{WR,Algo}}$. Eq. (11) is the most generic formula for the weak recovery threshold for any data matrix $\mathbf{\Phi}$ and phase retrieval channel $P_{\mathrm{out}}$. As emphasized in the following examples, it generalizes in particular several previously known formulas for different channels and random matrix ensembles.

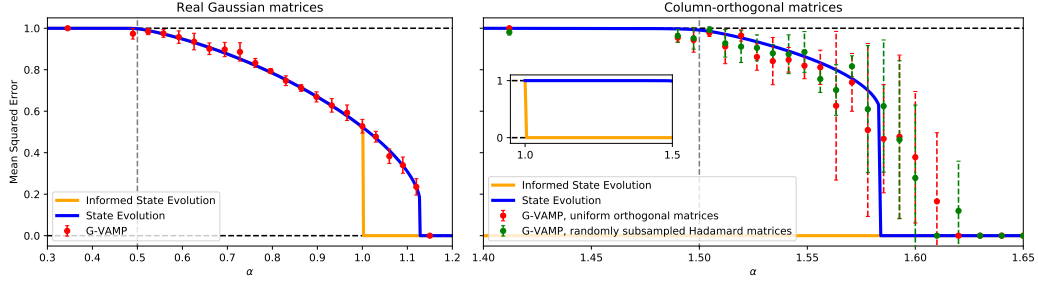

Figure 1: Comparison of MSE achieved by the Bayes-optimal estimator and the G-VAMP algorithm, for an i.i.d. real Gaussian (left) and a column-orthogonal (right) sensing matrix $\mathbf{\Phi}$ (i.e. $\mathbf{\Phi}^\intercal\mathbf{\Phi}/n = \mathbb{1}_n$), with a real Gaussian prior. Dots correspond to finite size simulations of G-VAMP (the mean and std are taken over 5 instances, with $n = 8000$ in the Gaussian case and $m = 8192$ in the orthogonal case), while full lines are obtained from the state evolution equations. The vertical grey dashed lines denote the algorithmic weak recovery threshold $\alpha_{\mathrm{WR,Algo}}$. Note the presence of a statistical-to-algorithmic gap in both ensembles, and that for column-orthogonal matrices $\alpha_{\mathrm{WR,Algo}} > \alpha_{\mathrm{FR,IT}}$.

**Gaussian sensing matrix —** For Gaussian i.i.d. matrices, $\langle\lambda\rangle_\nu = \alpha$ and $\langle\lambda^2\rangle_\nu = \alpha^2 + \alpha$, so that

$$\alpha_{\mathrm{WR,Algo}} = \Big[ \int_\mathbb{R} \mathrm{d}y \frac{|\int_\mathbb{K} \mathcal{D}_\beta z \,(|z|^2 - 1)\, P_\mathrm{out}(y|\sqrt{\rho}z)|^2}{\int_\mathbb{K} \mathcal{D}_\beta z\, P_\mathrm{out}(y|\sqrt{\rho}z)} \Big]^{-1}, \tag{12}$$

a result which was previously derived in [15] in the real and complex cases.

**Noiseless phase retrieval —** In the noiseless phase retrieval problem, one has $P_\mathrm{out}(y|z) = \delta(y - |z|^2)$. In particular, one can easily check that this implies:

$$\alpha_{\mathrm{WR,Algo}} = \Big(1 + \frac{\beta}{2}\Big) \frac{\langle\lambda\rangle_\nu^2}{\langle\lambda^2\rangle_\nu}. \tag{13}$$

This last formula allows to retrieve and generalize many results previously derived in the literature. For instance, for a Gaussian i.i.d. matrix, we find $\alpha_{\mathrm{WR,Algo}} = \beta/2$, which was derived in [15, 16]. For an orthogonal or unitary column matrix, $\alpha_{\mathrm{WR,Algo}} = 1 + (\beta/2)$, which was already known for $\beta = 2$ [15] (but not for $\beta = 1$). For the product of $p$ i.i.d. Gaussian matrices with sizes $k_0, \cdots, k_p$, with $k_0 = m$ and $k_p = n$, and $\gamma_l \equiv n/k_l$ for $0 \le l < p$, we have $\alpha_{\mathrm{WR,Algo}} = (\beta/2)[1 + \sum_{l=1}^p \gamma_l]^{-1}$, which generalizes the previously-known real case [14]. We emphasize that eq. (13) encapsulates all these results and goes beyond by considering an arbitrary spectrum for the sensing matrix, while eq. (11) also considers arbitrary channels $P_\mathrm{out}$.

**The weak-recovery IT transition —** So far, we only considered the *algorithmic* weak-recovery threshold. Extending our analysis to the *information-theoretic* treshold $\alpha_{\mathrm{WR,IT}}$ is an interesting open direction, which requires understanding the appearance of a global maximum in the replica-symmetric potential of eq. (6), but not necessarily continuously from the $q_x = q_z = 0$ solution. At the moment, we are not able to carry such an analysis, which is left for future work.

## 4 Statistical and algorithmic analysis of noiseless phase retrieval

While our results hold for any generalized estimation problem of the type introduced in Section 2 we now focus especially on noiseless phase retrieval. We fix $P_\mathrm{out}(y|z) = \delta(y - |z|^2)$ and take $P_0 = \mathcal{N}_\beta(0, 1)$. We can indeed consider $\rho = 1$, as the scaling is irrelevant under a noiseless channel.

**Full-recovery threshold for Gaussian signals —** We now turn our attention to the information-theoretical *full-recovery* threshold $\alpha_{\mathrm{FR,IT}}$. For high number of samples $\alpha \gg 1$, we expect the MMSE to plateau at a minimum achievable reconstruction error $\mathrm{MMSE}_0 \equiv \inf_\alpha \mathrm{MMSE}(\alpha)$, which is a function of the statistics of $\mathbf{\Phi}$. In this case, we define the information-theoretical full-recovery threshold $\alpha_{\mathrm{FR,IT}}$ as the smallest sample complexity such that $\mathrm{MMSE}_0$ is attained. In Appendix C

we show that the full-recovery can be *perfect* ($\mathrm{MMSE}_0 = 0$) or *partial* ($\mathrm{MMSE}_0 > 0$) depending on the rank of $\mathbf{\Phi}$. Indeed, we show that:

$$\alpha_{\mathrm{FR,IT}} \equiv \beta(1 - \nu(\{0\})). \tag{14}$$

Informally, $\nu(\{0\})$, the fraction of zeros in the spectrum of $\mathbf{\Phi}^\dagger\mathbf{\Phi}/n$, is the fraction of the signal "lost" by the sensing matrix. The stationary point of eq. (6) that corresponds to full recovery satisfies $\mathrm{MMSE}_0 = \nu(\{0\})$, while the reconstruction of the vector $\mathbf{\Phi}\mathbf{x}$ is perfect. The effect of rank deficiency is illustrated in Fig. 3-left, with the case of $\mathbf{\Phi}$ given by a product of two Gaussian matrices. We emphasize that $\alpha_{\mathrm{FR,IT}}$ is in general not well-defined for an arbitrary channel, which is why we only derived eq. (14) in the noiseless case.

**Evaluation of the thresholds and comparison to simulations —** Algorithmic weak-recovery and information-theoretical full-recovery thresholds can be readily obtained from eqs. (13),(14). Below, we solve the state evolution equations (9) for different real and complex ensembles of sensing matrix $\mathbf{\Phi}$, and compare it to numerical simulations of G-VAMP.

**Real case —** The case of a real signal $\mathbf{X}^\star \in \mathbb{R}^n$ has been previously studied in the literature for particular ensembles of real-valued sensing matrix $\mathbf{\Phi}$. A formula analogous to eq. (6) has been heuristically derived for real orthogonally invariant matrices $\mathbf{\Phi}$ and real signals drawn from generic but separable $P_0$ [12], and the specific i.i.d. Gaussian matrix case was rigorously proven in [13]. The heuristic analysis was later extended to non-separable signal distributions $P_0$ [14]. In Fig. 1, we illustrate the case of real Gaussian and real column-orthogonal sensing matrix $\mathbf{\Phi}$, the latter not having been investigated previously in the literature. We compute the MMSE by solving the State Evolution equations starting from an *informed* solution (close to full recovery). The minimal mean-squared error achievable with the G-VAMP algorithm is computed using the State Evolution equations starting from the *uninformed* $q_z = 0$ solution. We compare these predictions with numerical simulations of the G-VAMP algorithm on Gaussian matrices and uniformly sampled orthogonal matrices, as well as randomly subsampled Hadamard matrices. The simulations are in very good agreement with the prediction, and our results on Hadamard matrices suggest that the curves of Fig. 1-right are valid for more general ensembles than uniformly sampled orthogonal matrices, and that one can allow some controlled structure in the matrix without harming the performance of the algorithm.

**Complex case —** Previous works on complex signals $\mathbf{X}^\star \in \mathbb{C}^n$ have (to the best of our knowledge) focused solely on the study of the weak recovery threshold $\alpha_{\mathrm{WR}}$ (statistical or algorithmic), which was located for i.i.d. complex Gaussian matrices [15, 16] and uniformly sampled column-unitary matrices [17, 19]. We begin by extending the aforementioned results by identifying the full recovery threshold $\alpha_{\mathrm{FR,IT}}$ in these cases, and comparing the performance of the G-VAMP algorithm to the SE solution. Fig. 2 illustrates our results for these two ensembles. The algorithmic full-recovery threshold $\alpha_{\mathrm{FR,Algo}}$ is found numerically from the state evolution equations and is in good agreement with finite size simulations. The existence of a statistical-to-algorithmic gap $\Delta = \alpha_{\mathrm{FR,Algo}} - \alpha_{\mathrm{FR,IT}} \geq 0$ reflects the intrinsic hardness of phase retrieval in the real and complex case. However, it is interesting to note that even though full-recovery in the complex case requires more data than in the real case, the size of the statistical-to-algorithmic gap in the complex ensembles is smaller than in their real counterparts. In Fig. 3 we analyze the case of a product of two i.i.d. standard Gaussian matrices $\mathbf{\Phi} = \mathbf{W}_1\mathbf{W}_2$, with $\mathbf{W}_1 \in \mathbb{C}^{m \times p}$ and $\mathbf{W}_2 \in \mathbb{C}^{p \times n}$ for different aspect ratios $\gamma \equiv p/n$. We can identify the presence of a threshold $\alpha_{\mathrm{WR,Algo}} = \gamma/(1+\gamma)$ (computed in Section 3) that delimits the possibility of weak recovery both information-theoretically and in polynomial time. The information-theoretic full-recovery is achieved at $\alpha_{\mathrm{FR,IT}} = \min(2, 2\gamma)$, in agreement with eq. (14). Consistently with the real case results of [14], the full recovery algorithmic threshold is very close to the information-theoretic one, and precisely equal for $\gamma = 1$, although the gap is too small to be visible in the left and right parts of Fig. 3. Therefore, the performance of G-VAMP is exactly given by the Bayes-optimal estimator, apart for $\gamma \neq 1$ in a very small range $(\alpha_{\mathrm{FR,IT}}, \alpha_{\mathrm{FR,Algo}})$, whose size is of order $10^{-3}$ for $\gamma \in \{0.5, 1.5\}$. As $\gamma \to \infty$, one recovers the statistical-to-algorithmic gap present in the complex Gaussian case, which is again very small (around 0.027, cf Table 1). Although this hard phase is very small, we therefore postulate its existence for all $\gamma \neq 1$, generalizing the real case results of [14].

**Application to images —** Importantly, while the knowledge of the distribution of the true signal is required for our theoretical analysis, the G-VAMP algorithm is also well-defined beyond this scope, e.g. it can be used to infer natural images with Fourier matrices. Using a Gaussian prior to infer the

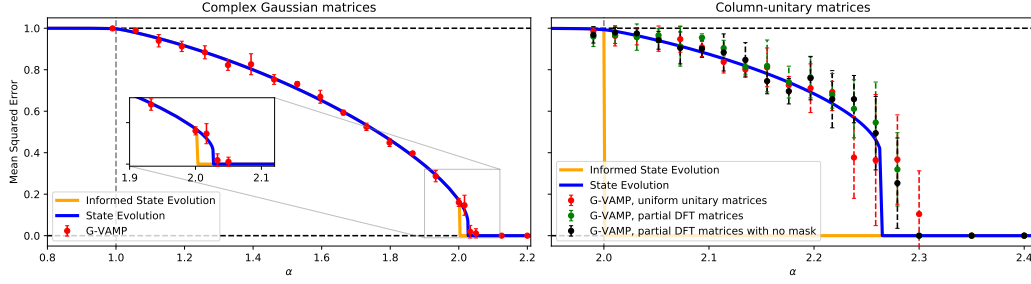

Figure 2: Comparison of MSE achieved by the Bayes-optimal estimator and G-VAMP algorithm for phase retrieval, for the case of an i.i.d. complex Gaussian (left) and a column-unitary (right) sensing matrix $\mathbf{\Phi}$ (i.e. $\mathbf{\Phi}^\dagger \mathbf{\Phi}/n = \mathbb{1}_n$), with a complex Gaussian prior. Dots correspond to finite size simulations of G-VAMP (with $n = 5000$, the mean and std are taken over $5$ independent instances), while full lines are obtained from the state evolution equations. Note the presence of a statistical-to-algorithmic gap in both ensembles.

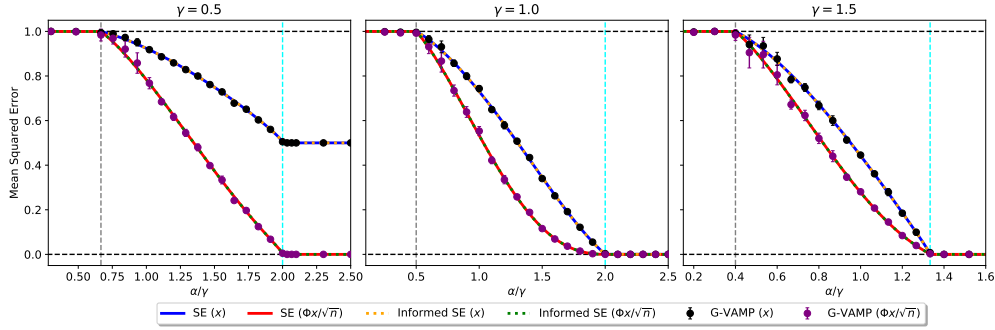

Figure 3: Mean squared error as a function of the measurement rate $\alpha$, for a sensing matrix $\mathbf{\Phi} = \mathbf{W}_1 \mathbf{W}_2$ a product of two complex i.i.d. standard Gaussian matrices $\mathbf{W}_1 \in \mathbb{C}^{m \times p}$, $\mathbf{W}_2 \in \mathbb{C}^{p \times n}$ with aspect ratios $\gamma = p/n \in \{0.5, 1.0, 1.5\}$. Red curves denote the recovery on $\mathbf{\Phi}\mathbf{X}^\star/\sqrt{n}$ and blue curves on $\mathbf{X}^\star$. Cyan dashed lines denote the full reconstruction threshold $\alpha_{\mathrm{FR,IT}}$. The G-VAMP experiments were performed with $n = 5000$, and the mean and std are taken over $5$ instances.

image can then actually be seen as the minimal assumption on the underlying signal, as it amounts to simply fix its norm: our theory can thus predict the performance of this G-VAMP algorithm for any signal, structured or not. We conducted a simple experiment on a natural image with a randomly subsampled DFT matrix $\mathbf{\Phi}$, described in Fig. 4. Although we are far from a Bayes-optimal setting, the achieved MSE is very close to values of Fig. 2 of the paper, for all values of $\alpha$. In particular, we achieve perfect recovery for $\alpha \geq 2.3$, just above $\alpha_{\mathrm{FR,Algo}} \simeq 2.27$ which was derived for random unitary matrices, i.i.d. data and in the Bayes-optimal setting.

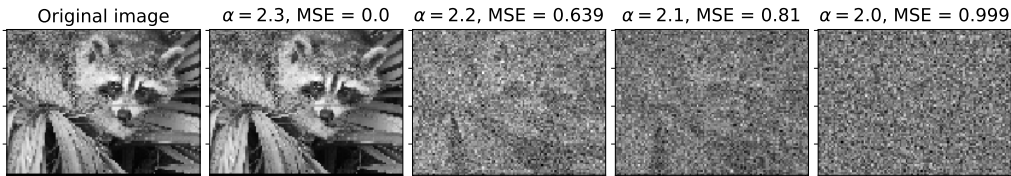

Figure 4: Performance of the G-VAMP algorithm for noiseless phase retrieval. We wish to recover a 77x102 image (on the left), and we use a complex Gaussian prior to infer the signal. The data matrix $\mathbf{\Phi}$ is a randomly subsampled DFT matrix.

## Acknowledgments and Disclosure of Funding

The authors would like to thank Yoshiyuki Kabashima for insightful discussions on the replica computations with orthogonally invariant matrices, and Yue M. Lu for fruitful discussions at the beginning of this work. Additional funding is acknowledged by AM from "Chaire de recherche sur les modèles et sciences des données", Fondation CFM pour la Recherche-ENS. This work is supported by the ERC under the European Union's Horizon 2020 Research and Innovation Program 714608-SMiLe, as well as by the French Agence Nationale de la Recherche under grant ANR-17-CE23-0023-01 PAIL and ANR-19-P3IA-0001 PRAIRIE.

## Broader Impact

Our work is theoretical in nature, and as such its potential societal consequences are difficult to foresee. We however anticipate that deeper theoretical understanding of the functioning of machine learning systems will lead to better anticipation of such societal consequences in the long term.

## Footnotes

[1] We actually assume the following, which is (slightly) stronger: the convergence should happen at a rate at least $n^{1+\epsilon}$ for an $\epsilon > 0$. This condition was not precised in the replica calculation of [12] for real matrices. In practice, in classical orthogonally (unitarily)-invariant random matrix ensembles, we often have $\epsilon = 1$.

[1] The rigorous statement on the limit of the MMSE requires adding a side information channel with arbitrarily small signal, cf Appendix D.5.

[2] In Theorem 2.2, we rely on some Gaussianity, either in the prior or in the data matrix. This is a not specific to our setting, but rather a fundamental limitation of the adaptive interpolation method used for the proof.

[1]Even though we do not provide a proof for the optimality of-GVAMP, we chose such notation in accordance with the previous literature on this topic, in which this optimality is often assumed.

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
