[Supplementary Material]

# Phase retrieval in high dimensions:
## Statistical and computational phase transitions

# SUPPLEMENTARY MATERIAL

Many notations and definitions used throughout this supplementary material are given in Sections F.1,F.2. The Python code that produced the numerical data used in Figures 1,2,3, as well as the data itself, are given alongside this material, and is dependent on the open-source TrAMP library [1]. We provide in particular an "example" notebook which contains a detailed presentation of the functions necessary to generate both the state evolution and the G-VAMP data for the complex Gaussian matrix case.

## A The replica computation of the free entropy

In this section, which has a more pedagogical purpose, we perform the replica calculation that gives Conjecture 2.1. This calculation for real matrices was already performed in [2], and as we will see it generalizes to complex valued signal and matrices. Note that we restricted ourselves to a Bayes-optimal inference problem, while the setting of [2] includes possibly mismatched models[1].

### A.1 Setting

We let $n, m \to \infty$ with $m/n \to \alpha > 0$. We assume that we have access to a prior distribution $P_0$ on $\mathbb{K}$ and a channel distribution $P_{\text{out}}(y|z)$, of "observations" $y \in \mathbb{R}$ conditioned by a latent variable $z \in \mathbb{K}$. We are given data $\mathbf{Y} \in \mathbb{R}^m$ generated as:

$$Y_\mu \sim P_{\text{out}}\Big(\cdot \Big| \frac{1}{\sqrt{n}} \sum_{i=1}^n \Phi_{\mu i} X_i^\star \Big),$$

in which $X_i^\star \overset{\text{i.i.d.}}{\sim} P_0$ (with $\mathbb{E}|X^\star|^2 = \rho > 0$), and $\mathbf{\Phi} \in \mathbb{K}^{m \times n}$ is a matrix that is both left and right orthogonally (respectively unitarily) invariant, meaning that for all $\mathbf{O}, \mathbf{U} \in \mathcal{U}_\beta(m) \times \mathcal{U}_\beta(n)$, $\mathbf{\Phi} \overset{d}{=} \mathbf{O}\mathbf{\Phi}\mathbf{U}$. Compared to Conjecture 2.1, we added a left-invariance hypothesis. However the analysis of G-VAMP [3, 4] shows that this left invariance is actually not needed for the result, and thus we state Conjecture 2.1 for matrices that are only right-invariant, but we use the left invariance to simplify the following (heuristic) calculation. Moreover, we assume that the asymptotic eigenvalue distribution of $\mathbf{\Phi}^\dagger \mathbf{\Phi}/n$ is well-defined and we denote it $\nu$, and that the eigenvalue distribution of $\mathbf{\Phi}^\dagger \mathbf{\Phi}/n$ has large deviations in a scale at least $n^{1+\eta}$ for an $\eta > 0$. The partition function is:

$$\mathcal{Z}_n(\mathbf{Y}) \equiv \int_{\mathbb{K}^n} \prod_{i=1}^n P_0(\mathrm{d}x_i) \prod_{\mu=1}^m P_{\text{out}}\Big(Y_\mu \Big| \frac{1}{\sqrt{n}} \sum_{i=1}^n \Phi_{\mu i} x_i \Big).$$

The replica trick [5] consists in computing the $p$-th moment of the partition function for arbitrary integer $p$, before extending this expression analytically to any $p > 0$ and using the formula:

$$\lim_{n\to\infty} \frac{1}{n} \mathbb{E}_{\mathbf{\Phi},\mathbf{Y}} \ln \mathcal{Z}_n(\mathbf{Y}) = \lim_{p\downarrow 0} \lim_{n\to\infty} \frac{1}{np} \ln \mathbb{E}_{\mathbf{\Phi},\mathbf{Y}}[\mathcal{Z}_n(\mathbf{Y})^p].$$

This method is obviously non-rigorous given the inversion of limits $p \downarrow 0$ and $n \to \infty$, as well as the analytic continuation to arbitrary $p > 0$ of the $p$-th moment. However, it has achieved tremendous success in the study of spin glasses and inference problems, see e.g. [6].

### A.2 Computing the $p$-th moment of the partition function

Thanks to Bayes-optimality, we can easily write the average of $\mathcal{Z}_n(\mathbf{Y})^p$ as an average over $p + 1$ replicas of the system, by considering $\mathbf{X}^\star$ as the replica of index 0. We obtain for any $p \geq 1$:

$$\mathbb{E}[\mathcal{Z}_n(\mathbf{Y})^p] = \mathbb{E}_{\mathbf{\Phi}} \int_{\mathbb{R}^m} \mathrm{d}\mathbf{Y} \prod_{a=0}^p \Big\{ \Big[ \int_{\mathbb{K}} \prod_{i=1}^n P_0(\mathrm{d}x_i^a) \int_{\mathbb{K}} \prod_{\mu=1}^m \mathrm{d}z_\mu^a P_{\text{out}}(Y_\mu|z_\mu^a) \Big] \delta\Big( \mathbf{z}^a - \frac{\mathbf{\Phi}\mathbf{x}^a}{\sqrt{n}} \Big) \Big\}. \quad (1)$$

The first step is to decompose eq. (1) into three terms, corresponding to the prior $P_0$, the channel $P_{\text{out}}$, and the "delta" term. Note that the matrix $\boldsymbol{\Phi}$ only appears in the last "delta" term. By left and right orthogonal (resp. unitary) invariance of $\boldsymbol{\Phi}$, the quantity

$$\mathbb{E}_{\boldsymbol{\Phi}}\Big[ \prod_{a=0}^{p} \delta\Big( \mathbf{z}^a - \frac{1}{\sqrt{n}}\boldsymbol{\Phi}\mathbf{x}^a \Big) \Big]$$

is determined by the value of the *overlaps* $\mathbf{Q}^z \equiv \{(\mathbf{z}^a)^\dagger \mathbf{z}^b / m\}_{a,b=0}^{p}$ and $\mathbf{Q}^x \equiv \{(\mathbf{x}^a)^\dagger \mathbf{x}^b / n\}_{a,b=0}^{p}$, which are positive symmetric (Hermitian in the complex case) matrices. As is standard in such replica calculations, we will constraint the terms in eq. (1) by the value of these overlaps, before performing a Laplace method on the resulting function of the overlaps. By $A_n \simeq B_n$, we will mean equivalence at leading exponential order, that is $(\ln A_n)/n = (\ln B_n)/n + \mathcal{o}_n(1)$. We introduce in eq. (1) the term:

$$1 \simeq \int \prod_{0 \le a \le b \le p} \mathrm{d}Q_{ab}^x \, \mathrm{d}Q_{ab}^z \Big[ \prod_{a \le b} \delta(nQ_{ab}^x - (\mathbf{x}^a)^\dagger \mathbf{x}^b) \Big] \Big[ \prod_{a \le b} \delta(mQ_{ab}^z - (\mathbf{z}^a)^\dagger \mathbf{z}^b) \Big].$$

We can use a Fourier transformation of the delta terms, which allows in the end to transform eq.(1) into the product of three independent terms. Performing the saddle-point on $\mathbf{Q}^x, \mathbf{Q}^z$, we obtain the corresponding result:

$$\lim_{n \to \infty} \frac{1}{n} \ln \mathbb{E}_{\mathbf{Y},\boldsymbol{\Phi}}[\mathcal{Z}_n(\mathbf{Y})^p] = \sup_{\mathbf{Q}^x, \mathbf{Q}^z} [I_0(p,\mathbf{Q}^x) + \alpha I_{\text{out}}(p,\mathbf{Q}^z) + I_{\text{int}}(p,\mathbf{Q}^x,\mathbf{Q}^z)],$$

in which the supremum is made over positive symmetric (Hermitian) matrices, and $I_0$, $I_{\text{out}}$ and $I_{\text{int}}$ are functions whose calculation will be detailed below.

### A.2.1 The prior term $I_0(p,\mathbf{Q}^x)$

We have by the Laplace method after Fourier transformation of the delta terms:

$$I_0(p,\mathbf{Q}^x) \simeq \frac{1}{n} \ln \int \prod_{0 \le a \le b \le p} \mathrm{d}\hat{Q}_{ab}^x \int_{\mathbb{K}} \prod_{a=0}^{p} \prod_{i=1}^{n} P_0(\mathrm{d}x_i^a) e^{-\frac{\beta}{2} \sum_{a,b=0}^{p} \hat{Q}_{ab}^x (\sum_i \overline{x_i^a} x_i^b - nQ_{ab}^x)},$$

$$\simeq \inf_{\hat{\mathbf{Q}}^x} \Big[ \frac{\beta}{2} \sum_{a,b} Q_{ab}^x \hat{Q}_{ab}^x + \ln \int_{\mathbb{K}} \prod_{a=0}^{p} P_0(\mathrm{d}x^a) e^{-\frac{\beta}{2} \sum_{a,b} \hat{Q}_{ab}^x \overline{x^a} x^b} \Big].$$

The infimum is again over positive symmetric (Hermitian) matrices. We also made use of the fact that the prior $P_0$ is i.i.d. over the elements of $\mathbf{x}$. A very important assumption of our calculation is replica symmetry. It amounts to assume that all the $(p+1)$ replicas are equivalent, and that this symmetry is not broken by the system at the solution of the Laplace method. Replica symmetry and replica symmetry breaking are a very rich field of study in statistical physics [5]. It has been argued that for an inference problem in the Bayes-optimal setting (as is the present case), replica symmetry is never broken [6]. We can therefore assume a replica symmetric form of $\mathbf{Q}^x, \hat{\mathbf{Q}}^x$ at the point at which the saddle point is reached, that we write as:

$$\mathbf{Q}^x = \begin{pmatrix} Q_x & q_x & \cdots & q_x \\ q_x & Q_x & \cdots & q_x \\ \vdots & \vdots & \ddots & \vdots \\ q_x & q_x & \cdots & Q_x \end{pmatrix}, \qquad \hat{\mathbf{Q}}^x = \begin{pmatrix} \hat{Q}_x & -\hat{q}_x & \cdots & -\hat{q}_x \\ -\hat{q}_x & \hat{Q}_x & \cdots & -\hat{q}_x \\ \vdots & \vdots & \ddots & \vdots \\ -\hat{q}_x & -\hat{q}_x & \cdots & \hat{Q}_x \end{pmatrix}. \tag{2}$$

Note that for $\beta \in \{1,2\}$, we have $Q_x, q_x, \hat{Q}_x, \hat{q}_x \in \mathbb{R}$. After a simple Gaussian transformation of the squared term using the general identity for $x \in \mathbb{K}$:

$$\exp\Big( \frac{\beta}{2} |x|^2 \Big) = \int_{\mathbb{K}} \mathcal{D}_\beta \xi \, \exp(\beta x \cdot \xi),$$

we reach the final expression:

$$I_0(p,Q_x,q_x) = \tag{3}$$
$$\inf_{\hat{Q}_x,\hat{q}_x} \Big\{ \frac{\beta(p+1)}{2} Q_x \hat{Q}_x - \frac{\beta p(p+1)}{2} q_x \hat{q}_x + \ln \int_{\mathbb{K}} \mathcal{D}_\beta \xi \Big[ \int_{\mathbb{K}} P_0(\mathrm{d}x) e^{-\frac{\beta(\hat{Q}_x + \hat{q}_x)}{2} |x|^2 + \beta \sqrt{\hat{q}_x} x \cdot \xi} \Big]^{p+1} \Big\}.$$

 **A.2.2 The channel term** $I_{\text{out}}(p, \mathbf{Q}^z)$

 This term is very similar to the prior term detailed in the previous section. We use completely similar
 replica symmetric assumptions for the overlaps $\mathbf{Q}^z$ to the ones on $\mathbf{Q}^x$ described in eq. (2). We reach:

$$
I_{\text{out}}(p, Q_z, q_z) = \inf_{\hat{Q}_z, \hat{q}_z} \left\{ \frac{\beta(p+1)}{2} Q_z \hat{Q}_z - \frac{\beta p(p+1)}{2} q_z \hat{q}_z + \frac{\beta(p+1)}{2} \ln(2\pi/(\beta \hat{Q}_z)) \right. \tag{4}
$$
$$
\left. + \ln \int_{\mathbb{R}} \mathrm{d}y \int_{\mathbb{K}} \mathcal{D}_\beta \xi \Big[ \int_{\mathbb{K}} \mathrm{d}z \Big( \frac{2\pi}{\beta \hat{Q}_z} \Big)^{-\beta/2} P_{\text{out}}(y|z)\, e^{-\beta \frac{\hat{Q}_z + \hat{q}_z}{2} |z|^2 + \beta \sqrt{\hat{q}_z} z \cdot \xi} \Big]^{p+1} \right\}.
$$

 We normalized the integrals so that in the limit $p \to 0$, the term inside the logarithm goes to 1, which
 will be a useful remark.

 **A.2.3 The delta term** $I_{\text{int}}(p, \mathbf{Q}^x, \mathbf{Q}^z)$

 We now turn to the computation of the delta term:

$$
I_{\text{int}}(p, \mathbf{Q}^x, \mathbf{Q}^z) \equiv \lim_{n \to \infty} \frac{1}{n} \ln \mathbb{E}_{\mathbf{\Phi}} \Big[ \prod_{a=0}^{p} \delta\Big( \mathbf{z}^a - \frac{1}{\sqrt{n}} \mathbf{\Phi} \mathbf{x}^a \Big) \Big], \tag{5}
$$

 assuming that $\mathbf{Q}^x, \mathbf{Q}^z$ are known. Computing this term is central in this replica calculation. We use,
 as is done in [2], the identity:

$$
\frac{1}{n} \ln \mathbb{E}_{\mathbf{\Phi}} \Big[ \prod_{a=0}^{p} \delta\Big( \mathbf{z}^a - \frac{1}{\sqrt{n}} \mathbf{\Phi} \mathbf{x}^a \Big) \Big] = \lim_{\epsilon \downarrow 0} \frac{1}{n} \ln \mathbb{E}_{\mathbf{\Phi}} \Big[ \frac{\exp\big\{ -\frac{\beta}{2\epsilon} \sum_a \big\| \mathbf{z}^a - \frac{1}{\sqrt{n}} \mathbf{\Phi} \mathbf{x}^a \big\|^2 \big\}}{(2\pi\epsilon/\beta)^{\frac{\beta m(p+1)}{2}}} \Big], \tag{6}
$$

 and we invert the $n \to \infty$ and the $\epsilon \to 0$ limit. Let us rewrite the right-hand-side of eq. (6). Since $\mathbf{\Phi}$
 is orthogonally (resp. unitarily) invariant, we can write this term as:

$$
\mathbb{E} \Big[ \frac{\exp\big\{ -\frac{\beta}{2\epsilon} \sum_a \big\| \mathbf{z}^a - \frac{1}{\sqrt{n}} \mathbf{\Phi} \mathbf{x}^a \big\|^2 \big\}}{(2\pi\epsilon/\beta)^{\frac{\beta m(p+1)}{2}}} \Big] = \mathbb{E} \Big[ \frac{\exp\big\{ -\frac{\beta}{2\epsilon} \sum_a \big\| \mathbf{O} \mathbf{z}^a - \frac{1}{\sqrt{n}} \mathbf{\Phi} \mathbf{U} \mathbf{x}^a \big\|^2 \big\}}{(2\pi\epsilon/\beta)^{\frac{\beta m(p+1)}{2}}} \Big], \tag{7}
$$

 in which the average on the right hand side is made over $(\mathbf{\Phi}, \mathbf{O}, \mathbf{U})$, with $(\mathbf{O}, \mathbf{U})$ uniformly sampled
 over the orthogonal groups $\mathcal{U}_\beta(m), \mathcal{U}_\beta(n)$. Note that since the overlap matrices $\mathbf{Q}^z, \mathbf{Q}^x$ are fixed, one
 can show that when $\mathbf{U}$ is uniformly distributed over $\mathcal{U}_\beta(n)$, the set of vectors $\{ \mathbf{U} \mathbf{x}^a \}_{a=0}^{p}$ is uniformly
 distributed over the set of $(p+1)$ vectors in $\mathbb{K}^n$ with overlap matrix $\mathbf{Q}^x$. There is a completely
 similar result for $\mathbf{z}$ as well. The consequence is that we can replace in eq. (7) the average over $\mathbf{O}, \mathbf{U}$
 by an average over the vectors satisfying this constraint:

$$
I_{\text{int}}(p, \mathbf{Q}^x, \mathbf{Q}^z) \tag{8}
$$
$$
\simeq \frac{1}{n} \ln \mathbb{E}_{\mathbf{\Phi}} \frac{\int_{\mathbb{K}} \prod_a \mathrm{d}\mathbf{x}^a\, \mathrm{d}\mathbf{z}^a \Big[ \prod_{a \le b} \delta(n Q_{ab}^x - (\mathbf{x}^a)^\dagger \mathbf{x}^b) \delta(m Q_{ab}^z - (\mathbf{z}^a)^\dagger \mathbf{z}^b) \Big] \frac{e^{-\frac{\beta}{2\epsilon} \sum_a \| \mathbf{z}^a - \frac{1}{\sqrt{n}} \mathbf{\Phi} \mathbf{x}^a \|^2}}{(2\pi\epsilon/\beta)^{\beta m(p+1)/2}}}{\int_{\mathbb{K}} \prod_a \mathrm{d}\mathbf{x}^a\, \mathrm{d}\mathbf{z}^a \Big[ \prod_{a \le b} \delta(n Q_{ab}^x - (\mathbf{x}^a)^\dagger \mathbf{x}^b) \delta(m Q_{ab}^z - (\mathbf{z}^a)^\dagger \mathbf{z}^b) \Big]}.
$$

 The numerator and the denominator correspond to two terms, that we denote $I_{\text{int}}(p, \mathbf{Q}^x, \mathbf{Q}^z) =$
 $I_c^{(n)}(p, \mathbf{Q}^x, \mathbf{Q}^z) - I_c^{(d)}(p, \mathbf{Q}^x, \mathbf{Q}^z)$. We can introduce the Fourier-transform of the delta distribution
 to compute both terms, as in the previous sections. Let us start with the denominator. It reduces after
 Fourier-transformation to a Gaussian integral involving a block-diagonal matrix:

$$
I_{\text{int}}^{(d)}(p, \mathbf{Q}^x, \mathbf{Q}^z) \simeq \frac{\beta}{2} \inf_{\mathbf{\Gamma}^x, \mathbf{\Gamma}^z} \Big[ \text{Tr}[\mathbf{Q}^x \mathbf{\Gamma}^x] + \alpha \text{Tr}[\mathbf{Q}^z \mathbf{\Gamma}^z] + (\alpha+1)(p+1) \ln \frac{2\pi}{\beta}
$$
$$
- \ln \det \mathbf{\Gamma}^x - \alpha \ln \det \mathbf{\Gamma}^z \Big],
$$

 with symmetric (Hermitian) positive matrices $\mathbf{\Gamma}^x, \mathbf{\Gamma}^z$ of size $(p+1)$. The infimum is readily solved
 by $\mathbf{\Gamma}^x = (\mathbf{Q}^x)^{-1}$ and $\mathbf{\Gamma}^z = (\mathbf{Q}^z)^{-1}$, which yields:

$$
I_{\text{int}}^{(d)}(p, \mathbf{Q}^x, \mathbf{Q}^z) \simeq \frac{\beta(\alpha+1)(p+1)}{2} (1 + \ln \frac{2\pi}{\beta}) + \frac{\beta}{2} \ln \det \mathbf{Q}^x + \frac{\alpha\beta}{2} \ln \det \mathbf{Q}^z. \tag{9}
$$

Let us now compute the numerator with the same technique. We obtain:

$$I_{\text{int}}^{(n)}(p, \mathbf{Q}^x, \mathbf{Q}^z) \simeq \frac{\beta(p+1)}{2} \ln \frac{2\pi}{\beta\epsilon^\alpha} + \frac{\beta}{2} \inf_{\mathbf{\Gamma}^x, \mathbf{\Gamma}^z} \left[ \text{Tr}[\mathbf{Q}^x \mathbf{\Gamma}^x] + \alpha \text{Tr}[\mathbf{Q}^z \mathbf{\Gamma}^z] - \frac{1}{n} \ln \det \mathbf{M}_n \right], \quad (10)$$

with a Hermitian matrix $\mathbf{M}_n$ having a block structure, that we write here in the tensor product form:

$$\mathbf{M}_n \equiv \begin{pmatrix} (\mathbf{\Gamma}^z + \frac{1}{\epsilon} \mathbb{1}_{p+1}) \otimes \mathbb{1}_m & \frac{1}{\epsilon} \mathbb{1}_{p+1} \otimes \frac{\mathbf{\Phi}}{\sqrt{n}} \\ \frac{1}{\epsilon} \mathbb{1}_{p+1} \otimes \frac{\mathbf{\Phi}^\dagger}{\sqrt{n}} & \mathbf{\Gamma}^x \otimes \mathbb{1}_n + \frac{1}{\epsilon} \mathbb{1}_{p+1} \otimes \frac{\mathbf{\Phi}^\dagger \mathbf{\Phi}}{n} \end{pmatrix}. \quad (11)$$

Using the block-matrix determinant calculation:

$$\det \begin{pmatrix} A & B \\ C & D \end{pmatrix} = \det A \times \det(D - CA^{-1}B),$$

we reach:

$$\frac{1}{n} \ln \det \mathbf{M}_n = \alpha \ln \det \left( \mathbf{\Gamma}^z + \frac{1}{\epsilon} \mathbb{1}_{p+1} \right)$$
$$+ \frac{1}{n} \ln \det \left( \mathbf{\Gamma}^x \otimes \mathbb{1}_n + \frac{1}{\epsilon} \mathbb{1}_{p+1} \otimes \frac{\mathbf{\Phi}^\dagger \mathbf{\Phi}}{n} - \frac{1}{\epsilon^2} \left( \mathbf{\Gamma}^z + \frac{1}{\epsilon} \mathbb{1}_{p+1} \right)^{-1} \otimes \frac{\mathbf{\Phi}^\dagger \mathbf{\Phi}}{n} \right),$$
$$= (\alpha - 1) \ln \det \left( \mathbf{\Gamma}^z + \frac{1}{\epsilon} \mathbb{1}_{p+1} \right) + \frac{1}{n} \ln \det \left( \mathbf{\Gamma}^x \mathbf{\Gamma}^z \otimes \mathbb{1}_n + \frac{1}{\epsilon} \mathbf{\Gamma}^x \otimes \mathbb{1}_n + \frac{1}{\epsilon} \mathbf{\Gamma}^z \otimes \frac{\mathbf{\Phi}^\dagger \mathbf{\Phi}}{n} \right),$$
$$= (\alpha - 1) \ln \det \left( \mathbf{\Gamma}^z + \frac{1}{\epsilon} \mathbb{1}_{p+1} \right) + \left\langle \ln \det \left( \mathbf{\Gamma}^x \mathbf{\Gamma}^z + \frac{1}{\epsilon} (\mathbf{\Gamma}^x + \lambda \mathbf{\Gamma}^z) \right) \right\rangle_\nu,$$

with $\lambda$ distributed according to $\nu$, the asymptotic eigenvalue distribution of $\mathbf{\Phi}^\dagger \mathbf{\Phi}/n$. This allows to write $I_{\text{int}}^{(n)}$ from eq. (10) and to take the $\epsilon \downarrow 0$ limit, keeping the terms that do not vanish:

$$I_{\text{int}}^{(n)}(p, \mathbf{Q}^x, \mathbf{Q}^z) \simeq \frac{\beta}{2} \inf_{\mathbf{\Gamma}^x, \mathbf{\Gamma}^z} [\text{Tr}[\mathbf{Q}^x \mathbf{\Gamma}^x] + \alpha \text{Tr}[\mathbf{Q}^z \mathbf{\Gamma}^z] - \langle \ln \det(\mathbf{\Gamma}^x + \lambda \mathbf{\Gamma}^z) \rangle_\nu]. \quad (12)$$

Finally, we again consider a replica-symmetric assumption for $\mathbf{\Gamma}^x, \mathbf{\Gamma}^z$, in the form:

$$\mathbf{\Gamma}^x = \begin{pmatrix} \Gamma_x & -\gamma_x & \cdots & -\gamma_x \\ -\gamma_x & \Gamma_x & \cdots & -\gamma_x \\ \vdots & \vdots & \ddots & \vdots \\ -\gamma_x & -\gamma_x & \cdots & \Gamma_x \end{pmatrix}, \qquad \mathbf{\Gamma}^z = \begin{pmatrix} \Gamma_z & -\gamma_z & \cdots & -\gamma_z \\ -\gamma_z & \Gamma_z & \cdots & -\gamma_z \\ \vdots & \vdots & \ddots & \vdots \\ -\gamma_z & -\gamma_z & \cdots & \Gamma_z \end{pmatrix}. \quad (13)$$

As for the overlap matrices, we have $\gamma_x, \gamma_z \in \mathbb{R}$. Combining eqs. (9) and (12) and using the replica symmetric assumption, we obtain:

$$\frac{2}{\beta} I_{\text{int}}(p, \mathbf{Q}_x, \mathbf{Q}_z) = \inf_{\Gamma_x, \gamma_x, \Gamma_z, \gamma_z} [(p+1)Q_x \Gamma_x - p(p+1)q_x \gamma_x + \alpha(p+1)Q_z \Gamma_z - \alpha p(p+1)q_z \gamma_z$$
$$- p \langle \ln(\Gamma_x + \gamma_x + \lambda \Gamma_z + \lambda \gamma_z) \rangle_\nu - \langle \ln[\Gamma_x - p\gamma_x + \lambda(\Gamma_z - p\gamma_z)] \rangle_\nu] - (\alpha + 1)(p+1) \ln 2\pi e / \beta$$
$$+ (p+1) \ln \frac{2\pi}{\beta} - p \ln(Q_x - q_x) - \ln(Q_x + pq_x) - \alpha p \ln(Q_z - q_z) - \alpha \ln(Q_z + pq_z). \quad (14)$$

**A note on quenched and annealed averages**    Note that here we did not consider the average over $\mathbf{\Phi}$ to compute $I_{\text{int}}$. Indeed, the result only depends on the eigenvalue distribution of $\mathbf{\Phi}^\dagger \mathbf{\Phi}/n$, which (by hypothesis) has large deviations in a scale at least $n^{1+\eta}$ with $\eta > 0$. Since we are looking at a scale exponential in $n$, we can thus consider that this eigenvalue distribution is equal to its limit value $\nu$. However, one must be careful that this argument breaks down if our result starts to be sensitive to the extremal eigenvalues of $\mathbf{\Phi}^\dagger \mathbf{\Phi}/n$. Since these variables typically have large deviations in the scale $n$ (for instance for Wigner or Wishart matrices [7]), this could invalidate our calculation. This phenomenon is well-known in the study of so-called "HCIZ" spherical integrals, cf [8] for an example of a rigorous analysis. We argue in Section A.4 that this possible issue, not discussed in [2], never arises for physical values of the overlaps.

#### 104 A.2.4 Expressing the $p$-th moment

105 Combining the results of the three previous sections, we finally obtain the asymptotics of the $p$-th
106 moment of the partition function as:

$$\lim_{n\to\infty} \frac{1}{n} \ln \mathbb{E}\mathcal{Z}_n(\mathbf{Y})^p = \sup_{\substack{Q_x,q_x \\ Q_z,q_z}} \left[ I_0(p,Q_x,q_x) + \alpha I_{\text{out}}(p,Q_z,q_z) + I_{\text{int}}(p,Q_x,q_x,Q_z,q_z) \right], \quad (15)$$

107 in which the three terms are given by eqs. (3),(4),(14).

### 108 A.3 The $p \downarrow 0$ limit

109 One can easily see that the function described in eq. (15) is analytic in $p$. The next step of the replica
110 method is to analytically extend this expression to arbitrary $p > 0$, before considering the limit $p \downarrow 0$.

#### 111 A.3.1 Consistency of the limit

112 One must be careful that, when extending our expression to arbitrarily small $p > 0$, we satisfy the
113 trivial condition $\lim_{p \downarrow 0} \ln \mathbb{E}Z^p = 0$. As we will see, this condition will yield constraints on the
114 diagonals of the overlap matrices. Taking the limit $p = 0$ in the three terms of eq. (15) yields:

$$I_0(0,Q_x,q_x) = \inf_{\hat{Q}_x} \left\{ \frac{\beta}{2} Q_x \hat{Q}_x + \ln \int_{\mathbb{K}} P_0(\mathrm{d}x) e^{-\frac{\beta \hat{Q}_x}{2}|x|^2} \right\}, \quad (16)$$

$$I_{\text{out}}(0,Q_z,q_z) = \inf_{\hat{Q}_z} \left\{ \frac{\beta}{2} Q_z \hat{Q}_z + \frac{\beta}{2} \ln \left( \frac{2\pi}{\beta \hat{Q}_z} \right) \right\}, \quad (17)$$

$$I_{\text{int}}(0,Q_x,q_x,Q_z,q_z) = \inf_{\Gamma_x,\Gamma_z} \left[ \frac{\beta}{2} Q_x \Gamma_x + \frac{\alpha\beta}{2} Q_z \Gamma_z - \frac{\beta}{2} \langle \ln[\Gamma_x + \lambda \Gamma_z] \rangle_\nu \right] \quad (18)$$
$$- \frac{\beta(\alpha+1)}{2}\left(1 + \ln\frac{2\pi}{\beta}\right) + \frac{\beta}{2}\ln\frac{2\pi}{\beta} - \frac{\beta}{2}\ln Q_x - \frac{\alpha\beta}{2}\ln Q_z.$$

One can easily solve the saddle point equations on $Q_z, \hat{Q}_z$, they give $\Gamma_z = 0$ and $\hat{Q}_z = 1/Q_z$. One
can then find all the remaining variables easily: $Q_x = \rho$, $\hat{Q}_x = 0$, $\Gamma_x = \rho^{-1}$, $Q_z = \rho\langle\lambda\rangle_\nu/\alpha$,
$\hat{Q}_z = 1/Q_z$, $\Gamma_z = 0$. Plugging these parameters yields (we drop the vacuous dependency on $q_x, q_z$):

$$\begin{cases} I_0(0, Q_x = \rho) = 0, & (19\text{a}) \\[2mm] I_{\text{out}}\left(0, Q_z = \frac{\rho\langle\lambda\rangle_\nu}{\alpha}\right) = \frac{\beta}{2} + \frac{\beta}{2}\ln\left(\frac{2\pi\rho\langle\lambda\rangle_\nu}{\beta\alpha}\right), & (19\text{b}) \\[2mm] I_{\text{int}}\left(0, Q_x = \rho, Q_z = \frac{\rho\langle\lambda\rangle_\nu}{\alpha}\right) = -\frac{\beta\alpha}{2}\left(1 + \ln\frac{2\pi}{\beta}\right) - \frac{\alpha\beta}{2}\ln\frac{\rho\langle\lambda\rangle_\nu}{\alpha}. & (19\text{c}) \end{cases}$$

115 Recall that we have

$$\lim_{p\downarrow 0}\lim_{n\to\infty} \frac{1}{n}\ln\mathbb{E}\mathcal{Z}_n(\mathbf{Y})^p = I_0 + \alpha I_{\text{out}} + I_{\text{int}},$$

116 so that we obtain from eq. (19) that indeed the limit is consistent.

#### 117 A.3.2 The replica symmetric result

118 Using eq. (15) for the $p$-th moment and the consistency conditions we just derived, we obtain after
119 using the replica trick:

$$\lim_{n\to\infty}\frac{1}{n}\mathbb{E}\ln\mathcal{Z}_n(\mathbf{Y}) = \sup_{q_x,q_z}\left[I_0(q_x) + \alpha I_{\text{out}}(q_z) + I_{\text{int}}(q_x,q_z)\right], \quad (20)$$

with the auxiliary functions:

$$I_0(q_x) = \inf_{\hat{q}_x \geq 0} \left[ -\frac{\beta \hat{q}_x q_x}{2} + \int_{\mathbb{K}} \mathcal{D}_\beta \xi P_0(\mathrm{d}x) e^{-\frac{\beta \hat{q}_x}{2}|x|^2 + \beta \sqrt{\hat{q}_x} x \cdot \xi} \ln \int_{\mathbb{K}} P_0(\mathrm{d}x) e^{-\frac{\beta \hat{q}_x}{2}|x|^2 + \beta \sqrt{\hat{q}_x} x \cdot \xi} \right],$$

$$I_{\mathrm{out}}(q_z) = \inf_{\hat{q}_z \geq 0} \left\{ -\frac{\beta \hat{q}_z q_z}{2} - \frac{\beta}{2} \ln(\hat{Q}_z + \hat{q}_z) + \frac{\beta \hat{q}_z}{2 \hat{Q}_z} + \int \mathrm{d}y \mathcal{D}_\beta \xi \, J(\hat{q}_z, y, \xi) \ln J(\hat{q}_z, y, \xi) \right\},$$

$$I_{\mathrm{int}}(q_x, q_z) = \inf_{\gamma_x, \gamma_z \geq 0} \left[ \frac{\beta}{2}(\rho - q_x)\gamma_x + \frac{\alpha\beta}{2}(Q_z - q_z)\gamma_z - \frac{\beta}{2} \langle \ln(\rho^{-1} + \gamma_x + \lambda \gamma_z) \rangle_\nu \right]$$
$$- \frac{\beta}{2} \ln(\rho - q_x) - \frac{\beta q_x}{2\rho} - \frac{\alpha\beta}{2} \ln(Q_z - q_z) - \frac{\alpha \beta q_z}{2 Q_z},$$

with $Q_z = \rho \langle \lambda \rangle_\nu / \alpha$ and $\hat{Q}_z = 1/Q_z$. Moreover, the domain of the supremum is $q_x \in [0, \rho]$ and $q_z \in [0, Q_z]$. The function $J(\hat{q}_z, y, \xi)$ appearing in the expression of $I_{\mathrm{out}}$ is defined as:

$$J(\hat{q}_z, y, \xi) \equiv \int_{\mathbb{K}} \mathcal{D}_\beta z P_{\mathrm{out}} \left( y \Big| \frac{z}{\sqrt{\hat{Q}_z + \hat{q}_z}} + \sqrt{\frac{\hat{q}_z}{\hat{Q}_z (\hat{Q}_z + \hat{q}_z)}} \xi \right).$$

Note that compared to the calculation presented in the previous sections, we moved a term $(\beta\alpha/2)(1 + \ln 2\pi/\beta)$ between $I_{\mathrm{out}}$ and $I_{\mathrm{int}}$, and we also made a few straightforward change of variables in the expression of $I_{\mathrm{out}}$. This is exactly the result given in Conjecture 2.1, which ends our replica calculation.

## A.4 Concentration of the spectrum of $\Phi^\dagger \Phi / n$ and the absence of saturation

As emphasized in the end of Section A.2.3, our calculation assumed that the extremization equations on $(\gamma_x, \gamma_z)$ always admitted a solution. Moreover, we assumed that this solution is not sensitive to the extremal eigenvalues of $\Phi^\dagger \Phi / n$. If this assumption is indeed true, the concentration of the spectrum of $\Phi^\dagger \Phi / n$ was assumed to be fast enough to justify our calculation. This important condition can be phrased by saying that for all physical values of $(q_x, q_z)$, we must not touch the edge of the spectrum:

$$\frac{1}{\rho} + \gamma_x + \gamma_z \lambda_{\min}(\nu) > 0. \tag{21}$$

We justify here eq. (21) for all physical values of $(q_x, q_z)$. We will combine three arguments:

(i) Note that in the replica calculation, cf Section A.2.3, the matrix $\Gamma^z$ is assumed to be Hermitian positive in the $p \downarrow 0$ limit. Since $\Gamma_z = 0$, this implies that we must have $\lambda_z \geq 0$.

(ii) The saddle point equation on $q_x$ yields[1]:

$$\hat{q}_x = \frac{q_x}{\rho(\rho - q_x)} - \gamma_x. \tag{22}$$

(iii) Finally, we will derive a lower bound on $q_x$. Note that, as one can see in $I_0$ from Section A.3.2, $q_x$ is the optimal overlap achievable in the following scalar inference problem [9]:

$$Y_0 = \sqrt{\hat{q}_x} X^\star + Z, \tag{23}$$

in which one observes $Y_0$ and is given $P_0$ the prior distribution on $X^\star$, and the noise $Z$ is distributed according to $\mathcal{N}_\beta(0, 1)$. It is known that the optimal estimator is given by the average of $\mathbb{E}[x|Y]$ under the posterior distribution, whose density is proportional to $P_0(x) e^{-\frac{\beta}{2}|y - \sqrt{\hat{q}_x} x|^2}$. If this is untractable for generic $P_0$, we can consider a suboptimal estimation by using a Gaussian prior with variance $\rho$ in the estimation procedure (so that the problem is mismatched). This yields the bound:

$$q_x \geq \int \mathcal{D}_\beta \xi \frac{\left[ \int_{\mathbb{K}} P_0(\mathrm{d}x) \, x \, e^{-\frac{\beta \hat{q}_x}{2}|x|^2 + \beta \sqrt{\hat{q}_x} x \cdot \xi} \right] \cdot \left[ \int_{\mathbb{K}} \mathrm{d}x \, x \, e^{-\frac{\beta|x|^2}{2\rho}} e^{-\frac{\beta \hat{q}_x}{2}|x|^2 + \beta \sqrt{\hat{q}_x} x \cdot \xi} \right]}{\int_{\mathbb{K}} \mathrm{d}x \, e^{-\frac{\beta|x|^2}{2\rho}} e^{-\frac{\beta \hat{q}_x}{2}|x|^2 + \beta \sqrt{\hat{q}_x} x \cdot \xi}}. \tag{24}$$

This can easily be simplified by performing the Gaussian integral, and yields the bound:

$$q_x \geq \frac{\rho^2 \hat{q}_x}{1 + \rho \hat{q}_x}. \tag{25}$$

Combining $(ii)$ and $(iii)$ gives:

$$q_x \geq \rho - \frac{\rho - q_x}{1 - \gamma_x(\rho - q_x)}. \tag{26}$$

Since $q_x \in [0, \rho]$, this implies in particular that $\gamma_x \geq 0$. Using this along with $(i)$, this implies:

$$\frac{1}{\rho} + \gamma_x + \gamma_z \lambda_{\min}(\nu) \geq \frac{1}{\rho} > 0, \tag{27}$$

which is what we wanted to show.

## B  Derivation of the weak-recovery threshold

We detail here the derivation of the algorithmic weak-recovery threshold $\alpha_{\mathrm{WR,Algo}}$. As discussed in Section 3, the weak-recovery threshold can be identified as the sample complexity for which the trivial fixed point $q_x = q_z = \hat{q}_x = \hat{q}_z = \gamma_x = \gamma_z = 0$ of the state evolution equations becomes linearly unstable (when it no longer is a local maximum of the free entropy potential). Consider therefore the state evolution equations, which we repeat here for convenience in a detailed form:

$$
\begin{cases}
q_x = \displaystyle\int_{\mathbb{K}} \mathcal{D}_\beta \xi \frac{\left| \int_{\mathbb{K}} P_0(\mathrm{d}x)\, x\, e^{-\frac{\beta}{2}\hat{q}_x |x|^2 + \beta\sqrt{\hat{q}_x} x \cdot \xi} \right|^2}{\int_{\mathbb{K}} P_0(\mathrm{d}x)\, e^{-\frac{\beta}{2}\hat{q}_x |x|^2 + \beta\sqrt{\hat{q}_x} x \cdot \xi}}, & \text{(28a)} \\[2em]
q_z = \dfrac{1}{\hat{Q}_z + \hat{q}_z} \left[ \dfrac{\hat{q}_z}{\hat{Q}_z} + \displaystyle\int \mathrm{d}y\, \mathcal{D}_\beta \xi \frac{\left| \int \mathcal{D}_\beta z\, z\, P_{\mathrm{out}}\left(y \middle| \frac{z}{\sqrt{\hat{Q}_z + \hat{q}_z}} + \sqrt{\frac{\hat{q}_z}{\hat{Q}_z(\hat{Q}_z + \hat{q}_z)}} \xi \right) \right|^2}{\int \mathcal{D}_\beta z P_{\mathrm{out}}\left(y \middle| \frac{z}{\sqrt{\hat{Q}_z + \hat{q}_z}} + \sqrt{\frac{\hat{q}_z}{\hat{Q}_z(\hat{Q}_z + \hat{q}_z)}} \xi \right)} \right], & \text{(28b)} \\[2em]
\hat{q}_x = \dfrac{q_x}{\rho(\rho - q_x)} - \gamma_x, & \text{(28c)} \\[1.5em]
\hat{q}_z = \dfrac{q_z}{Q_z(Q_z - q_z)} - \gamma_z, & \text{(28d)} \\[1.5em]
\rho - q_x = \left\langle \dfrac{1}{\rho^{-1} + \gamma_x + \lambda\gamma_z} \right\rangle_\nu, & \text{(28e)} \\[1.5em]
\alpha(Q_z - q_z) = \left\langle \dfrac{\lambda}{\rho^{-1} + \gamma_x + \lambda\gamma_z} \right\rangle_\nu. & \text{(28f)}
\end{cases}
$$

Letting $q_x = q_z = \hat{q}_x = \hat{q}_z = \gamma_x = \gamma_z = 0$, it is clear that the equations are satisfied if the signal distribution $P_0$ and the likelihood $P_{\mathrm{out}}$ satisfy the following symmetry conditions:

$$|x_1| = |x_2| \Rightarrow P_0(x_1) = P_0(x_2) \qquad \text{and} \qquad |z_1| = |z_2| \Rightarrow P_{\mathrm{out}}(y|z_1) = P_{\mathrm{out}}(y|z_2).$$

Assuming these conditions hold, we are interested in studying the linear stability of this local maximum. Recalling that $Q_z = \rho\langle\lambda\rangle_\nu/\alpha$, the first, third and fourth equations of eq. (28) can be linearized:

$$\delta q_x = \rho^2 \delta\hat{q}_x, \qquad \delta\hat{q}_x = \frac{\delta q_x}{\rho^2} - \delta\gamma_x, \qquad \delta\hat{q}_z = \frac{\alpha^2 \delta q_z}{\rho^2 \langle\lambda\rangle_\nu^2} - \delta\gamma_z. \tag{29}$$

Now focusing on the second state evolution equation (28), it can be linearized to give:

$$\delta q_z = \frac{\rho^2 \langle\lambda\rangle_\nu^2}{\alpha^2} \delta\hat{q}_z \left(1 + \int_{\mathbb{R}} \mathrm{d}y \frac{\left| \int_{\mathbb{K}} \mathcal{D}_\beta z\, (|z|^2 - 1)\, P_{\mathrm{out}}\left(y \middle| \sqrt{\frac{\rho\langle\lambda\rangle_\nu}{\alpha}} z \right) \right|^2}{\int_{\mathbb{K}} \mathcal{D}_\beta z\, P_{\mathrm{out}}\left(y \middle| \sqrt{\frac{\rho\langle\lambda\rangle_\nu}{\alpha}} z \right)} \right). \tag{30}$$

Finally, it remains to compute the infinitesimal variation for $\delta\gamma_x, \delta\gamma_z$:

$$
\begin{cases}
\delta\gamma_x = \dfrac{\langle\lambda^2\rangle_\nu}{\rho^2[\langle\lambda^2\rangle_\nu - \langle\lambda\rangle_\nu^2]}\delta q_x - \dfrac{\alpha\langle\lambda\rangle_\nu}{\rho^2[\langle\lambda^2\rangle_\nu - \langle\lambda\rangle_\nu^2]}\delta q_z, & \text{(31a)} \\[1.5em]
\delta\gamma_z = -\dfrac{\langle\lambda\rangle_\nu}{\rho^2[\langle\lambda^2\rangle_\nu - \langle\lambda\rangle_\nu^2]}\delta q_x + \dfrac{\alpha}{\rho^2[\langle\lambda^2\rangle_\nu - \langle\lambda\rangle_\nu^2]}\delta q_z. & \text{(31b)}
\end{cases}
$$

Combining eqs. (29),(30),(31), we can simplify the system to a closed set equations with only $(\delta q_x, \delta \hat{q}_x, \delta q_z, \delta \hat{q}_z)$. Given the usual heuristics of the replica method and its link with the state evolution equations of message-passing algorithms [2, 6, 10], one can conjecture that the simplest iteration scheme corresponds to the state evolution of the G-VAMP message passing algorithm:

$$
\begin{cases}
\delta q_x^{t+1} = \rho^2 \delta \hat{q}_x^t, & \text{(32a)} \\[2ex]
\delta q_z^{t+1} = \dfrac{\rho^2 \langle \lambda \rangle_\nu^2}{\alpha^2} \delta \hat{q}_z^t \left( 1 + \displaystyle\int_{\mathbb{R}} \mathrm{d}y \, \dfrac{\left| \int_{\mathbb{K}} \mathcal{D}_\beta z \, (|z|^2 - 1) \, P_{\mathrm{out}}\left(y \middle| \sqrt{\frac{\rho\langle\lambda\rangle_\nu}{\alpha}} z\right) \right|^2}{\int_{\mathbb{K}} \mathcal{D}_\beta z \, P_{\mathrm{out}}\left(y \middle| \sqrt{\frac{\rho\langle\lambda\rangle_\nu}{\alpha}} z\right)} \right), & \text{(32b)} \\[4ex]
\delta \hat{q}_x^{\,t} = -\dfrac{\langle\lambda\rangle_\nu^2}{\rho^2[\langle\lambda^2\rangle_\nu - \langle\lambda\rangle_\nu^2]} \delta q_x^t + \dfrac{\alpha\langle\lambda\rangle_\nu}{\rho^2[\langle\lambda^2\rangle_\nu - \langle\lambda\rangle_\nu^2]} \delta q_z^t, & \text{(32c)} \\[3ex]
\delta \hat{q}_z^{\,t} = \dfrac{\langle\lambda\rangle_\nu}{\rho^2[\langle\lambda^2\rangle_\nu - \langle\lambda\rangle_\nu^2]} \delta q_x^t + \left[ \dfrac{\alpha^2}{\rho^2\langle\lambda\rangle_\nu^2} - \dfrac{\alpha}{\rho^2[\langle\lambda^2\rangle_\nu - \langle\lambda\rangle_\nu^2]} \right] \delta q_z^t. & \text{(32d)}
\end{cases}
$$

From these equations, one can easily see that a linear instability of the trivial fixed points appears at $\alpha = \alpha_{\mathrm{WR,Algo}}$ satisfying the equation:

$$
\alpha_{\mathrm{WR,Algo}} = \frac{\langle\lambda\rangle_\nu^2}{\langle\lambda^2\rangle_\nu} \left( 1 + \left[ \int_{\mathbb{R}} \mathrm{d}y \, \frac{\left| \int_{\mathbb{K}} \mathcal{D}_\beta z \, (|z|^2 - 1) \, P_{\mathrm{out}}\left(y \middle| \sqrt{\frac{\rho\langle\lambda\rangle_\nu}{\alpha_{\mathrm{WR,Algo}}}} z\right) \right|^2}{\int_{\mathbb{K}} \mathcal{D}_\beta z \, P_{\mathrm{out}}\left(y \middle| \sqrt{\frac{\rho\langle\lambda\rangle_\nu}{\alpha_{\mathrm{WR,Algo}}}} z\right)} \right]^{-1} \right). \tag{33}
$$

Indeed at $\alpha = \alpha_{\mathrm{WR,Algo}}$, the modulus of all the eigenvalues of the size-4 matrix of the linear system (32) cross 1.

## C   The full recovery transition

In this section, we assume a Gaussian standard prior $P_0 = \mathcal{N}_\beta(0, 1)$ and a noiseless phase retrieval channel, and we show that information-theoretic full recovery is achieved exactly at $\alpha = \alpha_{\mathrm{FR,IT}} \equiv \beta(1 - \nu(\{0\}))$. We can assume without loss of generality that $\langle\lambda\rangle_\nu = \alpha$, as this amounts to a simple rescaling of $\boldsymbol{\Phi}$, irrelevant under the noiseless channel. This implies in particular that $Q_z = \hat{Q}_z = 1$.

### C.1   The state evolution equations

Since we assumed a Gaussian prior, we have, with $P_{\mathrm{out}}(y|z) = \delta(y - |z|^2)$:

$$
\begin{cases}
q_z = \dfrac{1}{1 + \hat{q}_z} \left[ \hat{q}_z + \displaystyle\int \mathrm{d}y \int_{\mathbb{K}} \mathcal{D}_\beta \xi \, \dfrac{\left| \int_{\mathbb{K}} \mathcal{D}_\beta z \, z \, P_{\mathrm{out}}\left(y \middle| \frac{z}{\sqrt{1+\hat{q}_z}} + \sqrt{\frac{\hat{q}_z}{1+\hat{q}_z}}\xi\right) \right|^2}{\int_{\mathbb{K}} \mathcal{D}_\beta z \, P_{\mathrm{out}}\left(y \middle| \frac{z}{\sqrt{1+\hat{q}_z}} + \sqrt{\frac{\hat{q}_z}{1+\hat{q}_z}}\xi\right)} \right], & \text{(34a)} \\[4ex]
\hat{q}_x = \dfrac{q_x}{1 - q_x}, & \text{(34b)} \\[2ex]
\hat{q}_z = \dfrac{q_z}{1 - q_z} - \gamma_z, & \text{(34c)} \\[2ex]
q_x = \alpha \gamma_z (1 - q_z), & \text{(34d)} \\[2ex]
\alpha(1 - q_z) = \left\langle \dfrac{\lambda}{1 + \lambda\gamma_z} \right\rangle_\nu. & \text{(34e)}
\end{cases}
$$

Comparing these equations to Conjecture 2.1, one can see that we imposed $\gamma_x = 0$, a straightforward consequence of the Gaussian prior (see Section E where this calculation is detailed for a different purpose).

### C.2   Noisy phase retrieval with small variance

We wish to show that the free entropy of the full recovery solution is the global maximum of the free entropy potential for $\alpha > \alpha_{\mathrm{IT}}$, while it is never the case for $\alpha < \alpha_{\mathrm{IT}}$. However, under a

173 noiseless channel, the free entropy potential might diverge in this point, which indicates towards a
174 regularization procedure. Therefore we consider a noisy Gaussian channel with noise $\Delta > 0$:

$$P_{\text{out}}(y|z) = \frac{1}{\sqrt{2\pi\Delta}} \exp\left\{ -\frac{1}{2\Delta}(y - |z|^2)^2 \right\}. \tag{35}$$

We will compute the limit, as $\Delta \downarrow 0$, of the free entropy of the "almost perfect" recovery fixed point.
We look for a solution close to the point which corresponds to the best possible recovery:

$$\begin{cases} q_x = 1 - \nu(\{0\}), & \text{(36a)} \\ q_z = 1. & \text{(36b)} \end{cases}$$

Indeed it is easy to see that $q_x \leq 1 - \nu(\{0\})$ since $\text{rk}[\mathbf{\Phi}^\dagger \mathbf{\Phi}] \sim n(1 - \nu(\{0\}))$. We are thus looking
for a fixed point of the state evolution equations (34) that satisfies:

$$\begin{cases} q_x = 1 - \nu(\{0\}) + \mathcal{O}_\Delta(1), & \text{(37a)} \\ q_z = 1 + \mathcal{O}_\Delta(1), & \text{(37b)} \\ \hat{q}_x^{-1} = \nu(\{0\})/(1 - \nu(\{0\})) + \mathcal{O}_\Delta(1), & \text{(37c)} \\ \hat{q}_z^{-1} = \mathcal{O}_\Delta(1). & \text{(37d)} \end{cases}$$

175 Let us now precise the asymptotics of these quantities as $\Delta \downarrow 0$. By eq. (34d), we find easily:

$$\gamma_z \sim \frac{1 - \nu(\{0\})}{\alpha(1 - q_z)}. \tag{38}$$

176 Then from eq. (34c), we also have:

$$\hat{q}_z \sim \frac{\alpha - 1 + \nu(\{0\})}{\alpha(1 - q_z)}. \tag{39}$$

177 Note that if $\alpha \leq 1$, then necessarily $\nu(\{0\}) \geq 1 - \alpha$, so that the quantity in the numerator is always
178 positive. We now turn to eq. (34a). We assume the scaling $\hat{q}_z^{-1} = c\Delta + \mathcal{O}_\Delta(\Delta)$. We have by Gaussian
179 integration by parts and using the specific form of $P_{\text{out}}$:

$$\int \mathrm{d}y \mathcal{D}_\beta \xi \frac{\left| \int \mathcal{D}_\beta z\, z\, P_{\text{out}}\left(y \middle| \frac{z}{\sqrt{1+\hat{q}_z}} + \sqrt{\frac{\hat{q}_z}{1+\hat{q}_z}}\xi\right) \right|^2}{\int \mathcal{D}_\beta z P_{\text{out}}\left(y \middle| \frac{z}{\sqrt{1+\hat{q}_z}} + \sqrt{\frac{\hat{q}_z}{1+\hat{q}_z}}\xi\right)}$$

$$= \frac{1}{(1+\hat{q}_z)} \int \mathrm{d}y \mathcal{D}_\beta \xi \frac{\left| \int \mathcal{D}_\beta z\, P'_{\text{out}}\left(y \middle| \frac{z}{\sqrt{1+\hat{q}_z}} + \sqrt{\frac{\hat{q}_z}{1+\hat{q}_z}}\xi\right) \right|^2}{\int \mathcal{D}_\beta z P_{\text{out}}\left(y \middle| \frac{z}{\sqrt{1+\hat{q}_z}} + \sqrt{\frac{\hat{q}_z}{1+\hat{q}_z}}\xi\right)} \sim \frac{4}{\Delta(1+\hat{q}_z)} \sim 4c.$$

180 Gaussian integration by parts and our conventions for derivatives of real functions of complex
181 variables are summarized in Section F.2. This yields that $1 - q_z = \Delta c(1 - 4c) + \mathcal{O}_\Delta(1)$. Combining
182 this result with eq. (39), we have

$$c(1 - 4c) = c\left[\frac{\alpha - 1 + \nu(\{0\})}{\alpha}\right].$$

This implies $c = (1 - \nu(\{0\}))/(4\alpha)$, and we finally obtain the leading order asymptotics of $q_z, \hat{q}_z, \gamma_z$
as $\Delta \to 0$:

$$\begin{cases} \hat{q}_z = \frac{4\alpha}{(1 - \nu(\{0\}))\Delta} + \mathcal{O}_\Delta\left(\Delta^{-1}\right), & \text{(40a)} \\[3mm] 1 - q_z = \frac{(1 - \nu(\{0\}))(\alpha - 1 + \nu(\{0\}))}{4\alpha^2}\Delta + \mathcal{O}_\Delta(\Delta), & \text{(40b)} \\[3mm] \gamma_z = \frac{4\alpha}{\Delta(\alpha - 1 + \nu(\{0\}))} + \mathcal{O}_\Delta(\Delta^{-1}). & \text{(40c)} \end{cases}$$

183    Let us now compute the asymptotics of the three auxiliary functions $I_0$, $I_{\text{out}}$ and $I_{\text{int}}$ of Conjecture 2.1:

$$I_0(q_x) = \frac{\beta}{2}[q_x + \ln(1 - q_x)],$$

$$I_{\text{out}}(q_z) = -\frac{\beta \hat{q}_z q_z}{2} - \frac{\beta}{2}\ln(1 + \hat{q}_z) + \frac{\beta \hat{q}_z}{2} + \int dy \mathcal{D}\xi \, J(\hat{q}_z, y, \xi) \ln J(\hat{q}_z, y, \xi),$$

$$J(\hat{q}_z, y, \xi) \equiv \int \mathcal{D}z P_{\text{out}}\Big(y \Big| \frac{z}{\sqrt{1 + \hat{q}_z}} + \sqrt{\frac{\hat{q}_z}{1 + \hat{q}_z}}\xi\Big),$$

$$I_{\text{int}}(q_x, q_z) = \frac{\beta}{2}[\alpha(1 - q_z)\gamma_z - \langle\ln(1 + \lambda\gamma_z)\rangle_\nu - \ln(1 - q_x) - q_x - \alpha \ln(1 - q_z) - \alpha q_z].$$

184    Using eq. (40) and the specific form of the channel, we reach:

$$I_0(q_x) + I_{\text{int}}(q_x, q_z) \sim -\frac{\beta(\alpha - 1 + \nu(\{0\}))}{2}\ln\Delta,$$

$$I_{\text{out}}(q_z) \sim \frac{(\beta - 1)}{2}\ln\Delta.$$

185    Therefore when considering the total free entropy we have

$$I_0(q_x) + I_{\text{int}}(q_x, q_z) + \alpha I_{\text{out}}(q_z) \sim \frac{\alpha(\beta - 1) - \beta(\alpha - 1 + \nu(\{0\}))}{2}\ln\Delta,$$

$$\sim \frac{\beta(1 - \nu(\{0\})) - \alpha}{2}\ln\Delta.$$

186    This implies that the full recovery point has a free entropy of $-\infty$ for $\alpha < \alpha_{\text{FR,IT}} \equiv \beta(1 - \nu(\{0\}))$,
187    and $+\infty$ for $\alpha > \alpha_{\text{FR,IT}}$. Thus this point is always the global maximum of the free entropy for
188    $\alpha > \alpha_{\text{FR,IT}}$, while it is never the case for $\alpha < \alpha_{\text{FR,IT}}$, which ends our argument.

# 189   D    Proof of Theorem 2.2

190    In all this section, we provide the proof of Theorem 2.2 under $(H0)$,(h1),(h2),(h3), and we will
191    work under these hypotheses. In Section D.6, we show how the proof can be extended to hypothe-
192    ses $(H0)$,$(h'1)$.
193    First, we simplify the conjectured expression of the free entropy of Conjecture 2.1 using the particular
194    form of the prior $P_0$ and of the sensing matrix $\mathbf{\Phi}$. Finally, using (h1),(h2),(h3) and a proof similar to
195    the one of [9, 11], we give a rigorous derivation of this simplified expression. Note that with respect
196    to the analysis of [9, 11], there are two main novelties in our setting:

197      $(i)$ The sensing matrix $\mathbf{\Phi}$ is not i.i.d. but has a well-controlled structure, see (h2).

198      $(ii)$ The variables can be complex numbers. We will argue that the arguments generalize to
199         this case. The physical reason of this generalization is that even in the complex setting, the
200         overlap will concentrate on a real positive number, as a consequence of Bayes-optimality.

201    First, we note that we can simplify the replica conjecture under the considered hypotheses:

202    **Proposition D.1.** *Under $(H0)$,(h1),(h2),(h3), the replica conjecture 2.1 for the free entropy $f_n \equiv$*
203    *$\frac{1}{n}\mathbb{E}\ln\mathcal{Z}_n(\mathbf{Y})$ is equivalent to:*

$$\lim_{n \to \infty} f_n = \sup_{\hat{q} \geq 0} \inf_{q \in [0, Q_z]} \Big[\frac{\beta \hat{q}}{2}(\mathbb{E}_{\nu_B}[X] - \delta q) - \frac{\beta}{2}\mathbb{E}_{\nu_B}\ln(1 + \hat{q}X) + \alpha\Psi_{\text{out}}(q)\Big], \qquad (41)$$

204    *with $Q_z = \mathbb{E}_{\nu_B}[X]/\delta$ and $\Psi_{\text{out}}$ defined in terms of the auxiliary functions introduced in eq. (7):*

$$\Psi_{\text{out}}(q) \equiv \mathbb{E}_\xi \int_{\mathbb{R}} dy \, \mathcal{Z}_{\text{out}}(y; \sqrt{q}\xi, Q_z - q) \ln \mathcal{Z}_{\text{out}}(y; \sqrt{q}\xi, Q_z - q).$$

205    Proposition D.1 is proven in Section E. To prove the free entropy statement of Theorem 2.2, we
206    therefore just need to show:

**Lemma D.2.** *Under the assumptions of Proposition D.1, the limit of the free entropy $f_n \equiv \frac{1}{n}\mathbb{E}\ln\mathcal{Z}_n(\boldsymbol{Y})$ is given by eq. (41).*

The following of this section is dedicated to the proof of Lemma D.2. We will conclude the proof of Theorem 2.2 in Section D.5 and Section D.6, dedicated respectively to the proof of the MMSE statement and the extension of the proof to hypotheses $(H0),(h'1)$. The main idea of our proof is to reduce the problem of Lemma D.2 to a Generalized Linear Model with a Gaussian sensing matrix, but a non-i.i.d. prior. We make use of the "SVD" decomposition of $\mathbf{B}/\sqrt{n} = \mathbf{USV}^\dagger$, with $\mathbf{U} \in \mathcal{U}_\beta(p)$, $\mathbf{V} \in \mathcal{U}_\beta(n)$, and $\mathbf{S} \in \mathbb{R}^{p\times n}$ a pseudo-diagonal matrix with positive elements. Leveraging on the fact that the prior $P_0$ is Gaussian, and that $\mathbf{W}$ is an i.i.d. Gaussian matrix independent of $\mathbf{B}$, one can see that our estimation problem is formally equivalent to an usual Generalized Linear Model with $m$ measurements, a signal of dimension $p$, and a Gaussian i.i.d. sensing matrix. This is very close to the setup of [9], a key difference being that here the prior distribution on the data $\mathbf{Z}^\star \in \mathbb{K}^p$ is defined as

- If $\delta \leq 1$, for every $k \in \{1, \cdots, p\}$, $Z_k^\star$ is distributed as $S_k X_k^\star$ with $X_k^\star \overset{\text{i.i.d.}}{\sim} P_0$.

- If $\delta \geq 1$, for every $k \in \{1, \cdots, n\}$, $Z_k^\star$ is distributed as $S_k X_k^\star$ with $X_k^\star \overset{\text{i.i.d.}}{\sim} P_0$, while for every $k \in \{n+1, \cdots, p\}$, $Z_k^\star$ is almost surely 0.

More precisely, we can define rigorously the prior $P_0^{(\mathbf{S})}$ described above by its linear statistics. For any continuous bounded function $g : \mathbb{K}^p \to \mathbb{R}$, one has:

$$\int_{\mathbb{K}^p} P_0^{(\mathbf{S})}(\mathrm{d}\mathbf{z})g(\mathbf{z}) \equiv \int_{\mathbb{K}^n} \left\{\prod_{i=1}^n P_0(\mathrm{d}x_i)\right\}g(\{\mathbb{1}[k \leq n]S_k x_k\}_{k=1}^p). \tag{42}$$

Hypothesis (h1) implies that we will consider $P_0 = \mathcal{N}_\beta(0,1)$. In the following of the section, we give the detailed sketch of the proof of Lemma D.2. Some facts and lemmas will be a generalization or a consequence of the works of [9] and [11], and we will refer to them when necessary.

## D.1 Interpolating estimation problem

Recall that $Q_z \equiv \rho\langle\lambda\rangle_\nu/\alpha = \mathbb{E}_{\nu_B}[X]/\delta$, and the definition of $\Psi_{\text{out}}$ in Proposition D.1. We define as well:

$$r_{\max} \equiv \sup_{q\in[0,Q_z]} \Psi_{\text{out}}(q), \tag{43}$$

$$\Psi_0^{(\nu)}(r) \equiv \frac{\beta}{2}\left[r\mathbb{E}_{\nu_B}[X] - \mathbb{E}_{\nu_B}\ln(1+rX)\right], \qquad 0 \leq r \leq r_{\max}. \tag{44}$$

Since $\nu_B \neq \delta_0$ by hypothesis, we can easily check that $\Psi_0^{(\nu)}$ is strictly convex, $\mathcal{C}^2$ and non-decreasing on $[0, r_{\max}]$. By Proposition 18 of [9], which directly generalizes to the complex case, we know as well that $\Psi_{\text{out}}$ is convex, $\mathcal{C}^2$, and non-decreasing on $[0, Q_z]$, and thus $r_{\max} = \Psi_{\text{out}}(Q_z)$. Let us fix an arbitrary sequence $s_n > 0$ that goes to 0 as $n$ goes to infinity. We fix $\epsilon_2 \in [s_n, 2s_n]$, and $\epsilon_1 \in \mathcal{D}_n^\beta$, with

$$\mathcal{D}_n^\beta \equiv \{\lambda \in \mathcal{S}_\beta(\mathbb{R}) : \forall l \in \{1, \beta\}, \lambda_{ll} \in (2\beta s_n, (2\beta+1)s_n), \forall l \neq l' \in \{1, \beta\}, \lambda_{ll'} \in (s_n, 2s_n)\}.$$

$\mathcal{D}_n^\beta$ is composed of strictly diagonally dominant matrices with positive entries, which implies that $\mathcal{D}_n \subset \mathcal{S}_\beta^+(\mathbb{R})$. Let $q_\epsilon : [0,1] \to [0, Q_z]$, $r_\epsilon : [0,1] \to [0, r_{\max}]$ be two continuous "interpolation" functions. For all $\epsilon \in \mathcal{D}_n^\beta \times [s_n, 2s_n]$, and all $t \in [0,1]$ we define:

$$\mathcal{S}_\beta^+(\mathbb{R}) \ni R_1(t,\epsilon) \equiv \epsilon_1 + \left(\int_0^t r_\epsilon(v)\mathrm{d}v\right)\mathbb{1}_\beta, \qquad \mathbb{R}_+ \ni R_2(t,\epsilon) \equiv \epsilon_2 + \int_0^t q_\epsilon(v)\mathrm{d}v. \tag{45}$$

We consider the following decoupled observation channels:

$$\begin{cases} \left\{Y_{t,\mu} \sim P_{\text{out}}\left(\cdot \left|\sqrt{\frac{1-t}{p}}[\mathbf{WZ}^\star]_\mu + \sqrt{R_2(t,\epsilon)}V_\mu + \sqrt{Q_zt - R_2(t,\epsilon) + 2s_n}A_\mu^\star\right)\right\}_{\mu=1}^m & \text{(46a)} \\ \tilde{\mathbf{Y}}_t = (R_1(t,\epsilon))^{1/2} \star \mathbf{Z}^\star + \zeta, & \text{(46b)} \end{cases}$$

where $V_\mu, A^\star_\mu \overset{\text{i.i.d.}}{\sim} \mathcal{N}_\beta(0,1)$, and $\boldsymbol{\zeta} \sim \mathcal{N}_\beta(0, \mathbb{1}_p)$. The prior distribution on $\mathbf{Z}^\star$ is given by $P_0^{(\mathbf{S})}$ in eq. (42). We assume that $\{V_\mu\}_{\mu=1}^m$ is known, and the inference problem is to recover both $\mathbf{A}^\star \in \mathbb{K}^m$ and $\mathbf{Z}^\star \in \mathbb{K}^p$ from the observations $(\tilde{\mathbf{Y}}_t, \{Y_{t,\mu}\}_{\mu=1}^m)$. Note that $R_1 \in \mathcal{S}_\beta^+(\mathbb{R})$, so its (matrix) square root is always uniquely defined. Recall finally the definition of the $\star$ product in Section F.1. In the following we will study the system of eq. (46). In order to state our results fully rigorously, we need to add an hypothesis that can easily be relaxed:

$(h1^\star)$ The prior $P_0$ has bounded support.

Under this hypothesis, $P_0^{(\mathbf{S})}$ is still defined by eq. (42), and we can study the system of eq. (46). Nonetheless, this assumption *a priori* rules out a Gaussian prior for $P_0$, and thus the correspondence between the system of eq. (46) and our original model. However, following the arguments of [9], hypothesis $(h1^\star)$ can very easily be relaxed to the existence of the second moment of $P_0$, which is then consistent with a Gaussian prior. In the following, we will thus work under hypothesis (h1), but we will sometimes as well use hypothesis $(h1^\star)$ without loss of generality. We define $u_y(z) \equiv \ln P_{\text{out}}(y|z)$, and

$$S_{t,\mu} \equiv \sqrt{\frac{1-t}{n}}[\mathbf{WZ}^\star]_\mu + \sqrt{R_2(t,\epsilon)}V_\mu + \sqrt{Q_z t - R_2(t,\epsilon) + 2s_n}A^\star_\mu, \qquad (47)$$

$$s_{t,\mu} \equiv \sqrt{\frac{1-t}{n}}[\mathbf{Wz}]_\mu + \sqrt{R_2(t,\epsilon)}V_\mu + \sqrt{Q_z t - R_2(t,\epsilon) + 2s_n}a_\mu. \qquad (48)$$

The posterior distribution in this model can then be written as:

$$\mathbb{P}_{n,t,\epsilon}\left(\mathbf{z}, \mathbf{a} \middle| \mathbf{Y}_t, \tilde{\mathbf{Y}}_t\right) \mathrm{d}\mathbf{z}\, \mathrm{d}\mathbf{a} \equiv \frac{1}{\mathcal{Z}_{n,t,\epsilon}(\mathbf{Y}_t, \tilde{\mathbf{Y}}_t)} P_0^{(\mathbf{S})}(\mathrm{d}\mathbf{z}) \mathcal{D}_\beta \mathbf{a}\, e^{-\mathcal{H}_{t,\epsilon}(\mathbf{z},\mathbf{a};\mathbf{Y}_t,\tilde{\mathbf{Y}}_t,\mathbf{W},\mathbf{V})}. \qquad (49)$$

To keep the notations lighter we omitted the conditioning on the variables $\mathbf{V}, \mathbf{W}$ which are assumed to be known. We defined the Hamiltonian:

$$\mathcal{H}_{t,\epsilon}(\mathbf{z}, \mathbf{a}; \mathbf{Y}_t, \tilde{\mathbf{Y}}_t, \mathbf{W}, \mathbf{V}) \equiv -\sum_{\mu=1}^m u_{Y_{t,\mu}}(s_{t,\mu}) + \frac{\beta}{2}\sum_{k=1}^p \left|\tilde{Y}_{t,k} - (R_1(t,\epsilon))^{1/2} \star z_k\right|^2. \qquad (50)$$

For any $t \in (0,1)$, we define the free entropy (the expectation is over all "quenched" variables, including $\mathbf{S}$ if it is random):

$$f_{n,\epsilon}(t) \equiv \frac{1}{n}\mathbb{E}\ln \mathcal{Z}_{n,t,\epsilon}(\mathbf{Y}_t, \tilde{\mathbf{Y}}_t).$$

The following lemma gives the $t=0$ and $t=1$ limits of the free entropy:

**Lemma D.3.** $f_{n,\epsilon}(t)$ *admits the following limit values for* $t \in \{0,1\}$:

$$\begin{cases} f_{n,\epsilon}(0) = f_n - \dfrac{\beta\delta}{2} + \mathcal{O}_n(1), \\[2mm] f_{n,\epsilon}(1) = \Psi_0^{(\nu)}\left(\displaystyle\int_0^1 r_\epsilon(t)\mathrm{d}t\right) - \dfrac{\beta}{2}\left[\delta + \mathbb{E}_{\nu_B}[X]\displaystyle\int_0^1 r_\epsilon(t)\mathrm{d}t\right] + \alpha\Psi_{\text{out}}\left(\displaystyle\int_0^1 q_\epsilon(t)\mathrm{d}t\right) + \mathcal{O}_n(1). \end{cases}$$

*Proof of Lemma D.3.* Using Lemma 5.1 of [11], there exists a constant $C > 0$ such that for all $\epsilon \in \mathcal{D}_n^\beta \times [s_n, 2s_n]$, one has $|f_{n,\epsilon}(0) - f_{n,(0,0)}(0)| \leq Cs_n$. The proof of the value of $f_{n,\epsilon}(0)$ is then straightforwardly done by plugging $t=0$ into the definition of $f_{n,\epsilon}$. At $t=1$, the interpolation channels of eq. (46) decouple, and we have:

$$f_{n,\epsilon}(1) = \frac{1}{n}\mathbb{E}\ln\int_{\mathbb{K}^p} P_0^{(\mathbf{S})}(\mathrm{d}\mathbf{z})\exp\left\{-\frac{\beta}{2}\sum_{k=1}^p\left|\tilde{Y}_{1,k} - \left(\epsilon_1 + \int_0^1 r_\epsilon(t)\mathbb{1}_\beta \mathrm{d}t\right)^{1/2}\star z_k\right|^2\right\}$$

$$+ \frac{m}{n}\mathbb{E}_{Y_1,V}\ln P_{\text{out}}\left(Y_1\middle|\left(\epsilon_2 + \int_0^1 q_\epsilon(t)\mathrm{d}t\right)^{1/2}V + \left(Q_z + 2s_n - \epsilon_2 - \int_0^1 q_\epsilon(t)\mathrm{d}t\right)^{1/2}a\right),$$

$$= \frac{1}{n}\sum_{i=1}^{\min(n,p)}\int_{\mathbb{K}}\mathrm{d}Y\mathcal{D}_\beta X\, \frac{e^{-\frac{\beta}{2}|Y - S_i(R_1(1,\epsilon))^{1/2}\star X|^2}}{(2\pi/\beta)^{\beta/2}}\ln\left\{\int \mathcal{D}_\beta x\, e^{-\frac{\beta}{2}|Y - S_i(R_1(1,\epsilon))^{1/2}\star x|^2}\right\}$$

$$+ \frac{1}{n}\sum_{i=\min(n,p)+1}^p \int_{\mathbb{K}}\mathrm{d}Y\, \frac{e^{-\frac{\beta}{2}|Y|^2}}{(2\pi/\beta)^{\beta/2}}\ln\left\{e^{-\frac{\beta}{2}|Y|^2}\right\}\right) + \alpha\Psi_{\text{out}}\left(\int_0^1 q_\epsilon(t)\mathrm{d}t\right) + \mathcal{O}_n(1).$$

263 Recall that $R_1(1, \epsilon) = (\int_0^1 r_\epsilon(t)\mathrm{d}t)\mathbb{1}_\beta + \mathcal{o}_n(1)$, so that up to $\mathcal{o}_n(1)$ terms the Gaussian integration
264 on $X, x$ can be performed, which yields a Gaussian integration on $Y$, and we reach in the end:

$$f_{n,\epsilon}(1) = -\frac{\beta p}{2n} - \frac{\beta}{2n} \sum_{i=1}^{\min(n,p)} \ln\left(1 + S_i^2 \int_0^1 r_\epsilon(t)\mathrm{d}t\right) + \alpha\Psi_{\text{out}}\left(\int_0^1 q_\epsilon(t)\mathrm{d}t\right) + \mathcal{o}_n(1).$$

265 Recall that $\nu_B$ is defined as the asymptotic eigenvalue distribution of $\mathbf{S}^\mathsf{T}\mathbf{S}$. By (h3) we have:

$$f_{n,\epsilon}(1) = \Psi_0^{(\nu)}\left(\int_0^1 r_\epsilon(t)\mathrm{d}t\right) - \frac{\beta}{2}\left[\delta + \mathbb{E}_{\nu_B}[X] \int_0^1 r_\epsilon(t)\mathrm{d}t\right] + \alpha\Psi_{\text{out}}\left(\int_0^1 q_\epsilon(t)\mathrm{d}t\right) + \mathcal{o}_n(1).$$

266 which is what we wanted to show. $\qquad\square$

## D.2 Free entropy variation

268 Lemma D.3 gives a way to compute the free entropy $f_n$ by the fundamental theorem of analysis:

$$f_n = f_{n,\epsilon}(0) + \frac{\beta\delta}{2} + \mathcal{o}_n(1) = \frac{\beta\delta}{2} + f_{n,\epsilon}(1) - \int_0^1 f'_{n,\epsilon}(t)\mathrm{d}t. \tag{52}$$

We define the *overlap* $Q$ and the *overlap matrix* $Q^{(M)}$ as

$$\begin{cases} Q \equiv \dfrac{1}{p}(\mathbf{Z}^\star)^\mathsf{T}\mathbf{z}, & Q^{(M)} \equiv Q & \text{if } \beta = 1, \quad (53a) \\[2ex] Q \equiv \dfrac{1}{p}(\mathbf{Z}^\star)^\dagger\mathbf{z}, & Q^{(M)} \equiv \dfrac{1}{p}\begin{pmatrix} \text{Re}[\mathbf{Z}^\star]^\mathsf{T}\text{Re}[\mathbf{z}] & \text{Re}[\mathbf{Z}^\star]^\mathsf{T}\text{Im}[\mathbf{z}] \\ \text{Im}[\mathbf{Z}^\star]^\mathsf{T}\text{Re}[\mathbf{z}] & \text{Im}[\mathbf{Z}^\star]^\mathsf{T}\text{Im}[\mathbf{z}] \end{pmatrix} & \text{if } \beta = 2. \quad (53b) \end{cases}$$

269 Note that $Q \in \mathbb{K}$, $Q^{(M)} \in \mathcal{S}_\beta(\mathbb{R})$ for $\beta = 1, 2$, and that $\text{Re}[Q] = \text{Tr}_\beta[Q^{(M)}]$. Finally, the Gibbs
270 bracket $\langle\cdot\rangle_{n,t,\epsilon}$ is defined as the average over the posterior distribution of eq. (49). Recall that
271 $u_y(z) \equiv \ln P_{\text{out}}(y|z)$. We can now state our identity for $f'_{n,\epsilon}(t)$, a counterpart to Proposition 3 of [9]
272 and Proposition 5.2 of [11]:

273 **Lemma D.4** (Free entropy variation). *For all $t \in (0, 1)$ and $\epsilon \in \mathcal{D}_n^\beta \times [s_n, 2s_n]$:*

$$f'_{n,\epsilon}(t) = -\frac{1}{2\beta}\mathbb{E}\left\langle\left(\frac{1}{n}\sum_{\mu=1}^m u'_{Y_{t,\mu}}(S_{t,\mu})^\dagger u'_{Y_{t,\mu}}(s_{t,\mu}) - \beta^2\delta r_\epsilon(t)\right)\cdot\left(Q - q_\epsilon(t)\right)\right\rangle_{n,t,\epsilon}$$

$$+ \frac{\beta\delta r_\epsilon(t)}{2}(q_\epsilon(t) - Q_z) + \mathcal{o}_n(1),$$

274 *in which $\mathcal{o}_n(1)$ is uniform in $t, \epsilon, q_\epsilon, r_\epsilon$.*

275 *Proof of Lemma D.4.* The proof is done in two steps. First, we show the following:

$$f'_{n,\epsilon}(t) = -\frac{\beta\delta r_\epsilon(t)}{2}(Q_z - q_\epsilon(t)]) + \frac{1}{2n\beta}\sum_{\mu=1}^m \mathbb{E}\left[\left(Q_z - \frac{\|\mathbf{Z}^\star\|^2}{p}\right)\frac{\Delta P_{\text{out}}(Y_{t,\mu}|S_{t,\mu})}{P_{\text{out}}(Y_{t,\mu}|S_{t,\mu})}\ln\mathcal{Z}\right] \tag{54}$$

$$+ \frac{1}{2\beta}\mathbb{E}\left\langle\left(\frac{1}{n}\sum_{\mu=1}^m u'_{Y_{t,\mu}}(S_{t,\mu})^\dagger u'_{Y_{t,\mu}}(s_{t,\mu}) - \beta^2\delta r_\epsilon(t)\right)\cdot\left(q_\epsilon(t) - Q\right)\right\rangle_{n,t,\epsilon}.$$

276 We will then build on this result by using the concentration of the free entropy of the interpolated
277 model, cf. Theorem D.5 (which is independent of Lemma D.4). From the definition of $f_{n,\epsilon}(t)$, we
278 have (denoting $\mathcal{Z} \equiv \mathcal{Z}_{n,t,\epsilon}(\mathbf{Y}_t, \tilde{\mathbf{Y}}_t)$ to lighten the notations):

$$f'_{n,\epsilon}(t) = -\frac{1}{n}\mathbb{E}[\partial_t\mathcal{H}_{t,\epsilon}(\mathbf{Z}^\star, \mathbf{A}^\star; \mathbf{Y}_t, \tilde{\mathbf{Y}}_t, \mathbf{W}, \mathbf{V})\ln\mathcal{Z}] - \frac{1}{n}\mathbb{E}\langle\partial_t\mathcal{H}_{t,\epsilon}(\mathbf{z}, \mathbf{a}; \mathbf{Y}_t, \tilde{\mathbf{Y}}_t, \mathbf{W}, \mathbf{V})\rangle_{n,t,\epsilon}. \tag{55}$$

279 The definition of $\mathcal{H}$ in eq. (50) gives, up to $\mathcal{o}_n(1)$ terms[1]:

$$\partial_t\mathcal{H}_{t,\epsilon}(\mathbf{Z}^\star, \mathbf{A}^\star; \mathbf{Y}_t, \tilde{\mathbf{Y}}_t, \mathbf{W}, \mathbf{V}) = -\frac{\beta r_\epsilon(t)}{2\sqrt{\int_0^t r_\epsilon(u)\mathrm{d}u}}\sum_{k=1}^p Z_k^\star \cdot \zeta_k + \sum_{\mu=1}^m \partial_t S_{t,\mu} \cdot u'_{Y_{t,\mu}}(S_{t,\mu}). \tag{56}$$

By Proposition F.1 (the Nishimori identity), we have:

$$\mathbb{E}\langle \partial_t \mathcal{H}_{t,\epsilon}(\mathbf{z},\mathbf{a};\mathbf{Y}_t,\tilde{\mathbf{Y}}_t,\mathbf{W},\mathbf{V})\rangle_{n,t,\epsilon} = \mathbb{E}[\partial_t \mathcal{H}_{t,\epsilon}(\mathbf{Z}^\star,\mathbf{A}^\star;\mathbf{Y}_t,\tilde{\mathbf{Y}}_t,\mathbf{W},\mathbf{V})],$$

$$= \mathbb{E}\Big[ \sum_{\mu=1}^{m} \partial_t S_{t,\mu} \cdot \frac{P'_{\mathrm{out}}(Y_{t,\mu}|S_{t,\mu})}{P_{\mathrm{out}}(Y_{t,\mu}|S_{t,\mu})} \Big] + \mathcal{O}_n(1) = \mathcal{O}_n(1),$$

as can be seen from eq. (56). The first term of eq. (55) can be written (up to $\mathcal{O}_n(1)$ terms) as the sum of four contributions that we will compute successively, using Stein's lemma (see eqs. (87),(88)). We start with the first one:

$$\frac{\beta r_\epsilon(t)}{2n\sqrt{\int_0^t r_\epsilon(u)\mathrm{d}u}} \sum_{k=1}^{p} \mathbb{E}[Z_k^\star \cdot \zeta_k \ln \mathcal{Z}] = \frac{r_\epsilon(t)}{2n\sqrt{\int_0^t r_\epsilon(u)\mathrm{d}u}} \sum_{k=1}^{p} \mathbb{E}\Big[Z_k^\star \cdot \frac{\mathrm{d}}{\mathrm{d}\zeta_k} \ln \mathcal{Z}\Big],$$

$$= \frac{-\beta r_\epsilon(t)}{2n\sqrt{\int_0^t r_\epsilon(u)\mathrm{d}u}} \sum_{k=1}^{p} \mathbb{E}[Z_k^\star \cdot \langle R_1(t,\epsilon)^{1/2} \star (Z_k^\star - z_k) + \zeta_k\rangle_{n,t,\epsilon}],$$

$$= \frac{-\beta r_\epsilon(t)}{2n} \sum_{k=1}^{p} \mathbb{E}[|Z_k^\star|^2 - Z_k^\star \cdot \langle z_k\rangle_{n,t,\epsilon}] + \mathcal{O}_n(1)$$

$$= \frac{-\beta \delta r_\epsilon(t)}{2}(Q_z - \mathbb{E}[\langle Q\rangle_{n,t,\epsilon}]) + \mathcal{O}_n(1). \tag{57}$$

We used the Nishimori identity Proposition F.1 in the last equation. We now turn to the second term, and in a similar way we reach, by integration by parts with respect to $\mathbf{W}$ (recall the definition of the Laplace operator in eq. (85)):

$$\frac{1}{\sqrt{p(1-t)}} \sum_{\mu=1}^{m} \mathbb{E}\Big[[\mathbf{W}\mathbf{Z}^\star]_\mu \cdot u'_{Y_{t,\mu}}(S_{t,\mu}) \ln \mathcal{Z}\Big]$$

$$= \frac{1}{\beta} \sum_{\mu=1}^{m} \mathbb{E}\Big[\frac{\|\mathbf{Z}^\star\|^2}{p}(\Delta u_{Y_{t,\mu}}(S_{t,\mu}) + |u'_{Y_{t,\mu}}(S_{t,\mu})|^2) \ln \mathcal{Z}$$

$$+ \Big\langle \Big[(u'_{Y_{t,\mu}}(S_{t,\mu}))^\dagger u'_{Y_{t,\mu}}(s_{t,\mu})\Big] \cdot \Big[\frac{(\mathbf{Z}^\star)^\dagger \mathbf{z}}{p}\Big]\Big\rangle_{n,t,\epsilon}\Big],$$

$$= \frac{1}{\beta} \sum_{\mu=1}^{m} \mathbb{E}\Big[\frac{\|\mathbf{Z}^\star\|^2}{p}\frac{\Delta P_{\mathrm{out}}(Y_{t,\mu}|S_{t,\mu})}{P_{\mathrm{out}}(Y_{t,\mu}|S_{t,\mu})}\ln \mathcal{Z} + \Big\langle[(u'_{Y_{t,\mu}}(S_{t,\mu}))^\dagger u'_{Y_{t,\mu}}(s_{t,\mu})]\cdot\Big[\frac{(\mathbf{Z}^\star)^\dagger\mathbf{z}}{p}\Big]\Big\rangle_{n,t,\epsilon}\Big].$$

We used in the last equation that $\Delta u_y(x) + |u'_y(x)|^2 = \Delta P_{\mathrm{out}}(y|x)/P_{\mathrm{out}}(y|x)$. Integrating by parts with respect to $V_\mu, A_\mu^\star$, we obtain in a similar way:

$$\mathbb{E}\sum_{\mu=1}^{m}\Big[\frac{q_\epsilon(t)V_\mu}{\sqrt{R_2(t,\epsilon)}} + \frac{(Q_z - q_\epsilon(t))A_\mu^\star}{\sqrt{Q_z t - R_2(t,\epsilon) + 2s_n}}\Big]\cdot u'_{Y_{t,\mu}}(S_{t,\mu})\ln\mathcal{Z}$$

$$= \frac{1}{\beta}\sum_{\mu=1}^{m}\mathbb{E}\Big[Q_z\frac{\Delta P_{\mathrm{out}}(Y_{t,\mu}|S_{t,\mu})}{P_{\mathrm{out}}(Y_{t,\mu}|S_{t,\mu})}\ln\mathcal{Z} + q_\epsilon(t)\langle u'_{Y_{t,\mu}}(S_{t,\mu})\cdot u'_{Y_{t,\mu}}(s_{t,\mu})\rangle_{n,t,\epsilon}\Big].$$

By using the Nishimori identity, we obtain after summing all the previous terms the sought eq. (54):

$$f'_{n,\epsilon}(t) = -\frac{\beta\delta r_\epsilon(t)}{2}(Q_z - q_\epsilon(t)) + \frac{1}{2n\beta}\sum_{\mu=1}^{m}\mathbb{E}\Big[\Big(Q_z - \frac{\|\mathbf{Z}^\star\|^2}{p}\Big)\frac{\Delta P_{\mathrm{out}}(Y_{t,\mu}|S_{t,\mu})}{P_{\mathrm{out}}(Y_{t,\mu}|S_{t,\mu})}\ln\mathcal{Z}\Big]$$

$$+ \frac{1}{2\beta}\mathbb{E}\Big\langle\Big(\frac{1}{n}\sum_{\mu=1}^{m}u'_{Y_{t,\mu}}(S_{t,\mu})^\dagger u'_{Y_{t,\mu}}(s_{t,\mu}) - \beta^2\delta r_\epsilon(t)\Big)\cdot(q_\epsilon(t) - Q)\Big\rangle_{n,t,\epsilon}.$$

To finish the proof, we must therefore just show that $\lim_{n\to\infty} B_n = 0$ uniformly in $t, \epsilon, q_\epsilon, r_\epsilon$, with

$$B_n \equiv \frac{1}{n}\sum_{\mu=1}^{m}\mathbb{E}\Big[\Big(Q_z - \frac{\|\mathbf{Z}^\star\|^2}{p}\Big)\frac{\Delta P_{\mathrm{out}}(Y_{t,\mu}|S_{t,\mu})}{P_{\mathrm{out}}(Y_{t,\mu}|S_{t,\mu})}\ln\mathcal{Z}\Big].$$

First, note that

$$\mathbb{E}\Big[\Big(Q_z - \frac{\|\mathbf{Z}^\star\|^2}{p}\Big)\frac{\Delta P_{\mathrm{out}}(Y_{t,\mu}|S_{t,\mu})}{P_{\mathrm{out}}(Y_{t,\mu}|S_{t,\mu})}\Big] = \mathbb{E}\Big[\Big(Q_z - \frac{\|\mathbf{Z}^\star\|^2}{p}\Big)\mathbb{E}\Big[\frac{\Delta P_{\mathrm{out}}(Y_{t,\mu}|S_{t,\mu})}{P_{\mathrm{out}}(Y_{t,\mu}|S_{t,\mu})}\Big|\mathbf{Z}^\star, \mathbf{S}_t\Big]\Big] = 0,$$

since $\int \mathrm{d}Y \nabla P_{\mathrm{out}}(Y|S) = 0$. Using this, we can write

$$B_n = \frac{1}{n}\sum_{\mu=1}^{m} \mathbb{E}\Big[\Big(Q_z - \frac{\|\mathbf{Z}^\star\|^2}{p}\Big)\frac{\Delta P_{\mathrm{out}}(Y_{t,\mu}|S_{t,\mu})}{P_{\mathrm{out}}(Y_{t,\mu}|S_{t,\mu})}(\ln \mathcal{Z} - f_{n,\epsilon}(t))\Big]. \tag{58}$$

We then follow exactly the lines of Appendix A.5.2 of [9], let us recall its main steps. Starting from eq. (58), one uses the Cauchy-Schwarz inequality alongside Theorem D.5 (which is independent of Lemma D.4), that gives $\mathbb{E}[(\ln \mathcal{Z}/n - f_{n,\epsilon}(t))^2] \to 0$ uniformly in $t$. The expectation of the square of the other terms in eq. (58) can easily be bounded using hypotheses $(H0),(h1^\star),(h3)$, uniformly in $t$. Combining these bounds then shows that $B_n \to 0$ uniformly in $t$, which finishes the proof. $\qquad\square$

## D.3 Concentration of the free entropy and the overlap

We denote the mean over $\epsilon$ as:

$$\mathbb{E}_\epsilon[\cdot] \equiv \frac{1}{s_n \mathrm{Vol}(\mathcal{D}_n^\beta)}\int_{\mathcal{D}_n^\beta}\mathrm{d}\epsilon_1 \int_0^1 \mathrm{d}\epsilon_2[\cdot].$$

In [9, 11, 12], the authors give a quite technical proof of the concentration of the free entropy and the overlap of an interpolated system close to the one described in Section D.1. We present here two results of this type. The first one concerns the concentration of the free entropy of the interpolated system[1]. It is very similar to Theorem 6 of [9].

**Theorem D.5** (Free entropy concentration). *Under the assumptions of Theorem 2.2, there exists a constant $C > 0$ that does not depend on $n, t, \epsilon, q_\epsilon, r_\epsilon$ such that for all $n, t, \epsilon, q_\epsilon, r_\epsilon$:*

$$\mathbb{E}\Big[\Big(\frac{1}{n}\ln \mathcal{Z}_{n,t,\epsilon}(\boldsymbol{Y}_t, \tilde{\boldsymbol{Y}}_t) - \frac{1}{n}\mathbb{E}\ln \mathcal{Z}_{n,t,\epsilon}(\boldsymbol{Y}_t, \tilde{\boldsymbol{Y}}_t)\Big)^2\Big] \leq \frac{C}{n}.$$

Our second theorem concerns the concentration of the overlap. It will follow as an almost immediate consequence of a result of [12]. Before stating it, we introduce a regularity notion for our interpolation functions of eq. (45):

**Definition D.6** (Regularity). *The families of functions $(q_\epsilon), (r_\epsilon)$ for $\epsilon \in \mathcal{D}_n^\beta \times [s_n, 2s_n]$ are said to be regular if there exists $\gamma > 0$ such that for all $t \in [0, 1]$ the mapping $\epsilon \mapsto R(t, \epsilon) \equiv (R_1(t, \epsilon), R_2(t, \epsilon))$ is a $\mathcal{C}^1$ diffeomorphism whose Jacobian $J_{n,\epsilon}(t)$ satisfies $J_{n,\epsilon}(t) \geq \gamma$ for all $t \in [0, 1]$ and all $\epsilon$.*

We can now state our theorem on the concentration of the overlap $Q$:

**Theorem D.7** (Overlap concentration). *Under $(H0),(h1^\star),(h2),(h3)$, and if the functions $(q_\epsilon, r_\epsilon)$ are regular (cf. Definition D.6), then there exists a sequence $s_n$ going to $0$ (arbitrarily slowly) such that*

$$\mathbb{E}_\epsilon \int_0^1 \mathrm{d}t\, \mathbb{E}\langle|Q - \mathbb{E}\langle Q\rangle_{n,t,\epsilon}|^2\rangle_{n,t,\epsilon} = \mathcal{O}_n(1),$$

*with $\mathcal{O}_n(1)$ uniform in the choice of $r_\epsilon, q_\epsilon$.*

The rest of this section is dedicated to the proofs of Theorem D.5 and Theorem D.7.

### D.3.1 Proof of Theorem D.5

The proof described in Section E.1 of [9] can be adapted verbatim in this setting. It relies on two concentration inequalities [13], that we recall here in the complex and real settings.

**Proposition D.8** (Gaussian Poincaré inequality). *Let $\boldsymbol{U} \in \mathbb{K}^n$ be distributed according to $\mathcal{N}_\beta(0, \mathbb{1}_n)$, and $g : \mathbb{K}^n \to \mathbb{R}$ a $\mathcal{C}^1$ function. Recall our conventions for derivatives, see Section F.2. Then*

$$\mathbb{E}[g(\boldsymbol{U})^2] - \mathbb{E}[g(\boldsymbol{U})]^2 \leq \frac{1}{\beta}\mathbb{E}[\|\nabla g(\boldsymbol{U})\|^2].$$

**Proposition D.9** (Bounded differences inequality). *Let $\mathcal{B} \subset \mathbb{K}$, and $g : \mathcal{B}^n \to \mathbb{R}$ a function such that there exists $c_1, \cdots, c_n \geq 0$ that satisfy for all $i \in \{1, \cdots, n\}$:*

$$\sup_{\substack{u_1, \cdots, u_n \in \mathcal{B}^n \\ u_i' \in \mathcal{B}}} |g(u_1, \cdots, u_i, \cdots, u_n) - g(u_1, \cdots, u_{i-1}, u_i', u_{i+1}, \cdots, u_n)| \leq c_i.$$

*Then if $\boldsymbol{U} \in \mathbb{K}^n$ is a random vector of independent random variables with value in $\mathcal{B}$, we have:*

$$\mathbb{E}[g(\boldsymbol{U})^2] - \mathbb{E}[g(\boldsymbol{U})]^2 \leq \frac{\beta}{4} \sum_{i=1}^n c_i^2.$$

Proposition D.8 is used to show the concentration of $(\ln \mathcal{Z}_{n,t,\epsilon})/n$ with respect to the Gaussian variables $\zeta, \mathbf{W}, \mathbf{A}^\star, \mathbf{V}$, while Proposition D.9 is used to show the concentration with respect to $\mathbf{Z}^\star$. Using this strategy, the proof of [9] is directly transposed here, and we do not repeat it.

### D.3.2 Proof of Theorem D.7

We start with a lemma on the average value of $Q^{(M)}$ under $\mathbb{E}\langle \cdot \rangle$, in the complex case.

**Lemma D.10.** *Assume $\beta = 2$. Then*

$$\begin{cases} \mathbb{E}\langle Q_{12}^{(M)} \rangle_{n,t,\epsilon} = \mathbb{E}\langle Q_{21}^{(M)} \rangle_{n,t,\epsilon} = \mathcal{O}_n(1), \\ \mathbb{E}\langle Q_{11}^{(M)} \rangle_{n,t,\epsilon} - \mathbb{E}\langle Q_{22}^{(M)} \rangle_{n,t,\epsilon} = \mathcal{O}_n(1), \end{cases}$$

*in which $\mathcal{O}_n(1)$ is uniform in $t, \epsilon, q_\epsilon, r_\epsilon$.*

*Proof of Lemma D.10.* By the classical theorems of continuity and derivability under the integral sign, it is easy to see that $\mathbb{E}\langle Q^{(M)} \rangle_{n,t,\epsilon}$ is a continuous function of $(R_1, R_2)$, and moreover that it admits a Lipschitz constant $K > 0$, independent of $t, \epsilon, q_\epsilon, r_\epsilon$. Indeed, thanks to hypotheses $(H0),(h1^\star),(h2),(h3)$, the domination hypotheses of these theorems are satisfied, and one can easily bound the differential of $\mathbb{E}\langle Q \rangle$ to obtain the existence of the Lipschitz constant $K > 0$. Moreover, for $\epsilon_1 = 0$, $\epsilon_2 = 0$, it is easy to check by the Nishimori identity Proposition F.1 that we have:

$$\begin{cases} \mathbb{E}\langle Q_{12}^{(M)} \rangle_{n,t,\epsilon} = \mathbb{E}\langle Q_{21}^{(M)} \rangle_{n,t,\epsilon} = 0, \\ \mathbb{E}\langle Q_{11}^{(M)} \rangle_{n,t,\epsilon} = \mathbb{E}\langle Q_{22}^{(M)} \rangle_{n,t,\epsilon}. \end{cases}$$

Using the Lipschitz constant $K > 0$ (which does not depend on the parameters $t, \epsilon, q_\epsilon, r_\epsilon$) and the fact that $\epsilon_1, \epsilon_2 = \mathcal{O}(s_n) = \mathcal{O}_n(1)$, this ends the proof. $\square$

Moreover, once averaged over $\epsilon_2 \in [s_n, 2s_n]$ and $t \in (0, 1)$, and using the concentration of the free entropy (Theorem D.5), the results of [12] imply the thermal and total concentration of the overlap matrix $Q^{(M)}$ defined in eq. (53):

**Lemma D.11.** *Assuming that $(q_\epsilon, r_\epsilon)$ are regular, there exists a sequence $s_n \to 0$ (slowly enough) and $\eta, C > 0$ such that (with $\|\cdot\|_F$ the Frobenius norm):*

$$\begin{cases} \mathbb{E}_\epsilon \int_0^1 \mathrm{d}t \, \mathbb{E}\langle \|Q^{(M)} - \langle Q^{(M)} \rangle_{n,t,\epsilon} \|_F^2 \rangle_{n,t,\epsilon} \leq \dfrac{C}{n^\eta}, \\ \mathbb{E}_\epsilon \int_0^1 \mathrm{d}t \, \mathbb{E}\langle \|Q^{(M)} - \mathbb{E}\langle Q^{(M)} \rangle_{n,t,\epsilon} \|_F^2 \rangle_{n,t,\epsilon} \leq \dfrac{C}{n^\eta}. \end{cases}$$

*Proof of Lemma D.11.* We can use the results of [12], under two conditions: $(i)$ the concentration of the free entropy, which is given here by Theorem D.5, and $(ii)$ the regularity of $(q_\epsilon, r_\epsilon)$. Indeed, the results of [9] give the concentration results as integrated over the matrix $R_1(t, \epsilon)$. Using the regularity assumption, we can lower bound these integrals by integrals over the perturbation matrix $\epsilon_1$ (up to a multiplicative constant, which is uniform in all the relevant parameters), which then yields Lemma D.11. This argument was also made in a very close setting in [9, 11]. $\square$

Using Lemma D.10 (if $\beta = 1$ this lemma is not needed) alongside Lemma D.11 yields Theorem D.7, since $Q = \mathrm{Tr}_\beta[Q^{(M)}]$.

## D.4 Upper and lower bounds

**Proposition D.12** (Fundamental sum rule). *Assume that $(q_\epsilon, r_\epsilon)$ are regular (cf Definition D.6), and that for all $\epsilon \in \mathcal{D}_n^\beta \times [s_n, 2s_n]$ and $t \in (0,1)$ we have $q_\epsilon(t) = \mathrm{Tr}_\beta[\mathbb{E}\langle Q^{(M)}\rangle_{n,t,\epsilon}]$. Then:*

$$f_n = \mathbb{E}_\epsilon\Big[\Psi_0^{(\nu)}\Big(\int_0^1 r_\epsilon(t)\mathrm{d}t\Big) + \alpha\Psi_{\mathrm{out}}\Big(\int_0^1 q_\epsilon(t)\mathrm{d}t\Big) - \frac{\beta\delta}{2}\int_0^1 q_\epsilon(t)r_\epsilon(t)\mathrm{d}t\Big] + o_n(1),$$

*in which $o_n(1)$ is uniform in the choice of $q_\epsilon, r_\epsilon$.*

*Proof of Proposition D.12.* The proof is based on Lemma D.3 and Lemma D.4. Replacing their results into eq (52), in order to finish the proof, we only need to show that $\lim_{n\to\infty}\Gamma_n = 0$ (uniformly in $r_\epsilon, q_\epsilon$), with

$$\Gamma_n \equiv \Big(\mathbb{E}_\epsilon\int_0^1 \mathrm{d}t\, \mathbb{E}\Big\langle\Big(\frac{1}{n}\sum_{\mu=1}^m u'_{Y_{t,\mu}}(S_{t,\mu})^\dagger u'_{Y_{t,\mu}}(s_{t,\mu}) - \beta^2\delta r_\epsilon(t)\Big)\cdot\Big(q_\epsilon(t)-Q\Big)\Big\rangle_{n,t,\epsilon}\Big)^2.$$

By the Cauchy-Schwarz inequality, we can bound:

$$\Gamma_n \leq \mathbb{E}_\epsilon\int_0^1 \mathrm{d}t\, \mathbb{E}\Big\langle\Big|\frac{1}{n}\sum_{\mu=1}^m u'_{Y_{t,\mu}}(S_{t,\mu})^\dagger u'_{Y_{t,\mu}}(s_{t,\mu}) - \beta^2\delta r_\epsilon(t)\Big|^2\Big\rangle_{n,t,\epsilon}$$
$$\times \mathbb{E}_\epsilon\int_0^1 \mathrm{d}t\, \mathbb{E}\langle|Q - q_\epsilon(t)|^2\rangle_{n,t,\epsilon}.$$

The first term is bounded by a constant $C > 0$ by Lemma F.2 (recall that $r_\epsilon(t)$ is bounded as well by $r_{\max}$). By Theorem D.7, the second term is $o_n(1)$, uniformly in $q_\epsilon, r_\epsilon$, since we assumed that $q_\epsilon(t) = \mathrm{Tr}_\beta[\mathbb{E}\langle Q\rangle]$. As the vanishing terms are uniform in $q_\epsilon, r_\epsilon$, this shows that $\lim_{n\to\infty}\Gamma_n = 0$, which ends the proof. $\square$

Before obtaining the two bounds from the fundamental sum rule, we need a final preparatory lemma, that will imply the regularity of the functions $(q_\epsilon, r_\epsilon)$ that we will chose to derive the bounds.

**Lemma D.13** (Regularity). *We define $F_n(t, R(t,\epsilon)) = (F_n^{(1)}(t, R(t,\epsilon)), F_n^{(2)}(t, R(t,\epsilon)))$, with:*

$$\begin{cases} F_n^{(1)}(t, R(t,\epsilon)) \equiv \Big(\frac{2\alpha}{\beta\delta}\Psi'_{\mathrm{out}}(\mathrm{Tr}_\beta[\mathbb{E}\langle Q^{(M)}\rangle_{n,t,\epsilon}])\Big)\mathbb{1}_\beta, \\ F_n^{(2)}(t, R(t,\epsilon)) \equiv \mathrm{Tr}_\beta[\mathbb{E}\langle Q^{(M)}\rangle_{n,t,\epsilon}]. \end{cases}$$

*Then $F_n$ is a continuous function from its domain to $\mathbb{R}^2$. Moreover, it admits partial derivatives with respect to both $R_1$ and $R_2$ on the interior of its domain. We have, uniformly over the choice of $(q_\epsilon, r_\epsilon)$:*

$$\begin{cases} \liminf_{n\to\infty}\inf_{t\in(0,1)}\inf_{\substack{\epsilon_1\in\mathcal{D}_n\\\epsilon_2\in[s_n,2s_n]}}\sum_{l=1}^\beta \frac{\partial(F_n^{(1)})_{ll}}{\partial(R_1)_{ll}}(t, R(t,\epsilon)) \geq 0, \\ \frac{\partial F_n^{(2)}}{\partial R_2}(t, R(t,\epsilon)) \geq 0. \end{cases}$$

*Proof of Lemma D.13.* The proof is very close to the arguments of Lemma 5.5 of [11]. The continuity and derivability follow from standard theorems of continuity and derivation under the integral sign, thanks to hypotheses $(H0),(h1^\star),(h3)$. Indeed, under these boundedness assumptions, the domination hypotheses of these theorems are straightforwardly satisfied. Let us start with the first inequality. We can easily write:

$$\sum_{l=1}^\beta \frac{\partial(F_n^{(1)})_{ll}}{\partial(R_1)_{ll}} = \frac{2\alpha}{\beta\delta}\Psi''_{\mathrm{out}}(\mathrm{Tr}_\beta[\mathbb{E}\langle Q^{(M)}\rangle])\sum_{l=1}^\beta \frac{\partial\mathrm{Tr}_\beta\mathbb{E}\langle Q^{(M)}\rangle}{\partial(R_1)_{ll}}.$$

The convexity of $\Psi_{\text{out}}$ was already derived so that $\Psi''_{\text{out}} \geq 0$. Moreover, since $R_1$ is the SNR matrix of a linear channel, we know that the matrix $\nabla_{R_1}\mathbb{E}\langle Q^{(M)}\rangle$ is positive [11]. In particular, its trace is always positive, and by Lemma D.10:

$$\sum_{l=1}^{\beta} \frac{\partial \text{Tr}_\beta \mathbb{E}\langle Q^{(M)}\rangle}{\partial (R_1)_{ll}} = \underbrace{\text{Tr}_\beta[\nabla_{R_1}\mathbb{E}\langle Q^{(M)}\rangle]}_{\geq 0} + \mathcal{O}_n(1),$$

with a $\mathcal{O}_n(1)$ uniform in $t, \epsilon, r_\epsilon, q_\epsilon$. This shows the first inequality. Let us sketch the argument for the second inequality. The trace of $Q^{(M)}$ is directly related to the MMSE on the complex vector $\mathbf{Z}^\star$ by:

$$\frac{1}{p}\text{MMSE}(\mathbf{Z}^\star|\mathbf{Y}_t, \tilde{\mathbf{Y}}_t, \mathbf{V}, \mathbf{W}) = \frac{1}{p}\mathbb{E}[\|\mathbf{Z}^\star - \langle \mathbf{z}\rangle\|^2] = Q_z - \text{Tr}_\beta[\mathbb{E}\langle Q^{(M)}\rangle].$$

The fact that the MMSE should decrease as the SNR $R_2$ increases, for a channel of the type of eq. (46a), is very natural, and it was proven in Proposition 6 of [9], which applies here. This proposition yields that $\text{Tr}_\beta[\mathbb{E}\langle Q^{(M)}\rangle]$ is a nondecreasing function of $R_2$, which ends the proof. $\square$

Finally, we define the *replica-symmetric potential*, that appears in Proposition D.1:

$$f_{\text{RS}}(q,r) \equiv -\frac{\beta\delta rq}{2} + \Psi_0^{(\nu)}(r) + \alpha\Psi_{\text{out}}(q).$$

### D.4.1 Lower bound

**Proposition D.14** (Lower bound). *Under the assumptions of Theorem 2.2, the free entropy $f_n$ satisfies:*

$$\liminf_{n\to\infty} f_n \geq \sup_{r\geq 0} \inf_{q\in[0,Q_z]} f_{\text{RS}}(q,r).$$

*Proof of Proposition D.14.* We fix $r \geq 0$ and $R_1(t) = \epsilon_1 + rt\mathbb{1}_\beta$. We then choose $R_2(t)$ as the unique solution to the ordinary differential equation:

$$R_2'(t) = \text{Tr}_\beta[\mathbb{E}\langle Q^{(M)}\rangle_{n,t,\epsilon}], \tag{64}$$

with boundary condition $R_2(0) = \epsilon_2$. We denote this unique solution as $R_2(t) = \epsilon_2 + \int_0^t q_\epsilon(r;v)\mathrm{d}v$. The ODE of eq. (64) can easily be seen to satisfy the hypotheses of the parametric Cauchy-Lipschitz theorem (as a function of the initial condition $\epsilon_2$), and by the Liouville formula (cf Lemma A.3 of [11]), the Jacobian $J_{n,\epsilon}(t)$ of $\epsilon \mapsto R(t,\epsilon) \equiv (R_1(t,\epsilon), R_2(t,\epsilon))$ verifies:

$$J_{n,\epsilon}(t) = \exp\Big(\int_0^t \frac{\partial\text{Tr}_\beta[\mathbb{E}\langle Q\rangle_{n,u,\epsilon}]}{\partial R_2}(u, R(u,\epsilon))\mathrm{d}u\Big) \geq 1,$$

in which the inequality is a consequence of Lemma D.13. The functions are thus regular in the sens of Definition D.6, and moreover the local inversion theorem implies that $\epsilon \mapsto R(t,\epsilon)$ is a $\mathcal{C}^1$ diffeomorphism. We can therefore use the fundamental sum rule Proposition D.12 as all its hypotheses are verified. We reach:

$$f_n = \mathbb{E}_\epsilon\Big[\Psi_0^{(\nu)}(r) + \alpha\Psi_{\text{out}}\Big(\int_0^1 q_\epsilon(r;t)\mathrm{d}t\Big) - \frac{\beta\delta r}{2}\int_0^1 q_\epsilon(r;t)\mathrm{d}t\Big] + \mathcal{O}_n(1),$$

$$= \mathbb{E}_\epsilon\Big[f_{\text{RS}}\Big(\int_0^1 q_\epsilon(r;t)\mathrm{d}t, r\Big)\Big] + \mathcal{O}_n(1),$$

$$\geq \inf_{q\in[0,Q_z]} f_{\text{RS}}(q,r) + \mathcal{O}_n(1).$$

Since this is true for all $r \geq 0$ we easily obtain the sought lower bound. $\square$

### D.4.2 Upper bound

We now prove the final upper bound, which will end the proof of Lemma D.2.

**Proposition D.15** (Upper bound). *Under the assumptions of Theorem 2.2, the free entropy $f_n$ satisfies:*

$$\limsup_{n\to\infty} f_n \leq \sup_{r\geq 0} \inf_{q\in[0,Q_z]} f_{\mathrm{RS}}(q,r).$$

*Proof of Proposition D.15.* We will choose $R(t,\epsilon) = (R_1(t,\epsilon), R_2(t,\epsilon))$ as the solution to the ordinary differential equation:

$$\partial_t R_1(t,\epsilon) = \frac{2\alpha}{\beta\delta}\Psi_{\mathrm{out}}\Big[\mathrm{Tr}_\beta[\mathbb{E}\langle Q^{(M)}\rangle_{n,t,\epsilon}]\Big]\mathbb{1}_\beta, \qquad \partial_t R_2(t,\epsilon) = \mathrm{Tr}_\beta[\mathbb{E}\langle Q^{(M)}\rangle_{n,t,\epsilon}], \qquad (65)$$

with initial conditions $R(0,\epsilon) = (\epsilon_1, \epsilon_2)$. Let us denote this equation as $\partial_t R(t) = (F_{n,1}(t, R(t)), F_{n,2}(t, R(t)))$. As in Section D.4.1, the parametric Cauchy-Lipschitz theorem implies the existence, unicity and $\mathcal{C}^1$ regularity of $R(t,\epsilon)$ as a function of $(t,\epsilon)$. We denote this unique solution[1] as $R_1(t,\epsilon) = \epsilon_1 + (\int_0^t r_\epsilon(v)\mathrm{d}v)\mathbb{1}_\beta$, $R_2(t,\epsilon) = \epsilon_2 + \int_0^t q_\epsilon(v)\mathrm{d}v$. Again, the Liouville formula yields that the Jacobian $J_{n,\epsilon}(t)$ of the map $\epsilon \mapsto R(t,\epsilon)$ is given by:

$$J_{n,\epsilon}(t) = \exp\Big(\int_0^t \Big\{\sum_{l=1}^\beta \frac{\partial(F_{n,1})_{ll}}{\partial(R_1)_{ll}}(s, R(s,\epsilon)) + \frac{\partial F_{n,2}}{\partial R_2}(s, R(s,\epsilon))\Big\}\mathrm{d}s\Big). \qquad (66)$$

Then, by Lemma D.13, we have that $\liminf_{n\to\infty} \inf_t \inf_\epsilon J_{n,\epsilon}(t) \geq 1$. In particular, this implies that $(q_\epsilon, r_\epsilon)$ are regular in the sense of Definition D.6. We have all that is needed to apply Proposition D.12 and we reach:

$$f_n = \mathbb{E}_\epsilon\Big[\Psi_0^{(\nu)}\Big(\int_0^1 r_\epsilon(t)\mathrm{d}t\Big) + \alpha\Psi_{\mathrm{out}}\Big(\int_0^1 q_\epsilon(t)\mathrm{d}t\Big) - \frac{\beta\delta}{2}\int_0^1 q_\epsilon(t)r_\epsilon(t)\mathrm{d}t\Big] + \mathcal{O}_n(1).$$

Since $\Psi_{\mathrm{out}}$ and $\Psi_0^{(\nu)}$ are convex, Jensen's inequality implies:

$$f_n \leq \mathbb{E}_\epsilon\int_0^1 \mathrm{d}t\Big[\Psi_0^{(\nu)}(r_\epsilon(t)) + \alpha\Psi_{\mathrm{out}}(q_\epsilon(t)) - \frac{\beta\delta}{2}q_\epsilon(t)r_\epsilon(t)\Big] + \mathcal{O}_n(1),$$

$$\leq \mathbb{E}_\epsilon\int_0^1 \mathrm{d}t\, f_{\mathrm{RS}}(q_\epsilon(t), r_\epsilon(t)) + \mathcal{O}_n(1)$$

Note that we have

$$f_{\mathrm{RS}}(q_\epsilon(t), r_\epsilon(t)) = \inf_{q\in[0,Q_z]} f_{\mathrm{RS}}(q, r_\epsilon(t)).$$

Indeed, the function $q \mapsto f_{\mathrm{RS}}(q, r_\epsilon(t))$ is convex, and its derivative is zero for $q = q_\epsilon(t)$ by definition of $(r_\epsilon, q_\epsilon)$, cf eq. (65). Therefore, we have:

$$f_n \leq \mathbb{E}_\epsilon\int_0^1 \mathrm{d}t \inf_{q\in[0,Q_z]} f_{\mathrm{RS}}(q, r_\epsilon(t)) + \mathcal{O}_n(1),$$

$$\leq \sup_{r\geq 0}\inf_{q\in[0,Q_z]} f_{\mathrm{RS}}(q, r_\epsilon(t)) + \mathcal{O}_n(1),$$

which ends the proof. $\qquad\qquad\qquad\qquad\qquad\qquad\qquad\qquad\qquad\qquad\qquad\qquad\qquad\square$

### D.5 Proof of the MMSE limit

As mentioned in the main part of this work, the MMSE statement in Conjecture 2.1 is stated informally. The main reason is that obtaining the MMSE limit generically requires many technicalities, to account for the possible symmetries of the system, see e.g. Theorem 2 of [9] which performs such an analysis.

To simplify the analysis, we "break" this symmetry by adding a side channel with an arbitrarily small signal-to-noise ratio. Formally, we consider the following inference problem made of two channels:

$$
\begin{cases}
Y_{t,\mu} \sim P_{\text{out}}\Big(\cdot \,\Big|\, \dfrac{1}{\sqrt{n}} \sum_{i=1}^{n} \Phi_{\mu i} X_i^{\star}\Big) & \mu = 1, \cdots, m \quad\quad (67\text{a}) \\[3mm]
\tilde{\mathbf{Y}}_t = \sqrt{\Lambda}\mathbf{X}^{\star} + \mathbf{Z}', & \mathbf{Z}' \sim \mathcal{N}_{\beta}(0, \mathbb{1}_n), \quad\quad (67\text{b})
\end{cases}
$$

with $\Lambda > 0$ (arbitrarily small). We can now state our precise statement on the MMSE:

**Proposition D.16.** *Consider the inference problem of eq. (67), under* $(H0),(h1),(h2),(h3)$. *We denote* $\langle\cdot\rangle$ *the average with respect to the posterior distribution of* $\boldsymbol{x}$ *under the problem of eq. (67). The minimum mean squared error is achieved by the Bayes-optimal estimator* $\hat{X}_{\text{opt}} = \langle\boldsymbol{x}\rangle$, *and it satisfies as* $n \to \infty$:

$$
\lim_{n\to\infty} \text{MMSE} = \lim_{n\to\infty} \frac{1}{n}\mathbb{E}\|\boldsymbol{X}^{\star} - \langle\boldsymbol{x}\rangle\|^2 = 1 - q_x^{\star}, \tag{68}
$$

*with* $q_x^{\star}$ *the solution of the extremization problem in eq. (6), taking into account the additional side information of eq. (67b).*

*Proof of Proposition D.16.* With the side channel added, this proposition will follow from an application of the classical I-MMSE theorem [14]. We denote $\langle\cdot\rangle$ the mean under the posterior distribution of $\mathbf{x}$ under the channels of eq. (67), and $\mathbb{E}$ the average with respect to the "quenched" variables $\boldsymbol{\Phi}, \mathbf{Z}', \mathbf{X}^{\star}$. The free entropy $f_n(\Lambda)$ is defined as the average of the log-normalization of the posterior distribution:

$$
f_n(\Lambda) \equiv \frac{1}{n}\mathbb{E}\ln \int_{\mathbb{K}^n} P_0(\mathrm{d}\mathbf{x})\Big[\prod_{\mu=1}^{m} P_{\text{out}}\Big(Y_{t,\mu}\,\Big|\,\frac{1}{\sqrt{n}}\sum_{i=1}^{n}\Phi_{\mu i}x_i\Big)\Big]\frac{e^{-\frac{\beta}{2}\sum_{i=1}^{n}\left|\tilde{Y}_{t,i}-\sqrt{\Lambda}x_i\right|^2}}{(2\pi/\beta)^{n\beta/2}}.
$$

We can easily replicate the adaptive interpolation analysis of Theorem 2.2 (see Section D) to this case, and we reach the following result for the asymptotic free entropy $f(\Lambda)$ of eq. (67):

**Lemma D.17.** *For all* $\Lambda > 0$, *we have* $\lim_{n\to\infty} f_n(\Lambda) = f(\Lambda)$, *given by:*

$$
f(\Lambda) = \sup_{q_x \in [0,1]} \sup_{q_z \in [0,Q_z]} [I_0(q_x, \Lambda) + \alpha I_{\text{out}}(q_z) + I_{\text{int}}(q_x, q_z)], \tag{69}
$$

*with* $I_{\text{out}}, I_{\text{int}}$ *given in Conjecture 2.1, and:*

$$
\begin{aligned}
I_0(q_x, \Lambda) \equiv \inf_{\hat{q}_x \geq 0}\Big[ &-\frac{\beta\hat{q}_x q_x}{2} + \int_{\mathbb{K}^2}\mathcal{D}_{\beta}\xi\,\mathrm{d}\tilde{y}\int P_0(\mathrm{d}x)\frac{e^{-\frac{\beta\hat{q}_x}{2}|x|^2 + \beta\sqrt{\hat{q}_x}x\cdot\xi - \frac{\beta}{2}|\tilde{y}-\sqrt{\Lambda}x|^2}}{(2\pi/\beta)^{\beta/2}} \\
&\ln \int P_0(\mathrm{d}x)\frac{e^{-\frac{\beta\hat{q}_x}{2}|x|^2 + \beta\sqrt{\hat{q}_x}x\cdot\xi - \frac{\beta}{2}|\tilde{y}-\sqrt{\Lambda}x|^2}}{(2\pi/\beta)^{\beta/2}}\Big].
\end{aligned}
$$

*Proof of Lemma D.17.* By Proposition D.1, one can simply replicate the adaptive interpolation analysis of Section D to this model, and this will prove the required formula. The precise form of $I_0(q_x)$ is very easy to compute. $\square$

We can then use the I-MMSE formula [14], that yields that for any $\Lambda$,

$$
\lim_{n\to\infty}\text{MMSE} = -\frac{2}{\beta}\partial_{\Lambda}f(\Lambda). \tag{70}
$$

Moreover, by Lemma D.17, $q_x^{\star}, \hat{q}_x^{\star}$ is a solution of the equation:

$$
q_x^{\star} = \frac{1}{(2\pi/\beta)^{\beta/2}}\int \mathcal{D}_{\beta}\xi\mathrm{d}\tilde{y}\frac{\left|\int P_0(\mathrm{d}x)\,x\,e^{-\frac{\beta\hat{q}_x^{\star}}{2}|x|^2 + \beta\sqrt{\hat{q}_x^{\star}}x\cdot\xi - \frac{\beta}{2}|\tilde{y}-\sqrt{\Lambda}x|^2}\right|^2}{\int P_0(\mathrm{d}x)e^{-\frac{\beta\hat{q}_x^{\star}}{2}|x|^2 + \beta\sqrt{\hat{q}_x^{\star}}x\cdot\xi - \frac{\beta}{2}|\tilde{y}-\sqrt{\Lambda}x|^2}}. \tag{71}
$$

From the expression of $I_0$ in Lemma D.17 and eq. (71), it is then a straightforward calculation to see that $-(2/\beta)\partial_{\Lambda}f(\Lambda) = 1 - q_x^{\star}$, which ends the proof. $\square$

### D.6 Proof of Theorem 2.2: the Gaussian matrix case

In this subsection, we place ourselves under $(H0),(h'1)$ and sketch how the proof performed in the previous sections directly extends under these hypotheses. Note that here $\langle\lambda\rangle_\nu = \alpha$, so $Q_z = Q_x = \rho$. First, we can state a very similar result to Proposition D.1, simplifying Conjecture 2.1 in this setting:

**Proposition D.18.** *Under $(H0),(h'1)$, the replica conjecture 2.1 reduces to:*

$$\lim_{n\to\infty}\frac{1}{n}\mathbb{E}\ln\mathcal{Z}_n(\boldsymbol{Y}) = \sup_{\hat{q}\geq 0}\inf_{q\in[0,\rho]}\Big[-\frac{\beta q\hat{q}}{2} + \Psi_{P_0}(\hat{q}) + \alpha\Psi_{\mathrm{out}}(q)\Big].$$

*with $q_z,\Psi_{\mathrm{out}}$ defined in Proposition D.1, and $\Psi_{P_0}(\hat{q})$ defined for $\hat{q}\geq 0$ by:*

$$\Psi_{P_0}(\hat{q}) \equiv \mathbb{E}_\xi\mathcal{Z}_0(\sqrt{\hat{q}}\xi,\hat{q})\ln\mathcal{Z}_0(\sqrt{\hat{q}}\xi,\hat{q}),$$

*with $\mathcal{Z}_0$ defined in eq. (7).*

*Proof of Proposition D.18.* The proof follows similar lines to the proof of Proposition D.1, see Section E. Let us briefly sketch the main steps. Since $\boldsymbol{\Phi}$ is Gaussian, $\nu$ is the Marchenko-Pastur distribution [15], and one can easily simplify $I_{\mathrm{int}}(q_x,q_z)$ as:

$$I_{\mathrm{int}}(q_x,q_z) = -\frac{\alpha\beta}{2}\Big[\frac{q_x(\rho - q_z)}{2\rho(\rho - q_x)} + \ln(\rho - q_x)\Big].$$

Using then the exact same sup-inf inversion arguments as in Section E, the supremum and infimum over $q_z$ and $\hat{q}_z$ are solved by:

$$\begin{cases} q_z = q_x + \dfrac{2}{\beta}(\rho - q_x)^2\Psi'_{\mathrm{out}}(q_x), & \text{(72a)} \\[2mm] \hat{q}_z = \dfrac{q_x}{\rho(\rho - q_x)} & \text{(72b)} \end{cases}$$

And finally, we reach that (with the notations of Conjecture 2.1) $\alpha I_{\mathrm{out}}(q_z)+I_{\mathrm{int}}(q_x,q_z) = \alpha\Psi_{\mathrm{out}}(q_x)$. Posing $q = q_x, \hat{q} = \hat{q}_x$ finishes the proof. $\qquad\square$

We turn now to proving the formula of Proposition D.18. The proof goes exactly as in the previous sections of Section D, by considering instead of eq. (46) the interpolation problem:

$$\begin{cases} \Big\{Y_{t,\mu}\sim P_{\mathrm{out}}\Big(\cdot\Big|\sqrt{\dfrac{1-t}{p}}[\boldsymbol{\Phi}\mathbf{X}^\star]_\mu + \sqrt{R_2(t,\epsilon)}V_\mu + \sqrt{\rho t - R_2(t,\epsilon) + 2s_n}A_\mu^\star\Big)\Big\}_{\mu=1}^m & \text{(73a)} \\[2mm] \tilde{\mathbf{Y}}_t = (R_1(t,\epsilon))^{1/2}\star\mathbf{X}^\star + \boldsymbol{\zeta}, & \text{(73b)} \end{cases}$$

where $V_\mu, A_\mu^\star \overset{\text{i.i.d.}}{\sim} \mathcal{N}_\beta(0,1)$, and $\boldsymbol{\zeta}\sim\mathcal{N}_\beta(0,\mathbb{1}_n)$. The prior distribution on $\mathbf{X}^\star$ is $P_0$. The rest of the proof is then a trivial verbatim of Sections D.1 to D.5.

## E    Proof of Proposition D.1

In this section, we prove Proposition D.1: we start from Conjecture 2.1 and derive eq. (41). Note that by (h2) we have $\langle\lambda\rangle_\nu = \alpha\mathbb{E}_{\nu_B}[X]/\delta$. We begin by recalling some sup-inf formulas, before turning to the actual proof.

### E.1    Some sup-inf formulas

We recall Corollary 8 of [9], stated here as a lemma:

**Lemma E.1** ([9]). *Let $f : \mathbb{R}_+ \to \mathbb{R}$ be a $\mathcal{C}^1$ convex, non-decreasing, Lipschitz function. Define $\rho \equiv ||f'||_\infty$. Let $g : [0,\rho] \to \mathbb{R}$ be a convex, non-decreasing, Lipschitz function. For $(q_1,q_2) \in \mathbb{R}_+ \times [0,\rho]$ we define $\psi(q_1,q_2) \equiv f(q_1) + g(q_2) - q_1q_2$. Then:*

$$\sup_{q_1\geq 0}\inf_{q_2\in[0,\rho]}\psi(q_1,q_2) = \sup_{q_2\in[0,\rho]}\inf_{q_1\geq 0}\psi(q_1,q_2).$$

We can state a corollary for functions of two variables.

**Corollary E.2.** *Let $f : \mathbb{R}_+^2 \to \mathbb{R}$ be a $\mathcal{C}^1$ convex, Lipschitz function which is nondecreasing in each of its variables. Define $\rho_1 \equiv ||\partial_1 f||_\infty, \rho_2 \equiv ||\partial_2 f||_\infty$. Let $g : [0, \rho_1] \to \mathbb{R}$, $g_2 : [0, \rho_2] \to \mathbb{R}$ be two convex, non-decreasing, Lipschitz functions. For $(x_1, x_2, y_1, y_2) \in \mathbb{R}_+^2 \times [0, \rho_1] \times [0, \rho_2]$ we define $\psi(x_1, x_2, y_1, y_2) \equiv f(x_1, x_2) + g_1(y_1) + g_2(y_2) - x_1 y_1 - x_2 y_2$. Then:*

$$\sup_{x_1, x_2 \geq 0} \inf_{y_1, y_2 \in [0, \rho_1] \times [0, \rho_2]} \psi(x_1, x_2, y_1, y_2) = \sup_{y_1, y_2 \in [0, \rho_1] \times [0, \rho_2]} \inf_{x_1, x_2 \geq 0} \psi(x_1, x_2, y_1, y_2).$$

*Proof of Corollary E.2.* The proof is a verbatim of the proof of Corollary 8 in [9], using that at fixed $y$, $x \mapsto f(x, y)$ is $\rho_1$-Lipschitz, while at fixed $x$, $y \mapsto f(x, y)$ is $\rho_2$-Lipschitz. $\qquad\square$

## E.2 Core of the proof

We now turn to the proof of Proposition D.1. We begin by simplifying the free entropy potential using the Gaussian prior. We start from Conjecture 2.1. Since $P_0$ is Gaussian by (h1), we can easily simplify the prior term $I_0$ as:

$$I_0(q_x) = \inf_{\hat{q}_x \geq 0} \left[ \frac{\beta \hat{q}_x (1 - q_x)}{2} - \frac{\beta}{2} \ln(1 + \hat{q}_x) \right] = \frac{\beta q_x}{2} + \frac{\beta}{2} \ln(1 - q_x).$$

We now turn to the term $I_{\text{int}}(q_x, q_z)$. We can write it as:

$$I_{\text{int}}(q_x, q_z) = \inf_{\gamma_x, \gamma_z \geq 0} \left[ \frac{\beta}{2}(1 - q_x)\gamma_x + \frac{\alpha\beta}{2}(Q_z - q_z)\gamma_z - \frac{\beta}{2}\langle \ln(1 + \gamma_x + \lambda\gamma_z)\rangle_\nu \right] \tag{74}$$
$$- \frac{\beta}{2}\ln(1 - q_x) - \frac{\beta q_x}{2} - \frac{\alpha\beta}{2}\ln(Q_z - q_z) - \frac{\alpha\beta q_z}{2Q_z}.$$

So we have, using Corollary E.2, that if $f \equiv \sup_{q_x \in [0,1]} \sup_{q_z \in [0, Q_z]} [I_0(q_x) + \alpha I_{\text{out}}(q_z) + I_{\text{int}}(q_x, q_z)]$ is the conjectured limit of the free entropy:

$$f = \sup_{q_x \in [0,1]} \sup_{q_z \in [0, Q_z]} \inf_{\gamma_x, \gamma_z \geq 0} \left[ \alpha I_{\text{out}}(q_z) + \frac{\beta}{2}(1 - q_x)\gamma_x + \frac{\alpha\beta}{2}(Q_z - q_z)\gamma_z \right.$$
$$\left. - \frac{\beta}{2}\langle \ln(1 + \gamma_x + \lambda\gamma_z)\rangle_\nu - \frac{\alpha\beta}{2}\ln(Q_z - q_z) - \frac{\alpha\beta q_z}{2Q_z} \right],$$
$$= \sup_{\gamma_x, \gamma_z \geq 0} \inf_{q_z \in [0, Q_z]} \inf_{q_x \in [0,1]} \left[ \alpha I_{\text{out}}(q_z) + \frac{\beta}{2}(1 - q_x)\gamma_x + \frac{\alpha\beta}{2}(Q_z - q_z)\gamma_z \right. \tag{75}$$
$$\left. - \frac{\beta}{2}\langle \ln(1 + \gamma_x + \lambda\gamma_z)\rangle_\nu - \frac{\alpha\beta}{2}\ln(Q_z - q_z) - \frac{\alpha\beta q_z}{2Q_z} \right].$$

The infimum on $q_x$ is very easily solved, as we have $\inf_{q_x \in [0,1]}[-\beta q_x \gamma_x/2] = -\beta\gamma_x/2$. Note that at fixed $\gamma_z \geq 0$, the variables $\gamma_x, q_z$ are completely decoupled in eq. (75), so we have $\sup_{\gamma_x} \inf_{q_z} = \inf_{q_z} \sup_{\gamma_x}$. This yields:

$$f = \sup_{\gamma_z \geq 0} \inf_{q_z \in [0, Q_z]} \sup_{\gamma_x \geq 0} \left[ \alpha I_{\text{out}}(q_z) + \frac{\alpha\beta}{2}(Q_z - q_z)\gamma_z \right.$$
$$\left. - \frac{\beta}{2}\langle \ln(1 + \gamma_x + \lambda\gamma_z)\rangle_\nu - \frac{\alpha\beta}{2}\ln(Q_z - q_z) - \frac{\alpha\beta q_z}{2Q_z} \right],$$
$$= \sup_{\gamma_z \geq 0} \inf_{q_z \in [0, Q_z]} \left[ \frac{\beta}{2}\left[ \alpha(Q_z - q_z)\gamma_z - \alpha\frac{q_z}{Q_z} - \langle \ln(1 + \lambda\gamma_z)\rangle_\nu - \alpha\ln(Q_z - q_z) \right] + \alpha I_{\text{out}}(q_z) \right].$$

Recall the form of $I_{\text{out}}$ in Conjecture 2.1 and that $\hat{Q}_z = 1/Q_z$. Using the form of $I_{\text{out}}$, we have with the notations of Proposition D.1:

$$f = \sup_{\gamma_z \geq 0} \inf_{q_z \in [0, Q_z]} \inf_{\hat{q}_z \geq 0} \left[ \frac{\beta}{2}\left[ \alpha(Q_z - q_z)\gamma_z - \alpha\frac{q_z}{Q_z} - \langle \ln(1 + \lambda\gamma_z)\rangle_\nu - \alpha\ln(Q_z - q_z) \right. \right.$$
$$\left. \left. - \alpha q_z \hat{q}_z - \alpha\ln(\hat{q}_z + 1/Q_z) + \alpha Q_z \hat{q}_z \right] + \alpha\Psi_{\text{out}}\left(\sqrt{Q_z^2 \hat{q}_z/(1 + Q_z \hat{q}_z)}\right) \right].$$

Again, we use that at fixed $q_z$, the variables $\hat{q}_z, \gamma_z$ are decoupled. So using again Lemma E.1, we have schematically $\sup_{\gamma_z} \inf_{q_z} \inf_{\hat{q}_z} = \sup_{q_z} \inf_{\hat{q}_z} \inf_{\gamma_z} = \sup_{\hat{q}_z} \inf_{\gamma_z} \inf_{q_z}$. We can then explicitly solve the infimum on $q_z$, which yields:

$$f = \sup_{\hat{q}_z \geq 0} \inf_{\gamma_z \geq 0} \left[ \frac{\beta}{2} \left[ - \langle \ln(1 + \lambda \gamma_z) \rangle_\nu + \alpha \ln(1 + \gamma_z(Q_z - q(\hat{q}_z))) \right] + \alpha \Psi_{\text{out}}(q(\hat{q}_z)) \right],$$

with

$$q(\hat{q}_z) \equiv \frac{Q_z^2 \hat{q}_z}{1 + Q_z \hat{q}_z}. \tag{76}$$

Note that $q$ is a strictly increasing smooth function of $\hat{q}_z$, with $q(0) = 0$ and $q(+\infty) = Q_z$. So we have:

$$f = \sup_{q \in [0, Q_z]} \inf_{\gamma_z \geq 0} \left[ \frac{\beta}{2} \left[ - \langle \ln(1 + \lambda \gamma_z) \rangle_\nu + \alpha \ln(1 + \gamma_z(Q_z - q)) \right] + \alpha \Psi_{\text{out}}(q) \right], \tag{77}$$

We then state a technical lemma:

**Lemma E.3.** *Under hypothesis (h2), one has for every $q \in [0, Q_z]$:*

$$\inf_{\gamma_z \geq 0} [\alpha \ln(1 + \gamma_z(Q_z - q)) - \langle \ln(1 + \lambda \gamma_z) \rangle_\nu] = \inf_{\hat{q} \geq 0} [\delta \hat{q}(Q_z - q) - \mathbb{E}_{\nu_B} \ln(1 + \hat{q} X)].$$

Using Lemma E.3 in eq. (77), and inverting the sup-inf by Lemma E.1 finishes the proof of Proposition D.1. In the remaining of the section we prove Lemma E.3

### E.3 Proof of Lemma E.3

If $q = Q_z$, the equality is trivially satisfied, so let us assume $0 \leq q < Q_z$. Let us denote $h(\gamma_z) \equiv \alpha \ln(1 + \gamma_z(Q_z - q)) - \langle \ln(1 + \lambda \gamma_z) \rangle_\nu$. Recall that $Q_z = \mathbb{E}_{\nu_B}[X]/\delta$. Since $\alpha \geq 1 - \nu(\{0\})$ and $q < Q_z$, one easily checks that $h$ is lower-bounded, so the infimum is always well-defined. We introduce $\mu$ the asymptotic measure of $\boldsymbol{\Phi\Phi}^\dagger/n$, and we denote $g_\mu(z) \equiv \langle (\lambda - z)^{-1} \rangle_\mu$ its Stieltjes transform. For every function $f$, one has $\langle f(\lambda) \rangle_\nu = \alpha \langle f(\lambda) \rangle_\mu + (1 - \alpha) f(0)$. This allows to write:

$$h(\gamma_z) = \alpha \ln(1 + \gamma_z(Q_z - q)) - \alpha \langle \ln(1 + \lambda \gamma_z) \rangle_\mu.$$

We will use the following equation, valid for every $\gamma_z \geq 0$ and any positively supported measure $\mu$:

$$\langle \ln(\gamma_z + \lambda) \rangle_\mu = \inf_{\tilde{\gamma}_z \geq 0} \left[ \gamma_z \tilde{\gamma}_z + \int_0^{\tilde{\gamma}_z} \mathcal{R}_\mu(-t) \mathrm{d}t - \ln \tilde{\gamma}_z - 1 \right], \tag{78}$$

in which $\mathcal{R}_\mu$ is the so-called "$R$-transform" of $\mu$, defined as $\mathcal{R}_\mu(-x) \equiv g_\mu^{-1}(x) + 1/x$. It is a classical result of random matrix theory [16] that if $\mu$ is positively supported, $t \mapsto \mathcal{R}_\mu(-t)$ is well-defined on $\mathbb{R}_+$. We finish the proof of Lemma E.3, before proving eq. (78). By a classical result of random matrix theory [15], we know the $R$-transform of $\mu$ as a function of $\nu_B$:

$$\mathcal{R}_\mu(-t) = \mathbb{E}_{\nu_B} \left[ \frac{X}{\delta + \alpha t X} \right]. \tag{79}$$

Combining eq. (78) and eq. (79), we reach:

$$\inf_{\gamma_z \geq 0} h(\gamma_z) = \inf_{\gamma_z \geq 0} \sup_{\tilde{\gamma}_z \geq 0} \left[ \alpha \ln(1 + \gamma_z(Q_z - q)) + \alpha - \alpha \frac{\tilde{\gamma}_z}{\gamma_z} + \alpha \ln \frac{\tilde{\gamma}_z}{\gamma_z} - \mathbb{E}_{\nu_B} \ln \left( 1 + \frac{\alpha}{\delta} X \tilde{\gamma}_z \right) \right].$$

Using Lemma E.1 to invert the inf-sup, we have:

$$\inf_{\gamma_z \geq 0} h(\gamma_z) = \inf_{\tilde{\gamma}_z \geq 0} \sup_{\gamma_z \geq 0} \left[ \alpha \ln(1 + \gamma_z(Q_z - q)) + \alpha - \alpha \frac{\tilde{\gamma}_z}{\gamma_z} + \alpha \ln \frac{\tilde{\gamma}_z}{\gamma_z} - \mathbb{E}_{\nu_B} \ln \left( 1 + \frac{\alpha}{\delta} X \tilde{\gamma}_z \right) \right].$$

The supremum on $\gamma_z$ is now completely tractable, and we have:

$$\inf_{\gamma_z \geq 0} h(\gamma_z) = \inf_{\tilde{\gamma}_z \geq 0} \left[ \alpha(Q_z - q) \tilde{\gamma}_z - \mathbb{E}_{\nu_B} \ln \left( 1 + \frac{\alpha}{\delta} X \tilde{\gamma}_z \right) \right].$$

Doing the replacement $\hat{q} \equiv \alpha\tilde{\gamma}_z/\delta$ yields Lemma E.3. We now prove eq. (78), which will finish the proof. It follows from a classical result used in random matrix theory, see e.g. [8] for an application of these calculations to spherical integrals. Recall that $g_\mu$ is smooth and strictly increasing on $(-\infty, 0)$, as $\mu$ is positively supported. It is easy to see by differentiation that the infimum in eq. (78) is attained at $\tilde{\gamma}_z = g_\mu(-\gamma_z)$. We then use some manipulations:

$$
\inf_{\tilde{\gamma}_z \geq 0}\left[\gamma_z\tilde{\gamma}_z + \int_0^{\tilde{\gamma}_z} \mathcal{R}_\mu(-t)\mathrm{d}t - \ln\tilde{\gamma}_z\right] = \gamma_z g_\mu(-\gamma_z) + \int_0^{g_\mu(-\gamma_z)} \mathcal{R}_\mu(-t)\mathrm{d}t - \ln g_\mu(-\gamma_z),
$$

$$
= \gamma_z g_\mu(-\gamma_z) + \int_\epsilon^{g_\mu(-\gamma_z)} g_\mu^{-1}(t)\mathrm{d}t - \ln\epsilon + \int_0^\epsilon \mathcal{R}_\mu(-t)\mathrm{d}t,
$$

this equation being valid for all $\epsilon > 0$ sufficiently small. By regularity of the $R$-transform around $0$ [16], $\int_0^\epsilon \mathcal{R}_\mu(-t)\mathrm{d}t = \mathcal{O}_\epsilon(1)$. Moreover, we can change variables in the other integral, and we reach:

$$
\inf_{\tilde{\gamma}_z \geq 0}\left[\gamma_z\tilde{\gamma}_z + \int_0^{\tilde{\gamma}_z} \mathcal{R}_\mu(-t)\mathrm{d}t - \ln\tilde{\gamma}_z\right] = \gamma_z g_\mu(-\gamma_z) + \int_{-g_\mu^{-1}(\epsilon)}^{\gamma_z} u g_\mu(-u)\mathrm{d}u - \ln\epsilon + \mathcal{O}_\epsilon(1),
$$

$$
\overset{(a)}{=} -\ln\epsilon - \epsilon g_\mu^{-1}(\epsilon) + \int_{-g_\mu^{-1}(\epsilon)}^{\gamma_z} g_\mu(-u)\mathrm{d}u + \mathcal{O}_\epsilon(1),
$$

$$
\overset{(b)}{=} 1 + \langle\ln(\lambda + \gamma_z)\rangle_\mu + \mathcal{O}_\epsilon(1),
$$

in which we used integration by parts in $(a)$ and the definition of the Stieltjes transform in $(b)$. Since $\epsilon$ was taken arbitrarily small, taking the limit $\epsilon \to 0$ ends the proof.

# F  Technical lemmas and definitions

## F.1  Some definitions

Let $\beta \in \{1, 2\}$. We denote $\mathbb{K} = \mathbb{R}$ if $\beta = 1$ and $\mathbb{K} = \mathbb{C}$ if $\beta = 2$. $\mathcal{U}_\beta(n)$ denotes the orthogonal (respectively unitary) group, and $\mathcal{S}_\beta(\mathbb{R}), \mathcal{S}_\beta^+(\mathbb{R})$ the space of *real* symmetric (resp. positive symmetric) matrices of size $\beta$. $\mathbb{1}_\beta$ is the identity matrix of size $\beta$. To improve clarity, we write $\mathrm{Tr}_\beta$ when taking the trace of a matrix in the space $\mathcal{S}_\beta(\mathbb{R})$. The standard Gaussian measure is defined on $\mathbb{K}$ as:

$$
\mathcal{D}_\beta z \equiv \left(\frac{\beta}{2\pi}\right)^{\beta/2} \exp\left(-\frac{\beta}{2}|z|^2\right) \mathrm{d}z. \tag{80}
$$

We define three different types of products in $\mathbb{K}$, using the identification $\mathbb{K} \simeq \mathbb{R}^\beta$.

$$
\begin{cases} zz' & \text{the usual product in } \mathbb{K}, & \text{(81a)} \\ z \cdot z' \equiv \mathrm{Re}[\bar{z}z'] & \text{the dot product in } \mathbb{R}^\beta. & \text{(81b)} \end{cases}
$$

For $\beta = 1$, and $M, z \in \mathbb{R}$, we also denote $M \star z \equiv Mz$. For $\beta = 2$, with $z = x + iy \in \mathbb{C}$, and $M \in \mathcal{S}_2$ written as:

$$
M \equiv a\mathbb{1}_2 + \begin{pmatrix} b & c \\ c & -b \end{pmatrix}, \tag{82}
$$

we define $M \star z$ as the matrix-vector product in $\mathbb{R}^\beta$:

$$
M \star z \equiv M\begin{pmatrix} x \\ y \end{pmatrix} = az + (b + ic)\bar{z}. \tag{83}
$$

Note that in the $\beta = 1$ case, all three products are equivalent.

## F.2  Conventions for derivatives

We often consider functions $f : \mathbb{K} \to \mathbb{R}$. The derivatives for such functions are defined in the usual sense if $\mathbb{K} = \mathbb{R}$, while for $\mathbb{K} = \mathbb{C}$ we set it in the "function of two variables" sense (with $z = x + iy$):

$$
f'(z) \equiv \partial_x f + i\partial_y f. \tag{84}
$$

We will also define its Laplacian if $\mathbb{K} = \mathbb{C}$ (if $\mathbb{K} = \mathbb{R}$ then $\Delta f(x) = f''(x)$):

$$\Delta f(z) \equiv \partial_x^2 f + \partial_y^2 f. \tag{85}$$

Importantly, this definition is different from the usual Wirtinger definition of a complex derivative, because we do not consider holomorphic functions here, but merely differentiable real functions of two variables. This definition satisfies the following chain rule formula, for $h(x) \equiv f(g(x))$ and $f : \mathbb{K} \to \mathbb{R}, g : \mathbb{R} \to \mathbb{K}$:

$$h'(x) = g'(x) \cdot f'(g(x)). \tag{86}$$

As a particular case, we have if $f(x) = x \cdot z$ that $f'(x) = z$. We then have the Stein lemma (or Gaussian integration by parts), for any $\mathcal{C}^2$ function $f : \mathbb{K} \to \mathbb{R}$:

$$\int \mathcal{D}_\beta z \, (z f(z)) = \frac{1}{\beta} \int \mathcal{D}_\beta z \, f'(z), \tag{87}$$

$$\int \mathcal{D}_\beta z \, (z \cdot f'(z)) = \frac{1}{\beta} \int \mathcal{D}_\beta z \, \Delta f(z). \tag{88}$$

### F.3 Nishimori identity

We state here the Nishimori identity, a classical consequence of Bayes optimality.

**Proposition F.1** (Nishimori identity). *Let $(X, Y)$ be random variables on a Polish space $E$. Let $k \in \mathbb{N}^\star$ and $(X_1, \cdots, X_k)$ i.i.d. random variables sampled from the conditional distribution $\mathbb{P}(X|Y)$. We denote $\langle \cdot \rangle_Y$ the average with respect to $\mathbb{P}(X|Y)$, and $\mathbb{E}[\cdot]$ the average with respect to the joint law of $(X, Y)$. Then, for all $f : E^{k+1} \to \mathbb{K}$ continuous and bounded:*

$$\mathbb{E}[\langle f(Y, X_1, \cdots, X_k) \rangle_Y] = \mathbb{E}[\langle (Y, X_1, \cdots, X_{k-1}, X) \rangle_Y]. \tag{89}$$

*Proof of Proposition F.1.* The proposition arises as a trivial consequence of Bayes' formula:

$$\mathbb{E}[\langle f(Y, X_1, \cdots, X_{k-1}, X) \rangle_Y] = \mathbb{E}_Y \mathbb{E}_{X|Y}[\langle f(Y, X_1, \cdots, X_{k-1}, X) \rangle_Y],$$
$$= \mathbb{E}_Y[\langle f(Y, X_1, \cdots, X_k) \rangle_Y].$$

$\square$

### F.4 Boundedness of an overlap fluctuation

**Lemma F.2** (Boundedness of an overlap fluctuation). *Under $(H0)$, one can find a constant $C > 0$ independent of $n, t, \epsilon$ such that for any $r \geq 0$:*

$$\mathbb{E}\Big\langle \Big| \frac{1}{n} \sum_{\mu=1}^m u'_{Y_{t,\mu}}(S_{t,\mu})^\dagger u'_{Y_{t,\mu}}(s_{t,\mu}) - \beta^2 \delta r \Big|^2 \Big\rangle_{n,t,\epsilon} \leq 2\beta^4 \delta^2 r^2 + C. \tag{90}$$

*Proof of Lemma F.2.* We directly have:

$$\mathbb{E}\Big\langle \Big| \frac{1}{n} \sum_{\mu=1}^m u'_{Y_{t,\mu}}(S_{t,\mu})^\dagger u'_{Y_{t,\mu}}(s_{t,\mu}) - \beta^2 \delta r \Big|^2 \Big\rangle_{n,t,\epsilon}$$

$$\leq 2\beta^4 \delta^2 r^2 + 2\mathbb{E}\Big\langle \Big| \frac{1}{n} \sum_{\mu=1}^m u'_{Y_{t,\mu}}(S_{t,\mu})^\dagger u'_{Y_{t,\mu}}(s_{t,\mu}) \Big|^2 \Big\rangle_{n,t,\epsilon}$$

We can bound $|u'_{Y_{t,\mu}}(s)|$ for any $s \in \mathbb{K}$ by using the formulation of the channel described in eq. (2), which allows to formally write:

$$u'_{Y_{t,\mu}}(s) = \lim_{\Delta \downarrow 0} \frac{\int P_A(\mathrm{d}a) \partial_s \varphi_{\mathrm{out}}(s, a)(Y_{t,\mu} - \varphi_{\mathrm{out}}(s, a)) e^{-\frac{1}{2\Delta}(Y_{t,\mu} - \varphi_{\mathrm{out}}(s,a))^2}}{\int P_A(\mathrm{d}a) e^{-\frac{1}{2\Delta}(Y_{t,\mu} - \varphi_{\mathrm{out}}(s,a))^2}},$$

in which we used a Gaussian representation of the delta distribution. This amounts to add a small Gaussian noise to the model of eq. (2), and effectively write it as:

$$Y_\mu \sim \varphi_{\mathrm{out}}(S_\mu, A_\mu) + \sqrt{\Delta} Z'_\mu, \tag{91}$$

with $Z'_\mu \overset{\text{i.i.d.}}{\sim} \mathcal{N}(0,1)$, and then take the $\Delta \to 0$ limit. We have $|Y_{t,\mu}| \le \|\varphi_{\text{out}}\|_\infty + \sqrt{\Delta}|Z'_\mu|$, and thus taking $\Delta \to 0$ we reach:

$$|u'_{Y_{t,\mu}}(s)| \le 2\,\|\varphi_{\text{out}}\|_\infty\,\|\partial_s \varphi_{\text{out}}\|_\infty\,.$$

The right-hand side of the last inequality is bounded by hypothesis $(H0)$, and in the end, we have:

$$\mathbb{E}\left\langle \left| \frac{1}{n} \sum_{\mu=1}^m u'_{Y_{t,\mu}}(S_{t,\mu})^\dagger u'_{Y_{t,\mu}}(s_{t,\mu}) - \beta^2 \delta r \right|^2 \right\rangle_{n,t,\epsilon} \le 2\beta^4 \delta^2 r^2 + 2^5\,\|\varphi_{\text{out}}\|_\infty^4\,\|\partial_s \varphi_{\text{out}}\|_\infty^4\,,$$

which ends the proof. $\qquad\square$

## Footnotes

[1]For a mismatched model, the replica symmetry assumption, discussed below, is generically not valid.

[1]This relation is valid even if $\lambda_x$ would "saturate" to a constant value that does not depend on $(q_x, q_z)$.

[1]Our conventions for derivatives of real functions of complex variables are reminded in Section F.2.

[1]Recall the definition of $\mathcal{Z}_{n,t,\epsilon}$ in eq. (49).

[1]Notice in particular that the first equation of eq. (65) implies that the derivative $\partial_t R_1(t,\epsilon)$ is always a diagonal matrix in $\mathcal{S}_\beta(\mathbb{R})$.