[Reviews · NeurIPS 2020]

Review 1

Summary and Contributions: This paper analyzes the performance of generalized vector-approximate message-passing (G-VAMP) on phase retrieval problem. Main contributions are 1) rigourous characterization of the algorithm for certain non-trivial cases of measurement matrix 2) characterize the weak recover threshold and strong recover threshold for different designs of measurement matrix.

Strengths: Theorems look sound and I believe the significance of the paper is somewhat around or above the borderline. The phase retrieval problem regarded as non-convex optimization problem is definitely related to NeurIPS community.

Weaknesses: One major weakness of the paper is that the rigourous analysis on the performance of G-VAMP is limitedly conducted to the case when either the measurement matrix is Gaussian or unknown signal is Gaussian. Also, the analysis of weak and strong threshold is also conducted on special cases and there is no unified framework to include all these special cases. One minor weakness (mainly due to the algorithm itself) is that G-VAMP as I understand require certain knowledge about the distribution of the true signal. I suggest the authors should point it out and present the algorithm as well for the convenience of the readers who are not familiar with AMP literature. After response: My appology for misunderstanding the importance of the results. I rise my score to 7.

Correctness: I am not very familiar with replica method and the appendix (27pages) is too long for me to check the proofs carefully in a very shot time. I applogize that I can not be certain whether the proofs are correct.

Clarity: It is clear to read the main paper.

Relation to Prior Work: The discussion of the related work looks good to me. One suggestion is to cite Approximate message passing for amplitude based optimization or Optimization-Based AMP for Phase Retrieval: The Impact of Initialization and Regularization, both of which are using AMP algorithm to solve phase retrival problem and thus closely related to the topic of this paper.

Reproducibility: Yes

Additional Feedback: I think this paper (if the proofs are all correct) is somewhere between 6-7. It would be 7 if all the results are rigourous or there is a unified framework for the analysis of weak and strong threshold. It would be a 8 or 9 if both of them are resolved.


Review 2

Summary and Contributions: This paper extends, generalizes, and unifies known results for phase retrieval in high dimensions. A general conjecture is presented that provides a single letter characterization of the free energy. This conjecture is proved rigorously in certain cases. The G-VAMP, conjectured to be optimal in many settings, is employed to estimated the signal. From this the authors are able to determine algorithmic thresholds for both weak and full recovery and determine information theoretic thresholds for full recovery.

Strengths: Soundness: the theoretical and empirical results appear sound to me Significance and novelty: phase retrieval is a significant problem. The main novelty is combining a variety of existing results in a common framework. And then introducing new results based on the common framework. These new results come in two forms: information theoretic thresholds for full recovery based determined rigorously by proving the conjecture under different conditions and algorithmic thresholds for weak and strong recovery based on the analysis of the G-VAMP algorithm. Relevance to NeurIPS: definitely relevant

Weaknesses: Soundness: no weakness that I can determine Significance and novelty: no weakness that I can determine Relevance to NeurIPS: no weakness

Correctness: The proofs are in a 27 page appendix. Having said that the theoretical results are believable because they follow a common recipe for the replica computation and the interpolation method for proving correctness of the replica computation.

Clarity: The paper is well written. Having said that it is a challenge to read given the abundance of notation and the variety of results mentioned.

Relation to Prior Work: Yes, prior work is discussed and compared to this work. As mentioned above this paper attempts to consolidate known results in a general framework and then extend those results.

Reproducibility: Yes

Additional Feedback:


Review 3

Summary and Contributions: The authors discuss the fundamental phase retrieval problem where for i=1,..,m Y_i=|<\Phi_j,x^*>| where \Phi_j, x^* are n-dimensional. They study the statistical and algorithmic reconstruction problem of x^* under various random generating assumptions on the model. The setting is when m/n=\alpha (constant) and m,n diverge to infinity. The authors first perform the non-rigorous replica method which gives rise to a conjectured on the optimal reconstruction of x^*. This conjecture applies to a large family of sensing matrices \Phi (real and complex) - much bigger from existing results- and separable priors on x^*. The authors prove the conjecture in two relatively general cases for the generative models of \Phi and x^*, yet far from the whole family the conjecture is applied to. The authors use the conjecture to derive the full-recovery IT limit of the problem (min \alpha for which full recovery of x^* is possible) as well the the algorithmic weak-recovery (min \alpha for which G-AMP achieves positive correlation with x^*). The latter result is possible as in this regime G-AMP performs gradient ascent in the replica potential started from zero. Finally, the authors provide simulations in which, among other remarks, an interesting all-or-nothing phase transition seems to be appearing.

Strengths: I think the paper is very interesting for the following reasons. 1. The replica-method is a heuristic that has been traditionally applied under a Gaussian assumption on the sensing matrix. While the authors are not the first who apply it in a non-strictly gaussian setting for phase retrieval, they apply it in the complex case under minimal assumptions on the spectrum of the sensing matrix. 2. The authors partially prove it, which appears to be a non-trivial task, even in the restricted setting they consider. Naturally, they are based on recent proof of predictions from the replica formula. 3. The authors show that (based on the Conjecture) there is potentially a statistical-computational gap in this generic setting for the sensing matrix. This is novel and interesting as compared to the standard case people have studied where the matrix is considered Gaussian.

Weaknesses: 1. I think the paper is not very clearly stating which results are rigorous and which are not. For example, I think it will be really nice if the authors explicitly mention that their derivations on Sections 3,4 are based on the Conjecture from the replica method and not any theorem (that is correct, right?) Furthermore, mentioning clearly which results are rigorous and which are not will help -hopefully - the future researchers in the field when they are reading the present work. I am also confused on this about Table 1, and the cases where \Phi=W_1W_2. Is W_1, W_2 random iid Gaussian here? If yes, how does this connect with Theorem 2.2. where W_2 needs to be fixed? 2. In the figures, saying "a real column-orthogonal matrix" does not read nicely. Please provide some details in the caption. 3. I am puzzled in the simulations what is the prior distribution. Is it a complex Gaussian when the matrix is complex and real otherwise? 4. On that note, in the abstract \alpha provides a distinction between the two fields: real and complex, something that becomes \beta in the main text. 5. I would suggest avoiding the use of the wording "the algorithmic threshold", as we have no guarantee that G-AMP is the optimal polynomial-time algorithm, right? 6. While the authors discuss extensively the \alpha_{WR,Alh} and the \alpha_{FR,IT} I am confused why they did not discuss about the \alpha_{WR,IT}? I think the paper will be more complete with such a study and discussion and given the Conjecture possibly such a characterization is not hard to achieve. 7. I got lost when reading the paper whether the authors *prove* that there is a gap between the algorithmic recovery threshold and the IT one. If yes, could they provide an explicit discussion under which assumption they prove or conjecture it to be true?

Correctness: As far as I have checked, the paper seems to be correct. As mentioned above, I think it will strongly benefit by clearly mentioning which parts are rigorous and which are not.

Clarity: The paper is clear to me, modulo the above comments.

Relation to Prior Work: I think the paper is clearly discussing the relation to prior work.

Reproducibility: Yes

Additional Feedback: A comment I wanted to make is about the "all-or-nothing" phase transition mentioned in line 89. This is potentially a very interesting finding, as to the best of my knowledge albeit a recent line of work initiated by [Gamarnik et al COLT 2017, Reeves et al 'COLT 2019] on all-or-nothing for sparse linear regression, the phenomenon has never been established in a "dense" setting like the present one. I think it will be very interesting for the authors to mention more clearly under which assumptions they see this transition (on the prior of the signal, sensing matrix etc) as this can be actually another contribution of the present work. Of course, an extensive study of this can be a topic of a future project. Finally a few corrections: Line 32, "fixed" instead of "finite". Line 44, "polynomial-time" instead of "polynomial"


Review 4

Summary and Contributions: The authors analyze the fundamental limits of phase retrieval under large random matrices, both real and complex-valued. They also analyze G-VAMP algorithm in the same context and show a few empirical performance results.

Strengths: The rigorous performance analysis is impressive. It generalizes several previous results and thus extends the frontier of theoretical understanding about phase retrieval with large random matrices.

Weaknesses: The theory applies to large random matrices and iid signals, not the Fourier matrices and natural images that one encounters in practice. It is encouraging that the empirical experiments show decent performance for randomly sampled Fourier matrices and iid signals, but I don't expect that these results will extend to natural images. For these reasons, and because phase retrieval is not of mainstream interest at Neurips, I worry that not too many Neurips attendees will be interested in this work. It may be better appreciated at a theoretical conference.

Correctness: Yes, the claims appear to be correct, although I did not check the proofs in detail.

Clarity: Yes, the paper is well written.

Relation to Prior Work: Yes, the authors do a good job of discussing how their work differs from, and improves upon, existing papers.

Reproducibility: Yes

Additional Feedback: ** I have read the other reviews and the author's response, and my opinion remains the same. **

[Author Response · NeurIPS 2020]

We thank all referees for their interest in our work and their comments that will help to clarify our paper.

**R1 -** **"There is no unified framework for the analysis of weak and strong thresholds."** We respectfully disagree: $\alpha_{\mathrm{FR,IT}}$ is in general not well-defined for an arbitrary channel, which motivates our restriction to the noiseless case. Our analysis is valid for any right-orthogonally invariant data matrix $\Phi$ with well-defined asymptotic density, and a Gaussian prior (which is limiting, although we believe it can be relaxed and we will work towards this). Concerning $\alpha_{\mathrm{WR,Algo}}$, eq. (11) holds in full generality, i.e. for any data matrix $\Phi$ as above, and any phase-retrieval probabilistic channel. Eqs.(12),(13) are examples derived from this generic formula. We will clarify on the generality of our results in the revised version. **"The rigorous analysis relies on some Gaussianity, either in the prior or in the data matrix"**. The referee classifies this as a major weakness: while we agree this is a restrictive assumption, it is a fundamental limitation of the interpolation method used for the proof, which will be clarified. We wish to indicate the reviews of **R2** and **R4**, that we thank for underlining the generality of our framework and of our rigorous analysis. **"One weakness is that G-VAMP requires knowledge of the distribution of the true signal"**. We would like to emphasize that the algorithm is also well-defined beyond this scope, e.g. it can be used to infer natural images with Fourier matrices. Using a Gaussian prior to infer is actually the minimal assumption on the underlying signal, as it amounts to simply fix its norm: we see this as a strength of our theory, which can predict the G-VAMP performance for any signal, structured or not. We discuss this further in the response to **R4**, and we will clarify this point. We finally thank the referee for pointing out typos, and providing additional references that we will add to the paper.

**R2 -** We thank the referee for her/his appreciation of our work.

**R3 -** **"The paper is not very clearly stating which results are rigorous + Confusion on the product of Gaussians"**. Our analysis in Sections 3-4 relies on Conjecture 2.1, and is thus rigorous whenever the conditions for Theorem 2.2 hold. In this theorem, the matrix $\mathbf{B}$ can be random or deterministic, as long as it satisfies the assumptions of the theorem, which is the case for $\mathbf{B} = \mathbf{W}_2$ i.i.d. Gaussian. We acknowledge this should be clarified in the text, and we hope this will answer the question of the referee. **'The conjectured optimality of G-VAMP"**. Indeed we refer to the G-VAMP threshold as "algorithmic", even if a proof for the optimality of G-VAMP is not given. We adopted this notation for consistency with the previous literature on this topic, in which this conjectured optimality is often assumed. We will add a note on this choice on the paper. **"Do the authors prove the existence of a gap ?"**. We emphasize that we provide scalar equations (rigorous when in the setting of Theorem 2.2) that can be used to find $\alpha_{\mathrm{FR,Algo}}$. Apart from the inevitable numerical solution, our analysis is thus well-controlled. This discussion will be added in the paper, and we thank the referee for helping clarify this point. **"Can we analyze $\alpha_{\mathrm{WR,IT}}$ ?"**. Extending our analysis to $\alpha_{\mathrm{WR,IT}}$ is an interesting open direction, which requires understanding the appearance of a global maximum in the replica-symmetric potential, but not necessarily continuously from the $q=0$ solution as in the case of $\alpha_{\mathrm{WR,Algo}}$. At the moment we are not able to carry such an analysis, and we will discuss this more extensively in the revised paper. **"On the all-or-nothing transition"**. We have observed these transitions for orthogonal/unitary matrices : as stated in the paper, we expect to see this phenomenon for other real matrices. We will discuss this further, as well as the possible dependency on the prior, and add the references provided. We finally thank the referee for the list of typos and comments, all of which will be addressed in the revised version.

**R4 -** **"I don't expect these results to extend to natural images"**. We thank the referee for indicating towards such an analysis, which would be a valuable add to our work. We conducted a simple experiment on a natural image, and the result is given in Fig. 1. Although we are far from a Bayes-optimal setting, the achieved MSE is very close to values of Fig. 2 of the paper, for all values of $\alpha$. In particular, we achieve perfect recovery for $\alpha \geq 2.3$, just above $\alpha_{\mathrm{FR,Algo}} \simeq 2.27$ which was derived for random unitary matrices, i.i.d. data and in the Bayes-optimal setting. As all normalized signals are equivalent under a Gaussian prior (it is a "maximum-entropy" prior), we indeed expect a structured signal to perform exactly as a random one as long as one also infers the signal using a Gaussian prior. This observation is coherent with Fig. 1 amd strengthens the relevance of our theoretical results for real data, and we will discuss this point further in the final version. We point out previous works that investigated the performance of AMP algorithms in phase retrieval [1, 2].

Figure 1: Performance of the G-VAMP algorithm for noiseless phase retrieval. We wish to recover a 77x102 image (on the left), and we use a complex Gaussian prior to infer the signal. The data matrix $\Phi$ is a randomly subsampled DFT matrix.

[1] Junjie Ma, Ji Xu, and Arian Maleki. Optimization-based amp for phase retrieval: The impact of initialization and l2 regularization. *IEEE Transactions on Information Theory*, 65(6):3600–3629, 2019.

[2] Philip Schniter and Sundeep Rangan. A message-passing approach to phase retrieval of sparse signals. In *Excursions in Harmonic Analysis, Volume 4*, pages 177–204. Springer, 2015.


[Meta-Review · NeurIPS 2020]

All reviewers recommend acceptance and I agree with them as this is a problem of interest to the community. I advise the authors to carefully revise the paper and incorporate suggestions from the reviewers.